# Exploring tracer information in a small stream to improve parameter identifiability and enhance the process interpretation in transient storage models

Enrico Bonanno[1,2], Günter Blöschl[2], Julian Klaus[3]

[1] Catchment and Eco-Hydrology Group, Luxembourg Institute of Science and Technology, Belvaux, Luxembourg.

[2] Institute of Hydraulic and Water Resources Engineering, Vienna University of Technology, Vienna, Austria.

[3] Institute of Geography, University of Bonn, Bonn, Germany.

*Correspondence to:* Enrico Bonanno (bonanno@hydro.tuwien.ac.at)

**Abstract**

The transport of solutes in river networks is controlled by the interplay of processes such as in-stream solute transport and the exchange of water between the stream channel and dead zones, in-stream sediments, and adjacent groundwater bodies. Transient storage models (TSMs) are a powerful tool for testing hypotheses related to solute transport in streams. However, model parameters often do not show a univocal increase of model performances in a certain parameter range (i.e. are non-identifiable) leading to an unclear understanding of the processes controlling solute transport in streams. In this study, we increased parameter identifiability in a set of tracer breakthrough experiments by combining global identifiability analysis and dynamic identifiability analysis in an iterative approach. We compared our results to inverse modelling approaches (OTIS-P) and the commonly used random sampling approach for TSMs (OTIS-MCAT). Compared to OTIS-P, our results informed about identifiability of model parameters on the entire feasible parameter range. Our approach clearly improved parameter identifiability compared to the standard OTIS-MCAT application, due to the progressive reduction of the investigated parameter range with model iteration. Non-identifiable results led to solute retention times in the storage zone and the exchange flow with the storage zone with a difference of up to four and two orders of magnitude compared to results with identifiable model parameters, respectively. The clear differences in the transport metrics between results obtained from our proposed approach and results from the classic random sampling approach also resulted in contrasting interpretation of the hydrologic processes controlling solute transport in a headwater stream in western Luxembourg. Thus, our outcomes point to the risks of interpreting TSM results when even one of the model parameters is non-identifiable. Our results showed that coupling global identifiability analysis with dynamic identifiability analysis in iterative approach clearly increased parameter identifiability in random sampling approaches for TSMs. Compared to the commonly used random sampling approach and inverse modelling results, our analysis was effective in obtaining higher accuracy of the evaluated solute transport metrics, which is advancing our understanding of hydrological processes that control in-stream solute transport.

## 1 Introduction

It is of crucial importance to understand how nutrients, solutes, and pollutants are transported in streams, since
this process can drastically affect stream water quality along river networks (Smith, 2005; Krause et al., 2011;
Rathfelder, 2016). A widely used technique to capture and study the processes controlling water transport
downstream is via in-stream tracer injections. The measurement of the concentration over time of a tracer released
in an upstream section (i.e. the breakthrough curve, BTC) reflects stream discharge (Beven et al., 1979;
Butterworth et al., 2000) and longitudinal tracer advection and dispersion (Gooseff et al., 2008). A milestone in
the study of solute transport was that in-stream solutes and water are exchanged with slowly-moving channel
waters, the dead zones (Hays, 1966), and with the saturated area that is physically influenced by water and solutes
exchange between the stream channel and the adjacent groundwater (i.e., the hyporheic zone, Triska et al., 1989;
White, 1993; Cardenas and Wilson, 2007). This hydrologic exchange results in a skewed non-Fickian BTC with
a pronounced tail, which makes the advection-dispersion equation (ADE) unable to correctly describe the
observed tracer transport in stream channels (Bencala and Walters, 1983; Castro and Hornberger, 1991). Despite
the large amount of studies, the results of TSM offer numerous contradictory model interpretation (Ward and
Packman, 2019), and model parameters are often non-identifiable, meaning that several parameter combination
return same model performances (Ward et al., 2017). These outcomes raise the question about how informative
such modelling results are (Knapp and Kelleher, 2020).

Considerable potential in reducing uncertainty of the processes controlling solute transport in streams lies in
modelling the tail of the BTC, since it contains information on the transient storage of the stream channels
(Bencala et al., 2011). For simulating the retentive effect of dead zones on solute transport, Hays (1966) modelled
the tail of the BTC by introducing a second differential equation in addition to the ADE. Following a similar
approach, Bencala and Walters (1983) described the solute transport in streams as a pure advection-dispersion
transport, coupled with a hydrologic exchange term between the stream channel and a single, homogeneously
mixed volume that delays the solute movement downstream (Transient Storage Model - TSM). The estimation of
model parameters often relies on the use of inverse modelling approaches via nonlinear regression algorithms that
return an estimation of model parameters with a narrow 95% confidence interval (OTIS-P; Runkel, 1998). While
this approach was widely applied in past decades, it does not allow a comprehensive assessment of parameter
identifiability (Ward et al., 2017; Knapp and Kelleher, 2020). The term "identifiability" describes whenever good
model performances are constrained in a relatively narrow parameter range (identifiable parameter) or spread
(non-identifiable parameter) across the entire distribution of the possible parameter values (Ward et al., 2017).
Yet, a good fit to observed data through inverse modelling does not provide information on performances and
parameter identifiability over the entire feasible parameter range (Ward et al., 2017). Also, calibrated parameter
obtained via inverse modelling approach are not necessarily meaningful, as non-identifiable parameters can
provide a good inverse model fit (Kelleher et al., 2019). These modelling uncertainties have led to a progressive
abandonment of the search for a single best set of parameters and advocated the identification of "behavioural"
parameter populations (i.e. parameter sets satisfying certain performance thresholds) via random sampling
approaches in transient storage modelling (Wlostowski et al., 2013; Ward et al., 2017; Ward et al., 2018; Kelleher
et al., 2019; Rathore et al., 2021).

Random sampling approaches provide information on parameter identifiability on the feasible parameter range,
however they rarely show identifiability for all the model parameters (Knapp and Kelleher, 2020). Kelleher et al.

(2013) found that the parameters associated with the transient storage process are not identifiable for a large variety of stream reaches and experiments that they investigated. Other studies have shown that model parameters are often poorly identifiable (Camacho and González, 2008; Wlostowski et al., 2013; Ward et al., 2017; Kelleher et al., 2019; Knapp and Kelleher, 2020) and highly interactive, meaning that different parameters can produce similar modelled BTCs (Kelleher et al., 2013). This, in turn, hampers the ability to distinguish the role of a specific parameter on the shape of the simulated BTC (Wagener et al., 2002; Ward et al., 2017).

The observed strong non-identifiability of model parameters in random sampling studies may have three causes. First, the parameters describing the advection-dispersion process (streamflow velocity, cross-sectional area of the stream channel, and longitudinal dispersion) are known to be the best identifiable in the TSM (Ward et al., 2017). However, due to the known high interactivity among model parameters, it is generally not recommended to use a fixed value for a rather identifiable parameter, since this strategy may result in a mis-estimation of the other model parameters (Knapp and Kelleher, 2020). Constraining the values of the stream area and longitudinal dispersion proved to have a role on the identifiability of transient storage parameters (Lees et al., 2000; Kelleher et al., 2013; Ward et al., 2017). However, no study so far evaluated the role of flow velocity on the identifiability of model parameters despite the velocity parameter was often considered to be known and thus fixed to equal the velocity of the arrival time of the BTC peak (i.e. $v_{peak}$, Ward et al., 2013; Kelleher et al., 2013; Wlostowski et al., 2017; Ward et al., 2017; Ward et al., 2018). This leads to the question on how meaningful, and identifiable the transient storage parameters are when streamflow velocity is considered as a calibration parameter or is kept fixed in identifiability analysis. A second cause for non-identifiable model parameters relates to the selected approach for addressing parameter identifiability. The identifiability analysis used in most studies is based on the Generalized Likelihood Uncertainty Estimation that assesses parameter certainty by evaluating model performance on the entire BTC (GLUE, Beven and Binley, 1992; Camacho and González, 2008; Kelleher et al., 2013; Ward et al., 2017; Kelleher et al., 2019). However, such global identifiability analysis is unable to assess if a certain parameter is more or less identifiable in certain sections of the BTC (Wagener et al., 2002; Wagener et al., 2003; Wagener and Kollat, 2007). This information is particularly important for BTC modelling, since advection-dispersion parameters are physically responsible for the bulk solute transport in the stream and they are therefore expected to act on the rising limb and peak of the BTC (Gooseff et al., 2008). Contrary, the parameters describing the exchange between the stream channel and the transient storage zone are responsible for delaying solute transport compared to the advective-dispersive transport, most likely acting on the falling limb and tail of the BTC (Runkel, 2002). By investigating parameter identifiability across the entire BTC, global identifiability analysis is unable to capture an increase in parameter identifiability towards the tail of the BTC. However, studies addressing the identifiability of model parameters in different sections of the BTC reported an increased identifiability for transient storage parameters on the tail of the BTC (Wagener et al., 2002; Scott et al., 2003; Wlostowski et al., 2013; Kelleher et al., 2013). Third, there is no common strategy for selecting parameter ranges and the number of parameter sets in TSM simulations. To obtain reliable results, Ward et al. (2017) suggested that modelling studies need to apply TSM on a large number of parameter sets (between 10,000 and 100,000) over a parameter range spanning at least two orders of magnitude. While for some studies the non-identifiability of parameters might be explained by the low number of parameter sets (less than 10,000) and the relatively narrow selected parameter range (Wagener et al., 2002; Camacho and González, 2008; Wlostowski et al., 2013), non-identifiability was also found when a rather large number of parameter sets and wide parameter range were used (Kelleher et al., 2013;

Ward et al., 2017; Kelleher et al., 2019). This is bringing up the question *if* and *when* model parameters are actually meaningful (Knapp & Kelleher, 2020).

A robust assessment of transient storage parameters would not only improve the model fit of tracer transport and increase parameter identifiability, but it might also lead to a more robust interpretation of the physical processes controlling solute transport in streams. Model parameters are often used to calculate metrics on the solute exchange between the stream channel and the transient storage zone and the residence time of solutes in the coupled system (Thackston and Schnelle, 1970; Morrice et al., 1997; Hart et al., 1999; Runkel, 2002). These metrics are pivotal to address the potential for nutrient cycling, microbial activity, and the development of hot-spots in river ecosystems (Triska et al., 1989; Mulholland et al., 1997; Smith, 2005; Krause et al., 2017). However, no study so far indicated and evaluated *if* and *how much* the interpretation of hydrologic processes changes when model parameters are identifiable and when they are not, due to the enunciated challenges in TSMs.

Despite the increasing need for achieving parameter identifiability in TSMs, only few studies have explored the reliability of results obtained from inverse modelling, and model interpretation is often based on a single set of parameters without testing their robustness (Knapp and Kelleher, 2020). We hypothesise that addressing the identifiability of model parameters in different sections of the BTC is key in increasing the identifiability of the parameters describing solute retention in streams. To address the enunciated TSM challenges, we have organised this contribution around three questions related to the key challenges of parameter identifiability in transient storage modelling:

1) How does the identifiability of model parameters change in the random sampling of TSM when velocity is considered as a calibration parameter and when it is assumed fixed and equal to $v_{peak}$?

2) Does the identifiability analysis on specific sections of the BTC reduce the parameter non-identifiability in random sampling of TSM?

3) How much does the identifiability of model parameters in random sampling approaches depend to the used parameter range and on the number of parameter sets?

With the outcomes of these questions we will address:

4) How does the hydrologic interpretation of TSM results vary when model parameters are identifiable and when they are not?

**2 Study site and methods**

**2.1 Study site and tracer data**

The studied stream reach (49°49'38"N, 5°47'44"E) is located in western Luxembourg, downstream of the Weierbach experimental catchment (Hissler et al., 2021; Fabiani et al., 2021). The stream channel is unvegetated with a slope of ≃6% and consists of deposited colluvium material and fragmented schists (up to 50 cm depth) with local outcrops of fractured slate bedrock in the streambed. The flow regime is governed by the interplay of seasonality between precipitation and evapotranspiration (Rodriguez and Klaus, 2019; Rodriguez et al., 2021) with a persistent discharge between autumn and spring, and little to no discharge during summer months (discharge arithmetic mean equal to 6.5 l/s, median of 1.7 l/s, St.Dev. of 11.52 l/s between Aug 2018 and Feb 2020; Bonanno et al., 2021). To answer our research questions, we utilise three tracer experiments with an instantaneous tracer injection at three different flow ($Q$) conditions: 6[th] December 2018, $Q$ = 2.52 l/s (E1); 23[rd] January 2019, $Q$ = 9.05 l/s (E2); 28[th] January 2019, $Q$ = 22.79 l/s (E3). For each experiment, we prepared a NaCl

solution using 2 l of stream water and 100 g of reagent-grade NaCl. We injected the solution into a turbulent pool at the beginning of the stream reach to assure complete mixing in the stream water. Electric conductivity (EC) was measured via a portable conductivity meter (WTW) 55 m downstream of the injection point. Automatic compensation of stream temperature occurred (nLF, according to EN 27 888). EC-Cl$^-$ conversion was obtained using a known-volume sample of stream water taken before tracer injection at the measurement location and adding known quantities of a solution with a known concentration of Na-Cl. Conversion into Cl$^-$ concentration was obtained via an EC-Cl$^-$ regression line ($R^2 = 0.9999$). Discharge was calculated for every slug injection via the dilution gauging method using the Cl$^-$ concentration obtained for each BTC (Beven et al., 1979; Butterworth et al., 2000).

**2.2 Advection-dispersion equation and Transient Storage Model formulation**

The one-dimensional Fickian-type advection and dispersion equation describes the combined effect of flow velocity and turbulent diffusion on solute transport (Beltaos and Day, 1978; Taylor, 1921, 1954). The differential form of ADE reads:

$$\frac{\partial C}{\partial t} = -v\frac{\partial C}{\partial x} + \frac{1}{A}\frac{\partial}{\partial x}\left(AD\frac{\partial C}{\partial x}\right) \qquad \text{Eq.1}$$

Where $t$ is time [T], $x$ is the distance from the injection point along the stream reach [L], $A$ [L$^2$] is the cross-sectional area of flow, $v$ [L/T] is the average flow velocity, $D$ [L$^2$/T] is the longitudinal dispersion coefficient, and $C$ is the concentration of the observed tracer above background levels [M/L$^3$]. The solution of the differential form of ADE for an instantaneous solute injection at x = 0 [L] reads:

$$C(t) = \frac{M}{A(4\pi Dt)^{1/2}} exp\left[-\frac{(L-vt)^2}{4Dt}\right] \qquad \text{Eq. 2}$$

Where $M$ is the injected solute mass [M], $t$ is time [T], and $L$ is the length of the investigated reach [L].

The TSM describes the solute transport in streams by combining the advection-dispersion process in the stream channel through a hydrologic exchange with an external storage zone. The model equations read (Bencala and Walters, 1983):

$$\begin{cases} \frac{\partial C}{\partial t} = -v\frac{\partial C}{\partial x} + \frac{1}{A}\frac{\partial}{\partial x}\left(AD\frac{\partial C}{\partial x}\right) + \alpha(C_{TS} - C) \\ \frac{\partial C_{TS}}{\partial t} = -\alpha\frac{A}{A_{TS}}(C_{TS} - C) \end{cases} \qquad \text{Eq.3}$$

where the hydrologic exchange with the transient-storage zone is driven by the exchange coefficient $\alpha$ [1/T] and the area of the transient storage zone, $A_{TS}$ [L$^2$]. Here, we will refer to $A$, $v$, and $D$ as "advection-dispersion parameters" and to $A_{TS}$ and $\alpha$ as "transient storage parameters". The solute concentrations in the main channel and the transient storage zone are $C$ and $C_S$ [M/L$^3$], respectively. The performances of both ADE and TSM results are evaluated using the Root Mean Squared Error objective function (*RMSE*). *RMSE* is an equivalent form of Residual Sum of Squares (*RSS*) and Mean Absolute Error (*MAE*) objective functions that are used in OTIS-P (the most frequently adopted inverse modelling approach for TSM, Runkel, 1998) and by the dynamic identifiability analysis (Wagener et al., 2002). *RMSE* allowed us a comparison of our TSM results with OTIS-P and with dynamic identifiability analysis consistently to previous studies (Wlostowski et al., 2013; Ward et al., 2017).

 **2.3 Random sampling and global identifiability analysis**

Several sampling approaches were previously used to estimate parameter identifiability in TSMs, such as Monte Carlo sampling (Wagner and Harvey, 1997; Wagener et al., 2002; Ward et al., 2013), Latin hypercube sampling (LHS, Ward et al., 2018; Kelleher et al., 2019), and Monte Carlo coupled with a behavioural threshold (Kelleher et al., 2013; Ward et al., 2017). Here, we use LHS to sample from the selected parameter range, due to LHS's higher efficiency compared to the classic Monte Carlo approach (Yin et al., 2011). A single combination of model parameters ($A$, $v$, and $D$ for ADE and $A$, $v$, $D$, $A_{TS}$, and $\alpha$ for TSM) obtained from the random sampling approach is herein referred to as "parameter set".

To obtain reliable TSM results, Ward et al. (2017) suggested a minimum amount of parameter sets between 10,000 and 100,000. Thus, in each TSM iteration we simulated 115,000 parameter sets. Results of each TSM iteration include $RMSE$ values for the 115,000 parameter sets, and results of identifiability analysis of the model parameters. The identifiability analysis includes parameter vs $RMSE$ plots (Wagener et al., 2003), parameter distribution plots (Ward et al., 2017), regional sensitivity analysis (Wagener and Kollat, 2007; Kelleher et al., 2019), and parameter distribution plots (Wagener et al., 2002; Ward et al., 2017). Since the above-mentioned identifiability analysis refers to model performance ($RMSE$) evaluated on the entire BTC, we refer to it as "global identifiability analysis." Globally identifiable parameters satisfy the following criteria: a univocal peak of performance in parameter vs $RMSE$ plots and in parameter distribution plots (Ward et al., 2017) and cumulative distribution function (CDF) corresponding to the best 0.1% of the results deviating from the 1:1 line and from parameter CDF corresponding to the best 10% of the results (Kelleher et al., 2019). We selected these behavioural thresholds (top 0.1% and top 10%) to assure consistency with previous work (Wagener et al., 2002; Wlostowski, 2013; Ward 2013; Ward 2017; Kelleher 2019). Parameter identifiability is usually evaluated via visual inspection of the plots from the global identifiability analysis (Wagener et al., 2002; Wlostowski et al., 2013; Ward et al., 2017; Ward et al. 2018; Kelleher et al., 2019). To couple visual inspection with a numerical measure able to express the degree of identifiability of a certain parameter, we evaluated the two-sample Kolmogorov-Smirnov (K-S) test that calculates the maximum distance $K$ and the corresponding $p$-value between two cumulative distribution functions, $F(P_{0.1})$ and $F(P_{10})$, by:

$$[K, p] = max|F(P_{0.1}) - F(P_{10})| \qquad\qquad \text{Eq. 4}$$

Where $F(P_{0.1})$ and $F(P_{10})$ are the cumulative distribution function of a parameter $P$ respectively for the best 0.1% and the best 10% of the results. Following the approach of Ouyang et al. (2014), we grouped parameter identifiability in four categories: highly identifiable ($K > 0.25$, $p \leq 0.05$), moderately identifiable ($0.1 \leq K \leq 0.25$, $p \leq 0.05$), poorly identifiable ($K < 0.1$, $p \leq 0.05$), and non-identifiable ($p > 0.05$).

**2.4 Identifiability analysis on specific sections of the BTC**

100 best-performing parameter sets for each iteration were analysed with the DYNamic Identifiability Analysis (DYNIA, Wagener et al., 2002) to address the role of model parameters on different sections of the BTC. Compared to the global identifiability analysis, the dynamic identifiability analysis evaluates the identifiability of a parameter on a moving window along the BTC. Following the approach of Wagener et al. (2002), we used a window size of three-time steps (~1 min for E1 and E2, and ~15 secs for E3). The dynamic identifiability analysis identifies regions of the observed data that are identifiable (or not) to the investigated model parameter, and it can

be used to test model structure, to design specific experiments, and to relate the model parameters to a specific simulated model response (Wagener et al., 2004). The dynamic identifiability analysis yields the distribution of the likelihood as a function of the parameter values and the information content of the parameters over time. The information content is expressed as one minus the width of the 90% confidence interval over the entire parameter range (Wagener et al., 2002). A wide 90% confidence interval indicates that various parameter values are associated to equally good performances resulting in low information content. Conversely, narrow 90% confidence intervals and corresponding high information content values suggest that the best-performing parameters are contained in a relatively narrow range compared to the feasible range. To evaluate the degree of identifiability of a certain parameter on specific sections of the BTC, we grouped parameter identifiability in three categories: highly identifiable (information content $\geq 0.66$), moderately identifiable ($0.33 \leq$ information content $< 0.66$), and poorly identifiable (information content $< 0.33$). We also specified sections of the BTC as follows: "peak" of the BTC is the section of the BTC corresponding to a neighbourhood interval of three time steps ($\pm \sim 1$ min for E1 and E2, and $\pm \sim 15$ secs for E3) around the maximum observed concentration; "rising limb" and the "tail" are respectively the BTC sections before and after the peak. A detailed description of how to read the plots used to address the global identifiability analysis and the description of the dynamic identifiability analysis algorithm are reported in Appendix A.

**2.5 Iterative approach to achieve model identifiability**

We simulated our tracer experiments with the ADE to avoid initial assumptions on advection-dispersion parameters that could affect the identifiability of transient storage parameters (Figure 1). The *RMSE* value of the best-performing ADE parameter set is referred to as *RMSE*$_{ADE}$. After obtaining identifiable advection-dispersion parameters, we simulated the observed BTC with the TSM by sampling advection-dispersion parameters from a parameter range defined based on the ADE results, while the transient storage parameters were based on literature values (Table 1). This first TSM simulation over 115,000 parameter sets is referred as to first TSM iteration.

Similar to the Monte Carlo approach coupled with behavioural thresholds (Kelleher et al., 2013; Ward et al., 2017) starting from the result of the first TSM iteration, we simulated the three tracer experiments through a step-wise approach with *n* TSM iterations (*n* is the number of iterations, Figure 1). The *n* TSM iterations sampled 115,000 parameter sets via LHS over parameter ranges defined by the results of the previous TSM iteration. Namely, if the global identifiability analysis from the previous TSM iteration indicated that the investigated parameter is identifiable, the best 10% of the results were used to define its parameter range in the successive TSM iteration (Figure 1). When the identifiability criteria were not met, the parameter range investigated in the successive TSM iteration was increased or, for the case of $A_{TS}$ and $\alpha$, it was reduced based on the dynamic identifiability analysis result (information content above 0.66 on the BTC tail). This condition was chosen by the evidence that transient storage parameters $A_{TS}$ and $\alpha$ are often non-identifiable via global identifiability analysis (Camacho and González, 2008; Ward et al., 2013; Ward et al., 2017; Kelleher et al., 2019), but are identifiable on the tail of the BTC (Wagener et al., 2002; Kelleher et al., 2013; Wlostowski et al., 2013).

While the first TSM iteration was conducted to investigate the identifiability of all the possible combinations in the feasible parameter range reported in the literature and from the results of ADE (Table 1), the successive iterations excluded pairs of *v* and *A* whose product was outside the value of the discharge evaluated via dilution gauging $\pm 10\%$. This condition was chosen to respect results from Schmadel et al. (2010), who reported that the

discharge error from the dilution gauging method is $\simeq 8\%$. The same approach (Figure 1) was used also in the case where $v$ was assumed fixed and equal to $v_{peak} = L/t_{peak}$, where $t_{peak}$ is the arrival time of the concentration peak. This choice was motivated by the fact that $v_{peak}$ is commonly adopted as a value for velocity in many transient storage studies (Ward et al., 2013; Kelleher et al., 2013; Wlostowski et al., 2017; Ward et al., 2017; Ward et al., 2018). The modelling was finalized once every model parameter indicated global identifiability via the enunciated criteria and the Kolmogorov-Smirnov test resulted in $K > 0.1$ and $p \leq 0.05$ for each model parameter.

**2.6 Number of parameter sets, parameter range, and identifiability of model parameters**

For each TSM iteration, we randomly extracted **N** parameter sets and their corresponding results. We then computed the mean and standard deviation of the top 10% of model results considering only the extracted subset of parameters **N** instead of the total 115,000. **N** increased from 1,000 to 115,000 with intervals of 1,000 parameter sets. We then evaluated the change in model performance with the changing number of sampled parameter sets for the different TSM iterations for the three experiments. A continuous decrease of the mean and the standard deviation of the top 10% results with increasing **N** shows that the number of chosen parameter sets clearly affect the performances of the random sampling approach for the investigated parameter range. On the contrary, constant mean and standard deviation of the top 10% results over increasing N point to the inability of the model and modelling procedure to increase the performances with an increasing number of parameter sets for that investigated parameter range (Pianosi et al., 2015).

**2.7 Comparison with an inverse modelling scheme and a Monte Carlo random sampling approach**

We compared our results with both inverse modelling results (OTIS-P) and the most-common random sampling approach for TSMs (OTIS-MCAT). OTIS-P is an inverse modelling scheme that minimises the residual sum of squares between the modelled and the observed BTC. OTIS-P model estimates the best-fitting model parameter values and their identifiability via the 95% confidence interval. We carried out multiple OTIS-P iterations starting from different initial parameter values to avoid a local minimum and interrupted the iterations when parameter values calibrated via OTIS-P changed less than 0.1% between subsequent runs (Runkel, 1998). OTIS-MCAT solves the TSM for the selected number of parameter sets and addresses their identifiability with a global identifiability analysis (Ward et al., 2017). Compared to our approach, OTIS-MCAT considers Monte Carlo parameter sampling instead of LHS, velocity equal to $v_{peak}$ and it does not foresee iterative parameter sampling from results of dynamic identifiability analysis. Thus, we here indicate as "OTIS-MCAT results" the results we obtained after the first TSM iteration when $v$ was assumed fixed and equal to $v_{peak}$.

**2.8 Metrics and hydrologic interpretation of TSM results**

The model parameter sets obtained from OTIS-P, OTIS-MCAT, and the proposed iterative TSM approach were used to compute some hydrologic metrics relate to solute transport in streams. Here we computed the average distance a molecule travels in the stream channel before entering the transient storage zone ($L_s$ [L], Mulholland et al., 1997):

$$L_s = \frac{v}{\alpha} \qquad\qquad\qquad\qquad\qquad\qquad\qquad\qquad\qquad\qquad \text{Eq.5}$$

The average time spent by a molecule in the transient storage zone ($T_{sto}$ [T]) is evaluated as (Thackston and Schnelle, 1970):

$$T_{sto} = \frac{A_{TS}}{\alpha A}$$  Eq.6

We computed the average water flux through the storage zone per unit length of the stream channel to interpret the magnitude of flux between the stream channel and the transient storage zone. Then we multiplied the obtained value by the reach length $L$ to obtain the total water flux through the storage zone for the entire stream reach ($q_s$ [L$^3$/T], modified from Harvey et al., 1996):

$$q_s = \alpha A L$$  Eq.7

However, the metrics $L_s$, $T_{sto}$, and $q_s$ do not encompass both the role of advective transport and of the transient storage. Thus, we also calculated $F_{MED}$ [-] that accounts for the median travel time due to advection-dispersion and transient storage and for the travel time only due to advection-dispersion (Runkel, 2002):

$$F_{MED} \cong \left(1 - e^{\left(-L\frac{\alpha}{v}\right)}\right) \frac{A_{TS}}{A_{TS}+A}$$  Eq.8

Increasing values of $F_{MED}$ have to be interpreted as increasing the relative importance of the storage zone in the
solute transport downstream (Runkel, 2002; Gooseff et al., 2013).

## 3. Results

### 3.1 ADE parameters

The global identifiability analysis showed a clear peak of performance toward univocal values for $v$, $A$, and $D$ for all three tracer experiments (E1, E2, E3, cfr. paragraph 2.1, plots not shown). The model performances varied
between $RMSE_{ADE}$ equal to 0.9894 mg/l (E3, Q = 22.79 l/s) and $RMSE_{ADE}$ equal to 1.9423 mg/l (E1, Q = 2.52 l/s).

### 3.2 TSM parameters

#### 3.2.1 Identifiability of model parameters when velocity is considered as a calibration parameter

After the first TSM iteration, the global identifiability analysis indicated that $v$, $D$, and $\alpha$ parameters are identifiable with a unique performance peak ($K$ of K-S test always > 0.22 and p < 0.05 for each tracer experiment). However,
$A$ and $A_{TS}$ appeared non- or poorly identifiable for the three investigated BTCs (Figure 2, green dots, $p$-value of the K-S test for $A_{TS}$ > 0.05 for each tracer experiment).

The global identifiability of model parameters increased with increasing iterations. In the TSM iterations where $A_{TS}$ or $\alpha$ were poorly or non-identifiable ($p$-value of the K-S test for $A_{TS}$ > 0.05), TSM performances approached at best $RMSE_{ADE}$ (Figure 2, green, yellow and blue dots). After four (for E1 and E2) or five (for E3) TSM iterations,
the parameter values plotted against the corresponding $RMSE$ values showed a univocal increase in performance toward unique values for $v$, $A$, $D$, $\alpha$, and $A_{TS}$ (Figure 2, orange dots), and the $RMSE$ of the best-performing parameter sets decreased below $RMSE_{ADE}$ (Figure 2, black horizontal line). Also, the CDF corresponding to the best 0.1% of the results deviated both from the 1:1 line and from the parameter CDF corresponding to the best 10% of the results (results not shown). These conditions, coupled with the $K$ of K-S test always larger than 0.1
(average $K$ for all the model parameters equal to 0.36, and $p$-value < 0.05) indicated parameter identifiability and the finalization of the iterative TSM approach.

**3.2.2 Identifiability of model parameters when velocity is set equal to $v_{peak}$**

The global identifiability of model parameters increased considerably through the iterative model approach also when velocity was not considered a calibration parameter. After the third TSM iteration, the best-performing parameter sets approached unique parameter values (Figure 3, blue dots) and the CDF corresponding to the best 0.1% of the results deviated from 1:1 line and from the CDF of the best 10% of the results (results not shown). These conditions, together with $K$ of K-S test always > 0.25 and $p$-value < 0.05 for each model parameter and tracer experiment, showed a clear increase in identifiability compared to the results after the first iteration (Figure 3, green dots). The increase in parameter identifiability was followed by a sharp increase in model performance, with the best-performing parameter sets at the end of the iterative approach having $RMSE$ values below $RMSE_{ADE}$ for all the investigated BTCs (Figure 3, blue dots and black line).

**3.3 Dynamic identifiability analysis**

**3.3.1 Dynamic identifiability analysis when velocity is considered as a calibration parameter**

The dynamic identifiability analysis provided clearer insights into the identifiability of the model parameters for different sections of the BTC compared to the global identifiability analysis (plots shown only for E1). After the first TSM iteration, $v$ and $\alpha$ proved to be the most identifiable and informative parameters on the rising limb, the peak, and the tail of the BTC (information content > 0.66; Figure 4a, b, g, h). $A$ and $D$ were mostly identifiable and informative during the rising limb and the tail of the BTC (Figure 4c-f). $A_{TS}$ was non-identifiable and poorly informative in most sections of the BTC (information content < 0.33; Figure 4i, j). However, the identifiability of $A_{TS}$ increased on the tail of the BTC, where the information content was above 0.66 for $A_{TS}$ between 0.77 m$^2$ and 5.35 m$^2$ (Figure 4i, j). Results from E2 and E3 showed that $\alpha$ and $A_{TS}$ were highly identifiable (information content > 0.66) for smaller sections of the tail of the BTC when the experiments were conducted at higher discharge stages (information content of $A_{TS}$ > 0.66 for 51% of the tail of the BTC for E1, for 23% for E2, and for 19% for E3, results not shown).

The dynamic identifiability analysis for the last TSM iteration showed that the advection-dispersion parameters were important in controlling the rising limb and the tail of the BTC (Figure 3k-p), while $\alpha$ was particularly important for controlling the tail (Figure 3q, r) and $A_{TS}$ for controlling the rising limb and the tail of the BTC (Figure 3s, t). Dynamic identifiability analysis after the last TSM iteration for E2 and E3 showed comparable results (not shown).

**3.3.2 Dynamic identifiability analysis when velocity is set equal to $v_{peak}$**

After the first TSM iteration, the dynamic identifiability analysis indicated that $A$ was poorly identifiable on the entire BTC (results reported only for E1, Figure 5a, b), while $D$ was moderately identifiable (information content between 0.66 and 0.33) on the rising limb and on the tail of the BTC (Figure 5c, d). $A_{TS}$ displayed high information content on the entire BTC (Figure 5g, h), with a narrow confidence interval on the tail of the BTC for values between 0.0014 m$^2$ and 0.43 m$^2$. $\alpha$ was non-identifiable on the majority of the BTC (Figure 5e), however, it showed high information content for values between 7.06$^{-05}$ 1/s and 0.0074 1/s at the tail of the BTC (Figure 5f). The dynamic identifiability analysis for the BTC of E2 and E3 yielded similar results, with narrow confidence

intervals for both $A_{TS}$ and $\alpha$ on the tail of the BTC and no clear trend between information content and discharge (results not shown).

The dynamic identifiability analysis for the last TSM iteration of E1 indicated that $A$ and $D$ control the tail and the rising limb of the BTC (Figure 5i-l). $\alpha$ acted both the rising limb and the tail of the BTC (Figure 5m-n) and $A_{TS}$ controlled mostly the tail of the BTC (Figure 5o, p). For E2 and E3, results after the last TSM iteration showed lower information content of $A_{TS}$ on the tail of BTC for increasing discharge stages compared to E1, while the information content of $\alpha$ was above 0.33 on the entire BTC (results not shown).

**3.4 Role of the used parameter range and the number of parameter sets for the identifiability of model parameters**

When a rather wide parameter range was used (first TSM iteration, green dots Figure 2), the performance of the global identifiability analysis was strongly dependent on the chosen number of sampled parameter sets. This can be derived from the strong decrease of the mean and the standard deviation of the top model results with the
number of sampled parameter sets **N** (results reported only for E1, Figure 6a). Also, for less than 97,000 parameter sets, the error between model performance using **N** parameter sets and using 115,000 parameter sets was always above 5% (vertical black lines, Figure 6a).

Our results showed that TSM results were poorly dependent by the sampled number of parameter sets when the model performance was studied for narrow parameter range around the peak of performance (last TSM iteration,
orange dots Figure 2). This was derived by the rather constant mean and standard deviation of the top model results with the number of subset **N**. Also, for a number of parameter sets **N** above 11,000 the error between model performance using **N** parameter sets and using 115,000 parameter sets was always below 2% (vertical black line, Figure 6b).

**3.5 Comparison with OTIS-P and OTIS-MCAT results**

Compared to results from our identifiability analysis, outcomes of OTIS-P were consistent with the best parameter sets obtained at the end of the iterative modelling approach (Table 2). Results from OTIS-P showed parameter identifiability with a narrow 95% confidence range for the $A_{TS}$ and $A$, while $D$ and $\alpha$ parameters were estimated with lower identifiability due to a larger 95% confidence range (Figure 2, 3). The parameter sets obtained via OTIS-P (Figure 2, 3, red vertical dashed line) were approaching the best fitting results obtained at the end of the
used iterative approach, regardless of whether flow velocity was considered as a calibration parameter (Figure 2) or was considered equal to $v_{peak}$ (Figure 3, Table 2).

The results of OTIS-MCAT showed low $p$-values for each model parameter after the K-S test ($p < 0.05$, $K > 0.12$) indicating parameter identifiability. However, compared to our results at the end of the iterative modelling approach, the global identifiability analysis of the OTIS-MCAT showed that the distribution of model parameters
did not converge towards univocal and optimal parameter values suggesting that model parameters were rather non-identifiable with the TSM performing less than the ADE (Figure 3, green dots).

**3.6 Variation of transport metrics with increasing identifiability of model parameters**

The evaluated transport metrics showed high uncertainty as long the model parameters were poorly or non-identifiable (Figure 2, 3, green and yellow dots). This was particularly evident after the first and second TSM

iterations, when the 100 best-performing parameter sets showed $T_{sto}$ values spanning over nine orders of magnitude (Figure 7d-f), while both $L_s$ and $q_s$ spanned over three orders of magnitude (Figure 7a-c, g-i). When the model parameters were poorly identifiable, the values of the transport metrics showed clear differences between simulations that were obtained with streamflow velocity as a calibration parameter (Figure 7, blue boxplots, first TSM iteration) and between simulations with streamflow velocity set equal to $v_{peak}$ (OTIS-MCAT,

Figure 7, orange boxplots, first TSM iteration). When $v$ was considered as a calibration parameter, the best-performing parameter sets after the first TSM iteration showed a non-negligible role of transient storage in solute transport for the investigated tracer experiments. This was indicated by the values of $L_s$ (from ~2 km for E1 to ~69 m for E3), by the simulated exchange flux $q_s$ (from 0.06 l/s for E1 to 8.8 l/s for E3), and by the solute residence time in the storage zone $T_{sto}$ (ranging from ~ 140 days for E1 to ~ 15 hrs for E3). Clearly different values for the

transport metrics were obtained when $v$ was set equal to $v_{peak}$. In this case, the results after the first TSM iteration showed a non-negligible exchange flux of the active stream with the transient storage zone ($q_s$ ranged from ~23 l/s for E1 to ~121 l/s for E3), a rather similar $L_s$ for the three tracer experiments (~10 m), and that $T_{sto}$ decreased between the experiments with increasing discharge (from ~12 sec for E1 to ~3 sec for E3).

        However, when the model parameters were identifiable, the transport metrics converged toward constrained

values and were consistent with OTIS-P results (Figure 7). This was achieved with a calibrated and a fixed (as in the OTIS-MCAT model) streamflow velocity. Results of the last TSM iteration showed that the investigated transport metrics have low dispersion around the median and that the median almost coincides with the result of the best-performing parameter set (Figure 7, red dots). When all model parameters were identifiable for each of the three tracer experiments, the transport metrics showed increasing $q_s$ (from ~2.7 l/s for E1 to ~23 l/s for E3),

increasing $L_S$ (from ~50 m for E1 to ~100 m for E3), and decreasing $T_{sto}$ (from ~150 s for E1 to ~33 s for E3) with increasing mean discharge of the experiments (from E1 to E3). $F_{med}$ did not change widely between the TSM iterations since the median of the best-performing 100 parameter sets varied always between 0.04 and 0.2 (Figure 7j-l). However, together with $q_s$, $L_S$, and $T_{sto}$ transport metrics , the dispersion of $F_{med}$ values around the median decreased with increasing identifiability of model parameters.

**4. Discussion**

        **4.1 The role of velocity in random sampling approaches for TSM**

        Our results showed that $v$ interacts with $\alpha$ and $A_{TS}$ in transient storage models. This was particularly evident when $v$ was considered as a calibration parameter, and the non-identifiability of $A_{TS}$ was coupled with identifiable $v$ and $\alpha$ (Figure 2, green and yellow dots). On the contrary, $A_{TS}$ was found to be identifiable and $\alpha$ to be non-identifiable

when $v$ was fixed equal to $v_{peak}$ (Figure 3, yellow dots). It is known that a separate evaluation of the advection-dispersion parameters from the transient storage parameters can result in misestimation of transient storage parameters due to the high parameter interaction (Knapp and Kelleher, 2020). Several studies addressed the identifiability of model parameters, yet, no study so far investigated the role of the flow velocity on the identifiability of $\alpha$ or $A_{TS}$, and studies rely on a flow velocity equal to $v_{peak}$ in random sampling approaches for

TSMs (Ward et al., 2013; Kelleher et al., 2013; Wlostowski et al., 2017; Ward et al., 2017; Ward et al., 2018). The practice of setting $v$ equal to $v_{peak}$ in past studies was justified by the notion that $v_{peak}$ can be considered as a reasonable good approximation for the advection process in the stream channel (Ward et al., 2013; Wlostowski et al., 2017) and by the modelling advantage that assuming $v$ equals $v_{peak}$ would reduce model dimensionality (Knapp

and Kelleher, 2020). While reducing the number of model parameters is advantageous for reduced model
dimensionality, considering $v$ as a calibration parameter is a needed testing strategy in TSMs. This is because
measurement uncertainty is inevitable in determining discharge or flow velocity, thus we don't know how big the
effect of measurement uncertainty is on model performance, especially considering parameter interaction. Also,
constraining the advection-dispersion parameters $A$ and $D$ already proved to affect the identifiability of the other
model parameters (Lees et al., 2000; Kelleher et al., 2013; Ward et al., 2017), but no study assessed the role of
velocity on parameter identifiability.

Our results provide valuable guidance for future studies addressing parameter identifiability in TSM. Specifically,
our results support the current praxis of considering velocity fixed and equal to $v_{peak}$, especially when research
aims at evaluating the distribution of "behavioural" parameter sets in TSMs (i.e. parameter sets satisfying certain
performance thresholds). This is due to the fact that using velocity as calibration parameter leads to the same
parameter identifiability compared to the case when velocity is considered fixed (Figure 2, 3, Table 2). Yet, setting
velocity equal to $v_{peak}$, requires a considerably lower amount of computational power due to the lower degrees of
freedom of the TSM. However, when research aims to evaluate the control of the model parameters on the shape
of the BTC, our results suggest that increasing the model complexity by considering velocity as a varying model
parameter can offer more detailed insights into the role of advection-dispersion processes on the tail of the BTC
and of the transient storage parameters on the rising limb and peak of the BTC (Figure 4, 5). Indeed, our results
highlighted how assuming $v$ equals $v_{peak}$ led to a stronger influence of $\alpha$ and weaker influence of $A_{TS}$ on the BTC
compared to the case when $v$ is considered as a calibration parameter. Also, our dynamic identifiability analysis
underestimated the role of $A$ and $A_{TS}$ on the rising limb and peak of the BTC and overestimated the role of $D$ and
$\alpha$ on the rising limb of the BTC for the case $v$ equals $v_{peak}$ compared to the case when $v$ was a calibration parameter
(Figures 4, 5).

The assumption used in previous work of streamflow velocity equalling $v_{peak}$ implies that $v_{peak}$ should encompass
the effect of advection on the entire BTC or at least in the rising limb and peak of the BTC (Ward et al., 2013;
Kelleher et al., 2013; Wlostowski et al., 2017; Ward 2018). However, when $v$ was used as a calibration parameter,
our results showed that $v$ is one of the least meaningful parameters for simulating the peak of the BTC at low
discharge (Figure 4k, i), while higher information content for $v$ is obtained at higher discharge rates for values
larger than $v_{peak}$ at the peak of the BTC (dynamic identifiability plots not shown).

**4.2 Control of model parameters on the rising limb, the peak, and the tail of the BTC**

The results of our dynamic identifiability analysis showed that both the advection-dispersion and the transient
storage parameters control solute arrival-time and solute retention in stream channels. This outcome is in
contradiction with the common interpretation of model parameters, where it is assumed that the advection-
dispersion parameters control the solute arrival time, while transient-storage parameters are assumed to control
the tail of the BTC (Bencala, 1983; Bencala and Walters, 1983; Runkel, 2002; Smith, 2005; Bencala et al., 2011).
Following this common interpretation of the role of model parameters on the BTC, some authors decomposed the
BTC into an advective part and a transient storage part (Wlostowski et al., 2017; Ward et al., 2019). This
decomposition allowed them to quantify the role of advection-dispersion and transient storage embedded in the
BTC. However, this modelling strategy also implicitly assumes a negligible role of advection-dispersion

parameters on the tail of the BTC and of transient-storage parameters on the rising limb and peak of the BTC, which is in not consistent with our findings (Figures 4, 5, 8).

Several studies addressed how different model parameters affect the shape of the BTC and showed partly similar but also contrasting outcomes to our findings (Figure 8g-l, Wagner and Harvey, 1997; Wagener et al., 2002; Scott et al., 2003; Wlostowski et al., 2013; Kelleher et al., 2013). Past studies found that the rising limb of the BTC was controlled by the stream channel area $A$ alone (Wagener et al., 2002), by the combination of $A$ and the longitudinal dispersion coefficient $D$ (Wagner and Harvey, 1997; Wlostowski et al., 2013; Kelleher et al., 2013), or by $A$, $D$, and $A_{TS}$ (Scott et al., 2003). The peak of the BTC was found to be controlled by advection-dispersion parameters in most past TSM applications (Wagener et al., 2002; Wlostowski et al., 2013; Scott et al., 2003; Kelleher et al., 2013). However, Wagner and Harvey (1997) reported a non-negligible role of the transient storage parameters $\alpha$ and $A_{TS}$ in controlling the arrival time of the peak concentration (Figure 8g). Eventually, while the majority of the studies found the transient storage parameters $\alpha$ and $A_{TS}$ to control the tail of the BTC (Wagner and Harvey, 1997; Scott et al., 2003; Wlostowski et al., 2013), results reported by Wagener et al., (2002) and by Kelleher et al. (2013) highlight the role of the stream channel area $A$ on controlling a large portion of the tail of the BTC.

The observed identifiability of model parameters in different sections of the BTC in past work and the differences compared to our findings (Figure 8a, c, e) might be driven by different physical settings or discharge conditions of the study sites, by the methods used to account for parameters identifiability, by the parameter sampling procedure, or by the strategy used to obtain the best-fitting parameter sets (Wagner and Harvey; 1997; Scott et al., 2003; Kelleher et al., 2013). For example, the identifiability of the TSM to $\alpha$ and $A_{TS}$ is expected to increase for dispersive streams and alluvial stream channels, compared to mountain reaches with low or null hydrologic exchange with the hyporheic zone (Kelleher et al., 2013). However, our analysis also suggests that the different results on the importance of model parameters for certain sections of the BTC (Figure 8) could be driven by the selected random sampling approach and the non-identifiability of model parameters.

Plots of the parameter values against the corresponding objective function in Wagener et al. (2002) and the regional sensitivity analysis in Wlostowski et al. (2013) do not indicate parameter identifiability for $A_{TS}$, $D$ and $\alpha$. These results together with our identifiability plots when model parameters were poorly identifiable (Figures 2, 3, green and yellow plots) suggest that the range and the number of the parameter sets chosen in different studies could have been insufficient to obtain global sensitivity and identifiability of $D$, $A_{TS}$, and $\alpha$ parameters. Similar to results by Wagener et al. (2002) and Wlostowski et al. (2013), our dynamic identifiability analysis showed no influence of $A_{TS}$ on the majority of the BTC, when $A_{TS}$ was non-identifiable (Figure 4i, j).

Compared to our results, the different role of the model parameters on controlling the shape of the BTC in previous studies (e.g. Kelleher et al., 2013) could be driven by the different approach used for evaluating the sensitivity (i.e. Sobol' sensitivity analysis). However, our results suggest that the number of parameter sets (42,000) selected by Kelleher et al. (2013) might not have been sufficient to obtain identifiability of the model parameters with the rather wide parameter range chosen for their Monte Carlo sampling (Table 1). Results by Kelleher et al., (2013) are very similar to our TSM iterations for cases where $\alpha$ was non-identifiable ($v$ equals to $v_{peak}$, Figure 3 yellow dots, dynamic identifiability plots not shown). We also demonstrated that our results after the first and second TSM iterations are not sufficient for interpreting the transient storage process, because of the non-identifiability of the model parameters and the low model performances ($RMSE \geq RMSE_{ADE}$ (Figure 3a-l, green and yellow dots).

This study offers significant insights in understanding which model parameter influence the shape of the BTC, suggesting that only behavioural parameter sets should be considered in models aiming to understand the control of model parameters on the rising limb, peak, and tail of the BTC. Future work should address the interaction of model parameters on controlling different sections of the BTC for more complex model formulations (e.g. TSM with two or several transient storage zones, Choi et al., 2002; Bottacin-Busolin et al., 2011).

**4.3 On the importance of parameter range, parameter sets, and challenges associated to parameter identifiability in TSM**

The applied iterative approach was effective in drastically improving parameter identifiability with the increase of TSM iterations. Identifiability of parameters in TSMs is commonly studied via random sampling approaches using between 800 and 100,000 parameter sets sampled from a parameter range spanning several orders of magnitude (Table 1). Despite a large number of parameter sets used in previous studies, model parameters were found identifiable only in a few studies (Ward et al., 2017, 2018), while at least one model parameter was found to be non-identifiable in the majority of current TSM studies. Many authors found identifiable $A_{TS}$ coupled with non-identifiable $\alpha$ (Camacho and González, 2008; Kelleher et al., 2013; Wagener et al., 2002; Wlostowski et al., 2013), while other TSM applications found $\alpha$ to be identifiable coupled with non-identifiability for $A_{TS}$ (Kelleher et al., 2019), or $\alpha$ and $A_{TS}$ to be both non-identifiable (Camacho and González, 2008; Ward et al., 2013; Ward et al., 2017). Our results offer a possible explanation for the observed non-identifiability of model parameters in published work. Our study demonstrated that it is unlikely to reach parameter identifiability via a random sampling approach using less than 100,000 parameter sets when a rather wide range of model parameters is used (Table 1, Figure 6a). While the range and the order of magnitude of advection-dispersion parameters can be estimated by using the ADE, the ranges where $\alpha$ and $A_{TS}$ are identifiable are not known a-priori and random sampling approaches need to target a parameter range wide enough to capture the distribution of transient storage parameters on their entire feasible range (Ward et al., 2013; Kelleher et al., 2013; Ward et al., 2017). We here proved that investigating the most identifiable parameter range is more effective for achieving parameter identifiability than just using a large number of parameter sets on a wide parameter range (Figure 6). The peak of performance for the transient storage parameters can be so narrow that it can be missed by the random sampling approach or by only a low number of selections when the sampled parameter range spans many orders of magnitude. Similar conclusions have been obtained by Ward et al. (2017), who found by using the OTIS-MCAT model via 100,000 parameter sets that the model parameters were identifiable only for one of the three investigated BTCs. Other studies coupled random sampling approaches with behavioural thresholds to reduce parameter non-identifiability, yet this was done to constrain only the range of $A$ (Kelleher et al., 2013; Ward et al., 2017). Here, we demonstrated the importance of the parameter range over the number of parameter sets in random sampling approaches for TSMs (Figure 6). The adopted identifiability analysis was effective in finding behavioural parameter sets after a few iterations regardless of the modelling approach used (OTIS-MCAT as well as considering $v$ as a calibration parameter). Of particular interest is our finding that high information content (> 0.66, e.g. Figure 4j. 5f) of $\alpha$ and $A_{TS}$ on the tail of the BTC after the dynamic identifiability analysis can be used to reduce the parameter range in successive TSM iterations. This result is in agreement with recent findings of Rathore et al (2021), who found the tail of the BTC to contain fundamental information for transient storage processes and the parameters describing it.

The adopted iterative approach allowed to achieve parameter identifiability and to obtain physically realistic transport metrics. However, this approach is based on the specific objective function used (*RMSE*) and on the subjective thresholds to control the refinement of the parameter range for successive iterations (top 10% results for the global identifiability analysis, and information content > 0.66 for the dynamic identifiability analysis). Future work should explore the impact of the selection of the thresholds and of different objective functions on

the physical realism of the modelling results and of the identifiability of the parameters.

     The applied iterative approach is foremost a tool for achieving parameter identifiability by investigating the entire range of feasible parameter values via existing identifiability tools (global identifiability analysis and dynamic identifiability analysis). The larger amount of time and computational power required by the adopted identifiability analysis compared to the rather straightforward application of OTIS-P paid off in terms of

completeness of results and granted a more comprehensive view of the possible modelling outcomes on the feasible parameter range. Also, compared to the standard random sampling approach, the identifiability analysis used in the present work proved effective in iteratively constraining the parameter range to reduce the dimensionality of the model, eventually providing both identifiable model parameters and optimal parameter sets with model performances approaching (or even outperforming) calibrated results via inverse modelling (Table 2).

Our simulations with OTIS-P resulted in excellent model performances for the investigated BTCs, with low *RMSE* values and with calibrated model parameters comparable to the behavioural parameter populations obtained via our global identifiability analysis (Figure 2, 3). While the obtained performances of the OTIS-P calibration are certainly specific to the investigated BTCs, the use of OTIS-P alone would have not provided enough information to address the reliability of the obtained model parameters. This, in turn, would have raised concerns about the

credibility of the transport metrics obtained, eventually compromising the robustness of the derived physical process involved at the study site. Compared to random sampling approaches coupled with global identifiability analysis, inverse modelling approaches are often considered not as meaningful for interpreting modelling outcomes (Ward et al., 2013; Knapp and Kelleher, 2020). This is because parameters calibrated via inverse modelling might be non-identifiable despite an overall good model performance (Kelleher et a., 2019) and because

identifiability analysis informs on behavioural parameter set which is a preferable and more informative outcome for hydrological models than a single set of parameter values (Beven, 2001; Wagener et al., 2002). Thus, our identifiability analysis over different investigated parameter ranges can offer an explanation about why in past studies identifiability analysis over a probably too large parameter range indicated non-identifiability and lack of convergence with OTIS-P results (Ward et al., 2017).

Eventually, even if random sampling approaches are generally considered more informative than the inverse-modelling approach (Ward et al., 2013; Ward et al., 2017; Ward et al., 2018; Knapp and Kelleher, 2020), our results indicate that random sampling outcomes that show non-identifiability of transient storage parameters should not be used for process interpretation in TSM. This was evident from TSM iterations showing non-identifiability of $\alpha$ and $A_{TS}$, with the best model performances approaching the $RMSE_{ADE}$ (Figure 2, 3, black line)

indicating an underestimation of the transient storage process with the optimal modelled BTCs mimicking the ADE.

**4.4 Implications of identifiable model parameters for hydrologic interpretation of modelling results**

Our results demonstrated that poor or non-identifiability of model parameters can result in a wrong hydrological interpretation of the processes controlling solute transport in streams. Additionally, our results showed that with increasing discharge conditions $L_s$ and $q_s$ increased, $T_{sto}$ decreased, and $F_{med}$ was rather stable for simulations where the model parameters were identifiable (cfr. paragraph 3.2). The low uncertainty and the values of the investigated transport metrics suggested that the transient storage at the experimental site was most probably controlled by in-stream dead zones (Boano et al., 2014; Smettem et al., 2017). Our modelling outcomes are also in line with the physical understanding of the studied stream reach. The study site is equipped with a dense network of groundwater monitoring wells that showed that the stream channel is almost entirely in gaining conditions for the investigated tracer injections with the groundwater gradients pointing toward the stream channel (Bonanno et al., 2021). This is in line with the obtained TSM transport metrics that indicate a very limited or even a lack of hyporheic exchange. Other modelling and experimental studies also outlined that the stream above the study section is dominated by inflow of groundwater or surface water from wetlands (Antonelli et al. 2020; Glaser et al., 2016, 2020). The observed link of $L_s$, $q_s$, and $T_{sto}$ values with discharge (Figure 7) also suggested that the transient storage at our site became less important in controlling solute transport with increasing discharge. The decrease of $A_{TS}$ and $T_{sto}$ with increasing discharge has been argued to indicate an increase of groundwater gradients toward the stream channel with a consequent decrease in the hyporheic zone at different study sites (Morrice et al., 1997; Fabian et al., 2011). However, the observed groundwater gradients at the study site exclude the presence of significant hyporheic exchange during the three simulated tracer experiments. The observed trend between modelling results with discharge might be interpreted by the fact that, as the discharge increases, the wetted profile at the study site incorporates into the advective part of the channel the dead zones and the low-flow areas that are responsible for in-stream transient storage at lower flow rates (Zarnetske et al., 2007; Gooseff et a., 2008). This would cause a progressive increase in piston-flow transport and a reduced role of in-stream solute retention with increasing flow and water level in the stream channel.

However, if we would have based the process interpretation on simulations before we reached identifiability of model parameters, the conclusions would have been different. The values for the transport metrics obtained when $v$ and the other model parameters were considered as a calibration parameter, together with published results on solute residence time in the hyporheic zone and in the stream channel (Gooseff et al., 2005; Boano et al., 2014) could have been interpreted in a way that the transient storage was controlled by in-stream dead zones during high-discharge events and by a low rate hyporheic exchange at low flow conditions (Figure 7, blue boxplots and first TSM iteration). Conversely, results from first TSM simulation when $v$ was considered fixed and equal to $v_{peak}$ might have been interpreted in a way that transient storage of the studied stream channel was controlled by dead zones at the lowest flow conditions and by in-stream turbulences that caused solute retention in the transient storage zone to last ~3 seconds during high-flow events (Figure 7, orange boxplots and first TSM iteration).

Our results also open developments for research seeking to increase the physical realism of the TSM and its results. Increased model complexity is both associated with a better analytical fitting to the observed BTC, but also with an increased degree of freedom of the model with a consequent reduction of parameter identifiability (Knapp and Kelleher, 2020). Our approach offers a promising flexible tool to target parameter identifiability and physical interpretation also in TSM formulation with increasing complexity, such as multiple storage zone models (Choi et al., 2002), or for TSMs considering sorption kinetics (Gooseff et al., 2005) or different residence time

distribution laws such as log-normal distribution (Wörman et al., 2002), exponential plus pumping distribution (Bottacin-Busolin et al., 2011), and power law distribution (Haggerty et al., 2002).

**5 Conclusion**

There is a clear need in stream hydrology to better identify parameters for simulating solute transport in streams. Here we addressed the challenge of parameter identifiability in TSMs by combining global identifiability analysis with dynamic identifiability analysis in an iterative modelling approach. Our results showed that the value of stream velocity interacts with the transient storage parameters. Namely, when stream velocity was a randomly sampled calibration parameter (within a physical reasonable range), we found non-identifiable $A_{TS}$ and identifiable

$\alpha$. On the contrary, when stream velocity was assumed to be equal to $v_{peak}$, $A_{TS}$ was found identifiable and $\alpha$ non-identifiable. We proved that such a non-identifiability of transient storage parameters can result in the modelled BTC approaching the ADE. Our work demonstrates that both transient storage and advection dispersion parameters control the shape of the BTC, when these model parameters are identifiable. This is contrary to previous studies that reported that advection-dispersion parameters control the rising limb and the peak of the

BTC and that the transient storage parameters control the tail of the BTC. We also showed that non-identifiable model parameters could severely misestimate the solute retention time in the transient storage zone ($T_{sto}$) and the exchange flux between the stream channel with the transient storage zone ($q_s$). The differences of $T_{sto}$ and $q_s$ between identifiable and non-identifiable parameters were up to four and two orders of magnitude, respectively. The modelling approach in this study constrained the parameter range iteratively. This strategy successfully

reduced model dimensionality and allowed us to obtain identifiable model parameters for the three tracer experiments. As a complement to the existing body of literature, our work shows that the non-identifiability of model parameters in past studies might be related to the rather small number of sampled parameter sets compared to the investigated parameter range. The low uncertainty of the model parameters and the derived transport metrics were pivotal for obtaining a robust assessment of the hydrological processes driving the solute transport at the

study site. On the contrary, using non-identifiable model parameters, or relying on OTIS-P results alone, would have led to uncertain and rather different process interpretation at the study site.

Our study provides enhanced understanding on the relevance of identifiable parameters of TSM models. We also provide insights how parameter calibration without an assessment of their identifiability likely results an unrealistic conceptualization of processes and unrealistic values for different solute transport metrics.

**6 Acknowledgements**

This work was supported by the funding from the Luxembourg National Research Fund (FNR) for doctoral training (PRIDE15/10623093/HYDRO-CSI). We would like to acknowledge the financial support of the Austrian Science Fund (FWF) as part of the Vienna Doctoral Programme on Water Resource Systems (DK W1219-N28). We thank Ginevra Fabiani, Adnan Moussa, Laurent Pfister, Rémy Schoppach for their fruitful input and

discussions.

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

**Table 1.** Parameter names, abbreviations, and units together with a summary of publications that address identifiability of model parameters with random sampling approaches. We reported the used number of parameter sets and the parameter ranges, while in parenthesis it is reported the method used for the parameter sampling. "Double step" indicates that the sampling procedure was divided into two steps. In the first step, $A$ varied across a broad range and in the second step, it was varied across a narrower range to cover the most sensitive range of the parameter domain. Each of the two steps investigated a number of parameter sets equal to half of the total number indicated in the table.

| Parameters | Units | Symbol | |
|---|---|---|---|
| Streamflow velocity | [m/s] | $v$ | |
| Stream channel area | [m²] | $A$ | |
| Longitudinal dispersion coefficient | [m²/s] | $D$ | |
| Stream-storage zone exchange rate | [1/s] | $\alpha$ | |
| Transient storage area | [m²] | $A_{TS}$ | |
| **Authors** | **Number of parameter sets** | **Range of TSM parameters** | |
| Wagner and Harvey, 1997 | 800 (Monte Carlo) | $A$ | 0.02-0.6 |
| | | $D$ | 0.025-0.8 |
| | | $A_{TS}$ | 0.01-2 |
| | | $\alpha$ | 0.000005-0.001 |
| Wagener et al., 2002 | 1,000 (Monte Carlo) | $A$ | 0.3-1.05 |
| | | $D$ | 0.1-0.225 |
| | | $A_{TS}$ | 0.1-0.5 |
| | | $\alpha$ | 0.00035-0.0025 |
| Wlostowski et al., 2013 | 2,000 (Monte Carlo) | $A$ | 0.5-1.0 |
| | | $D$ | 0.5-1.5 |
| | | $A_{TS}$ | 0.05-0.5 |
| | | $\alpha$ | 10-4-10-3 |
| Kelleher et al., 2013 | 42,000 (Double step Monte Carlo) | $A$ | 0.001-1.0 (in the second step, limits chosen via the top 1,000 results of first step) |
| | | $D$ | 0.001-1.0 |
| | | $A_{TS}$ | 0.001-0.01 |
| | | $\alpha$ | 10-5-10-3 |
| Ward et al., 2013 | 100,000 (Monte Carlo) | $A$ | +-50% $A_{peak}$ |
| | | $D$ | 0.0001-5 |
| | | $A_{TS}$ | 0.01-10 |
| | | $\alpha$ | 10-8-10-1 |
| Ward et al., 2017 | 100,000 (Double step Monte Carlo) | $A$ | 0.1-1 (0.3-0.5 in the second step) |
| | | $D$ | 0.01-10 |
| | | $A_{TS}$ | 0.01-1 |
| | | $\alpha$ | 10-5-10-1 |
| Kelleher et al., 2019 | 27,000 (LHS) | $A$ | 1.0 - 3.0 |
| | | $D$ | 0.001 - 10 |
| | | $A_{TS}$ | 0.01 - 1 |
| | | $\alpha$ | 10-6 - 10-2 |
| This manuscript | Second step ADE – 35,000 (LHS) | $v$ | $v_{peak} \cdot 0.8$ - velocity of the first increase of concentration |
| | | $A$ | +-20% $A_{peak}$ |
| | | $D$ | 0.0001 - $D_{best} \cdot 1.2$ |
| This manuscript | First TSM iteration – 115,000 (LHS) | $v$ | +-50% $v_{ADE}$ |
| | | $A$ | +-50% $A_{ADE}$ |
| | | $D$ | 0.0001 - $D_{ADE} \cdot 2$ |
| | | $A_{TS}$ | 0.00001 - 20 |
| | | $\alpha$ | 0.00001 - 0.1 |

**Table 2: Summary of the TSM results. OTIS-MCAT results refer to the case $v = v_{peak}$ without any successive modification of the parameter via dynamic identifiability analysis results. "Iterative TSM" indicate the best parameter sets obtained after the iterative TSM approach presented in Figure 1 and applied for the cases $v$ considered as a calibrated parameter ($v = calib.$) and when it was considered fixed and equal to $v_{peak}$ ($v = v_{peak}$). The best TSM results are indicated in bold font.**

| | | | $v$ [m/s] | $A$ [m$^2$] | $D$ [m$^2$/s] | $\alpha$ [1/s] | $A_{TS}$ [m$^2$] | $RMSE$ |
|---|---|---|---|---|---|---|---|---|
| E1 | *ADE* | | 0.0681 | 0.0395 | 0.0965 | / | / | 1.9423 |
| | *OTIS-P* | | 0.0739 | 0.0364 | 0.0637 | 0.0006 | 0.0074 | **0.6159** |
| | *OTIS-MCAT* | | 0.0739 | 0.0351 | 0.1339 | 0.0119 | 0.0051 | 2.7421 |
| | Iterative TSM | $v = calib.$ | 0.0728 | 0.0369 | 0.0522 | 0.0013 | 0.0073 | 0.7229 |
| | | $v = v_{peak}$ | 0.0739 | 0.0359 | 0.0534 | 0.0013 | 0.0077 | 0.7681 |
| E2 | *ADE* | | 0.1746 | 0.054 | 0.1599 | / | / | 0.9982 |
| | *OTIS-P* | | 0.1774 | 0.0509 | 0.1151 | 0.0016 | 0.0077 | 0.4152 |
| | *OTIS-MCAT* | | 0.1774 | 0.0604 | 0.1271 | 0.0137 | 0.0033 | 1.4429 |
| | Iterative TSM | $v = calib.$ | 0.1790 | 0.0523 | 0.1131 | 0.0018 | 0.0067 | **0.3377** |
| | | $v = v_{peak}$ | 0.1774 | 0.0528 | 0.1154 | 0.0015 | 0.0065 | 0.3696 |
| E3 | *ADE* | | 0.262 | 0.0874 | 0.2525 | / | / | 0.9894 |
| | *OTIS-P* | | 0.275 | 0.081 | 0.1404 | 0.005 | 0.0144 | **0.2544** |
| | *OTIS-MCAT* | | 0.275 | 0.0849 | 0.2441 | 0.0259 | 0.0073 | 1.2612 |
| | Iterative TSM | $v = calib.$ | 0.2861 | 0.0818 | 0.1286 | 0.0064 | 0.0145 | 0.2697 |
| | | $v = v_{peak}$ | 0.275 | 0.083 | 0.1603 | 0.0037 | 0.0123 | 0.3109 |






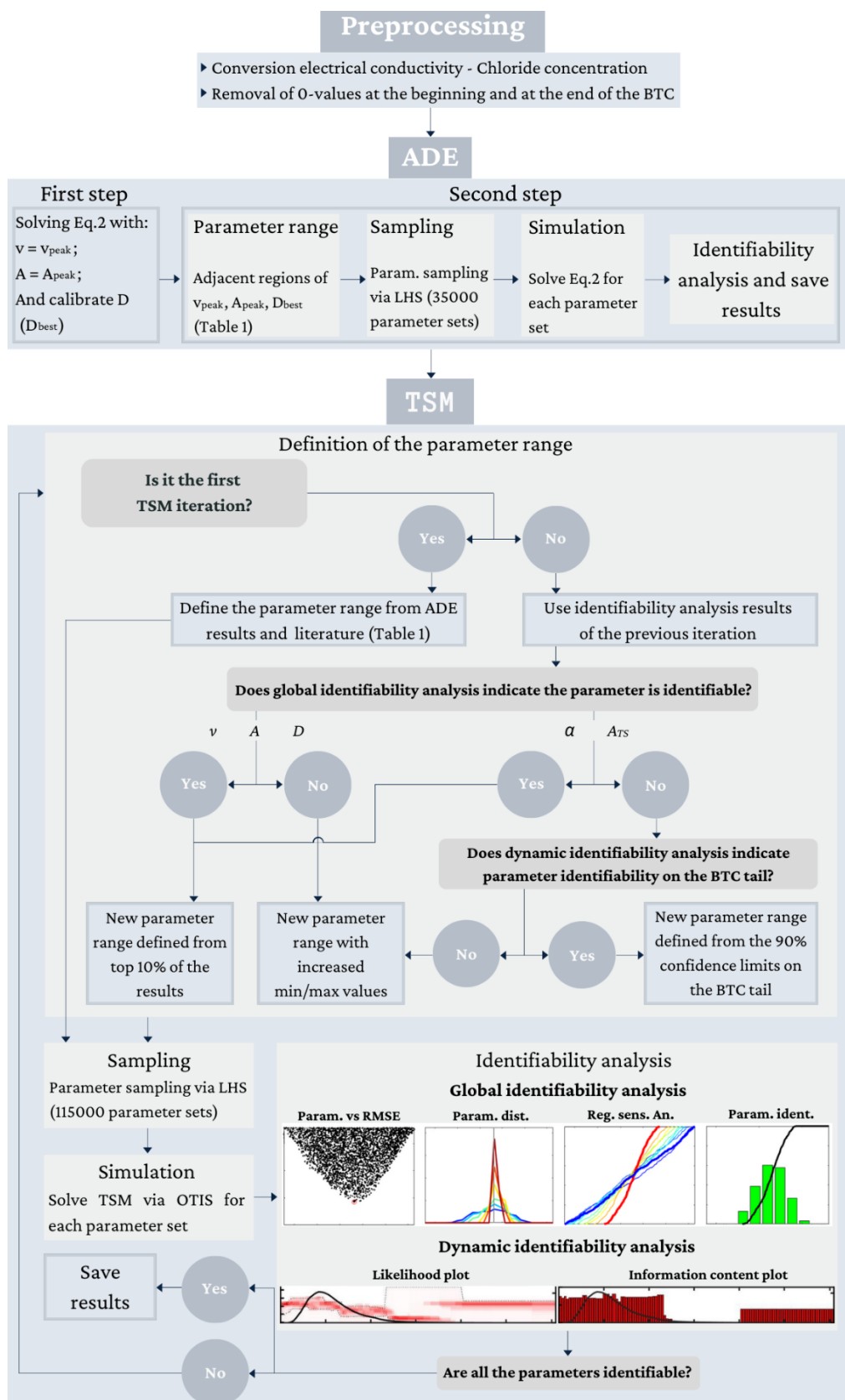

**Figure 1: Conceptual modelling workflow. The parameters have the following unit of measurements: velocity $v$ [m/s],**

**cross-sectional area $A$ [m²], longitudinal dispersion coefficient $D$ [m²/s], exchange coefficient $\alpha$ [1/s], area of the transient**

**storage zone $A_{TS}$ [m²].**

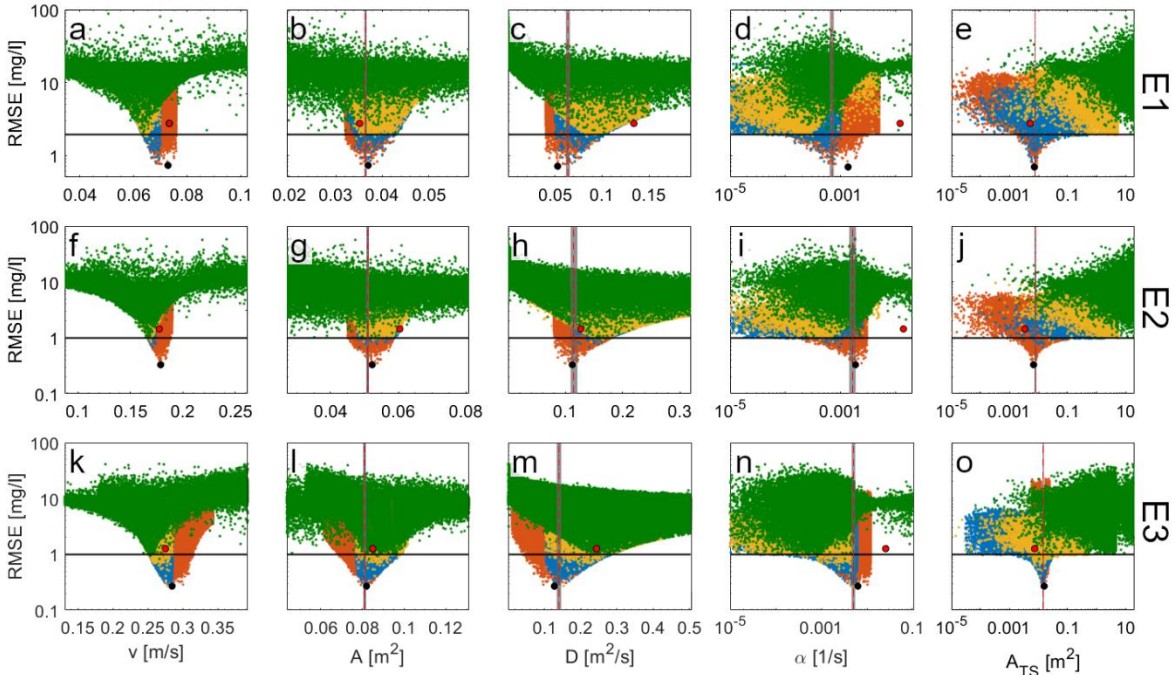

**Figure 2. Parameter values plotted against the corresponding *RMSE* values for the TSM results conducted for the tracer injections (a-e) E1, (f-j) E2, and (k-o) E3. (a-j) Green, yellow, blue and orange dots indicate results respectively for the first, second, third, and fourth TSM iterations. (k-o) Green dots indicate results for the first and second TSM iterations, while yellow, blue and orange dots indicate results respectively for the, third, fourth, and fifth TSM iterations. Each TSM iteration was conducted via 115,000 parameter sets. The red dots indicate OTIS-MCAT results (best parameter set after the first TSM iteration for *v* equals $v_{peak}$) while the black dots indicate the best-performing parameter value after the used iterative TSM approach. The horizontal black line indicates the $RMSE_{ADE}$ (Table 2). Vertical dashed red line indicates OTIS-P results, while the 95% confidence range for OTIS-P results are indicated via vertical grey areas.**

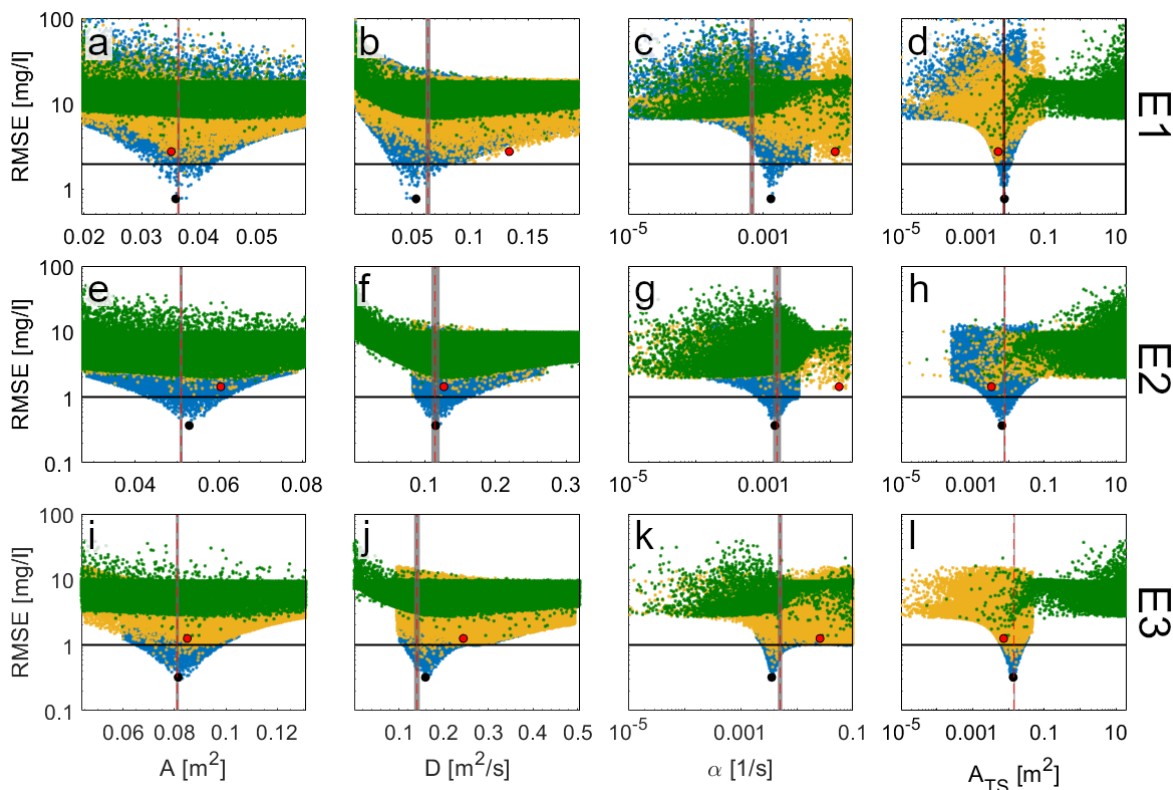

**Figure 3. Same as Figure 2, but reporting TSM results when velocity was considered equal to $v_{peak}$.**





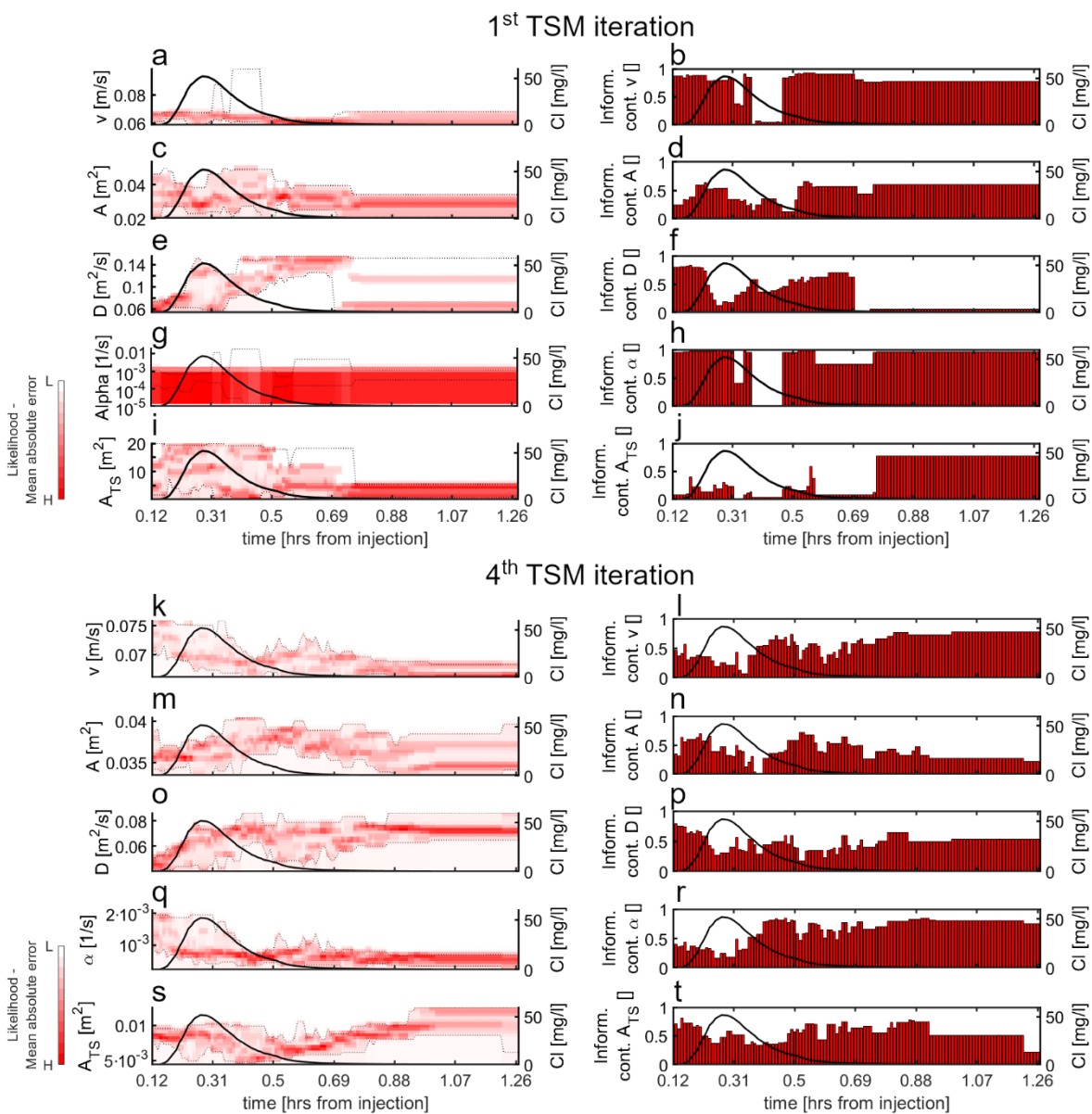

**Figure 4. Dynamic identifiability analysis of model parameters for E1 when *v* was considered as a varying model parameter. Results report for the (a-j) first TSM iteration and the (k-t) last TSM iteration. (a), (c), (e), (g), (i), (k), (m), (o), (q), (s) likelihood distribution as function of parameter values at each time step. Black lines indicate the observed BTC, and dashed black lines indicate the 90% confidence limits. (b), (d), (f), (h), (j), (l), (n), (p), (r), (t) indicate parameter information content (red bars) at each time step while the black lines indicate the observed BTC.**




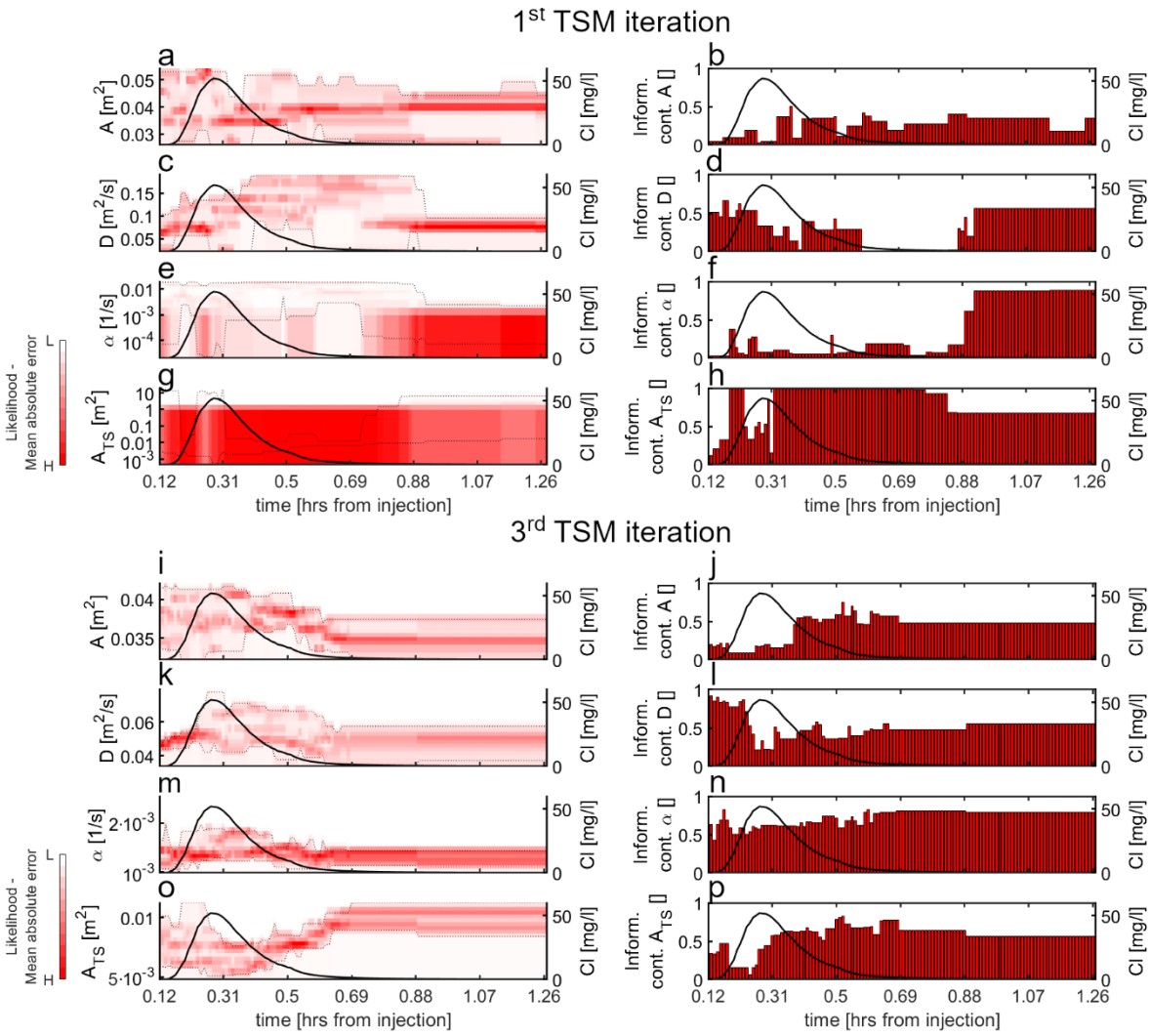

**Figure 5. Same as Figure 4, but reporting dynamic identifiability results for E1 when $v$ was set equal to $v_{peak}$.**

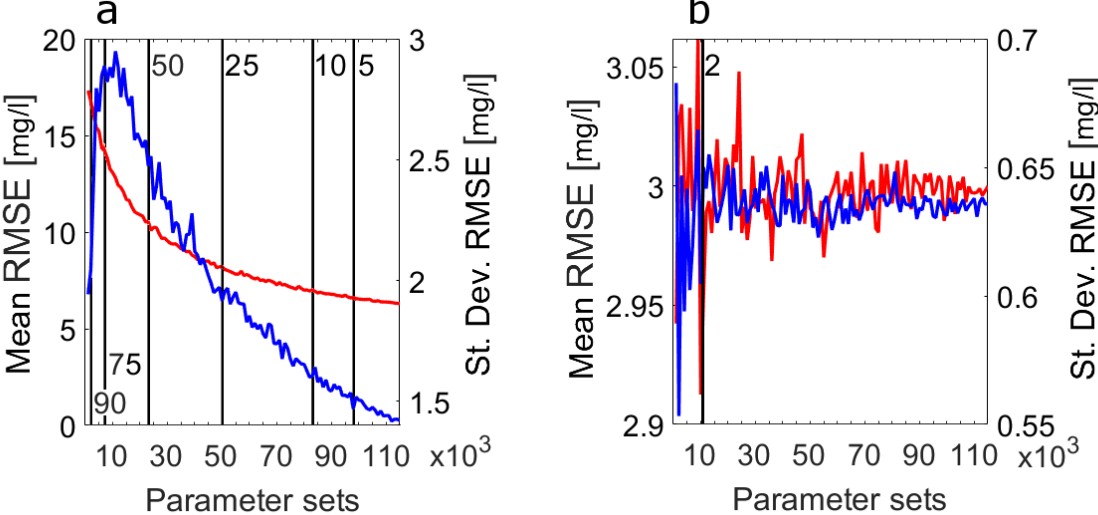

**Figure 6: Mean (red lines, left axes) and standard deviation (blue lines, right axes) for *RMSE* values relative to the top 10% of the modelling results as a function of the number of parameter sets used in the TSM. The results are reported for the (a) first TSM iteration and the (b) last TSM iteration (E1). Vertical black lines indicate the number of parameter sets needed to have the shown percentage difference between the mean *RMSE* value calculated at the indicated number of parameter sets and at 115,000 parameter sets. Eg: In plot (a) only using at least 50,000 parameter sets there is less than 25% difference in the top 10% *RMSE* values compared to results using 115,000 parameter sets.**

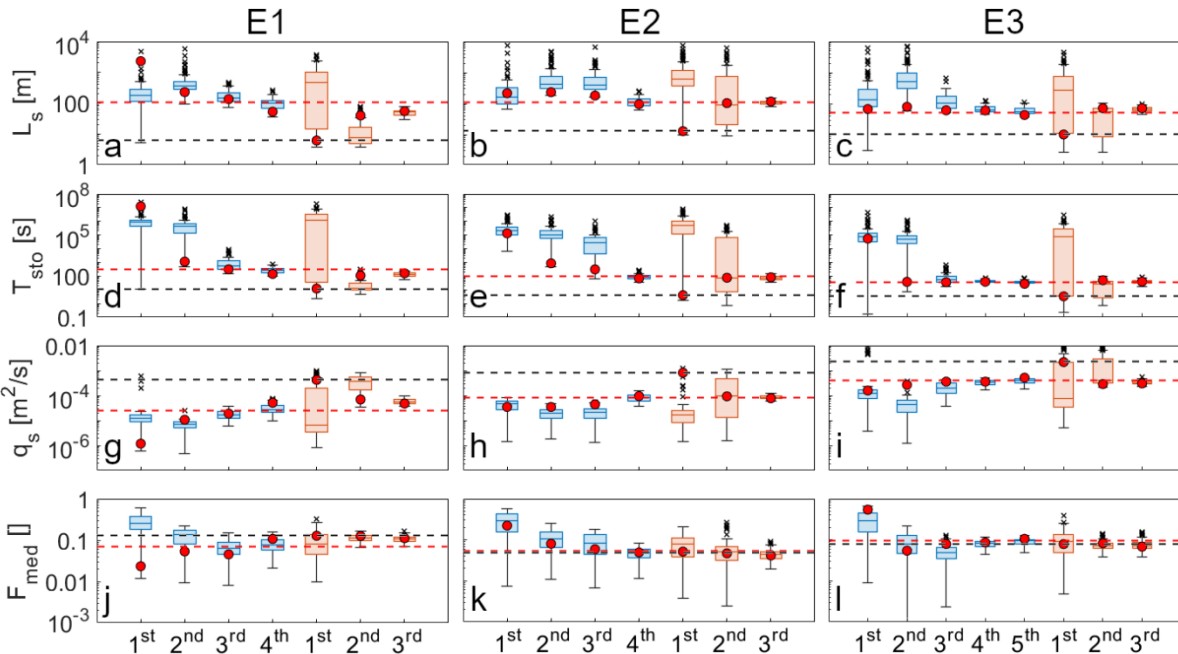

**Figure 7. Boxplots of the investigated transport metrics for the best 100 parameter sets for the three simulated experiments. (a-c) $L_s$, (d-f) $T_{sto}$, (g,i) $q_s$, (j-l) $F_{med}$. Results are reported for (a, d, g, j) E1, (b, e, h, k) E2, and (c, f, i, l) E3. On the x axis, we indicated the *n-th* TSM iteration. Blue and orange boxplots indicate results when velocity was a varying model parameter and when it was kept fixed and equal to $v_{peak}$, respectively. Red dots indicate the transport metric values obtained via the parameter sets with lower *RMSE*. The red and the black horizontal dashed lines indicate respectively the transport metric obtained using the OTIS-P results and OTIS-MCAT results (first TSM simulation when velocity it was kept fixed and equal to $v_{peak}$).**

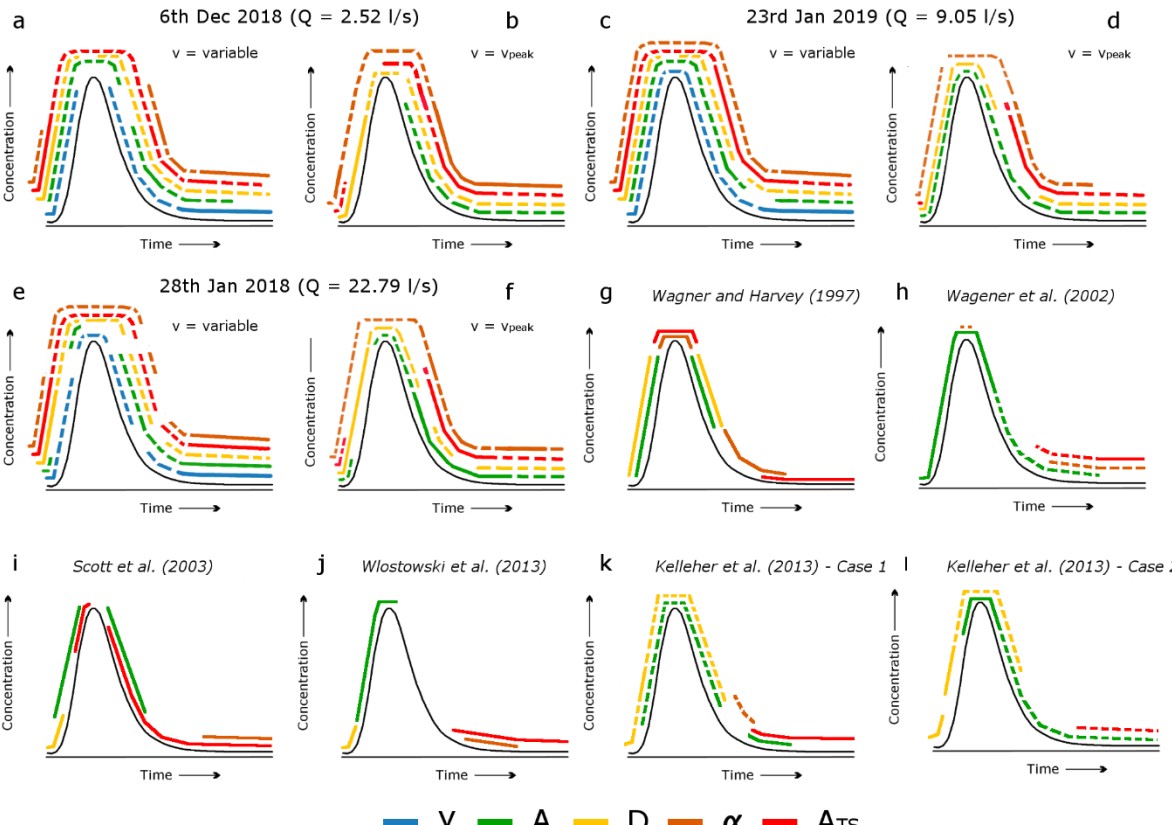

**Figure 8. Qualitative plots of the TSM parameter influence on different sections of the BTC. (a) and (b) qualitative parameter information content on the BTC for E1, (c, d) E2, and (e, f) E3. In plots (a-f) solid lines indicate an information content above 0.66 while dashed lines indicate an information content between 0.33 and 0.66. (g) Wagner and Harvey, 1997; parameter influence described via sensitivity evaluation (cfr. p. 1733, Wagner and Harvey, 1997), therefore the parameter influence is described using only solid lines. (h) Wagener et al., (2002); Plot (h) has been**

**modified from Figure 7 in Wagener et al., (2002) in order to fit our 0.66 and 0.33 threshold classification in term of information content. (i) Scott et al., (2003); parameter influence described via dimensionless sensitivity (cfr. Table 1 in Scott et al., 2002), therefore the parameter influence is described using only solid lines. (j) Wlostowski et al., 2013; Plot (j) describes the parameter influence after the dynamic identifiability analysis, however information content plots were not reported by the authors, therefore the solid lines indicate the areas for the best-performing parameters as indicated**

**in Figure 2 of Wlostowski et al. (2013). (k) Kelleher et al., (2013) for the case of a dispersive mountain stream (Case 1) and (l) Kelleher et al., (2013) for the case of a small low-flow mountain stream (Case 2); Plots (k) and (l) indicate by solid and dashed lines if the parameters influence the model output by itself or through interactions (cfr. Section 6.1 Kelleher et al., 2013).**


**Appendix A - Parameter sensitivity and identifiability**

The interpretation of the parameter range is based on the sensitivity and identifiability of the *i-th* parameter on the
chosen model (the TSM) via a selected objective function used to compare model results with the observation (the
BTC) (Kelleher et al., 2019; Wagener et al., 2003; Wagener and Kollat, 2007; Ward et al., 2017; Wlostowski et
al., 2013). A parameter is called sensitive whenever a variation in the parameter value causes variations in the
TSM performances (Kelleher et al., 2019). A parameter is identifiable whenever the best-fit value of that
parameter is constrained on a relative narrow range across the entire distribution of the possible parameter values
(Ward et al., 2017). To assess identifiability of parameters of TSM, we used parameter vs likelihood plots,
identifiability plots, regional sensitivity analysis plots and parameter distribution plots.

Parameter vs likelihood plots visualize the distribution of the investigated values of a certain parameter plotted
against the corresponding values of the objective function (Wagener et al., 2003; Wagener and Kollat, 2007).
Identifiable parameters are described in parameter vs likelihood plots by a univocal increase of model
performances approaching a certain optimum-value of the parameter (Figure A1a). Non-identifiable parameters are
described in parameter vs likelihood plots by a not-univocal increase of performances of the model in certain
parameter range (Figure A1b). Parameter distribution plots show probability density function (PDF) divided by
behavioural sets (from top 20% to top 0.1% of the results for the selected objective function) (Ward et al., 2017).
Identifiable parameters are indicated by narrow range of the PDF relative to the smaller behavioural sets (top
0.1%, 0.5% and 1% of the results) compared to a wider range of the PDF relative to the larger behavioural sets
(top 5%, 10% and 20% of the results) (Figure A1c). Non-identifiable parameters are defined by equally wide PDF
for the different investigated behavioural sets (Figure A1d). Regional sensitivity analysis plots are obtained after
dividing the population of the parameter by behavioural sets (from top 10% of the results to top 1% of the results
with 1% step for the selected objective function, Ward et al., 2017; Kelleher et al., 2019). Each objective function
population so obtained was transformed into cumulative distribution functions (CDFs) for equal size bins of the
parameter range (Kelleher et al., 2019; Wagener and Kollat, 2007). Sensitive parameters are identified by CDF
for the top 1% of the results deviating from the CDF for the top 10% of the results (Figure A1e). If the CDFs lay
on the 1:1 line, then the objective function is uniformly distributed across the parameter range which indicates
parameter unsensitivity (Figure A1f). Identifiability plots display the CDF of the objective function across the
selected parameter range (Wagener et al., 2002; Ward et al., 2017). The slope of the CDF will be higher in the
parameter interval where the model is more sensitive to that parameter. The measure of the local gradient of the
cumulative distribution will be represented by the height of the bar plot in each equally-sized bin across the
parameter range. Higher bars and steeper gradients of the CDF line indicate greater model performances in that
parameter range and, therefore, parameter sensitivity and identifiability (Figure A1g). On the contrary, equal eight
of the bars and similar gradients of the CDF line indicate that the parameter is unsensitive and non-identifiable
(Figure A1h).

The plots used to address the global sensitivity analysis indicate parameter identifiability and sensitivity on the
entire observed BTC, however they are unable to address if the *i-th* parameter describes the process it is meant to
represent or if the role of the *i-th* parameter on the model is constant in time (Wagener and Kollat, 2007). To
address identifiability and sensitivity of the *i-th* parameter on the different sections of the BTC we applied dynamic
identifiability analysis which steps are reported in Figure A2 (Wagener et al., 2002).

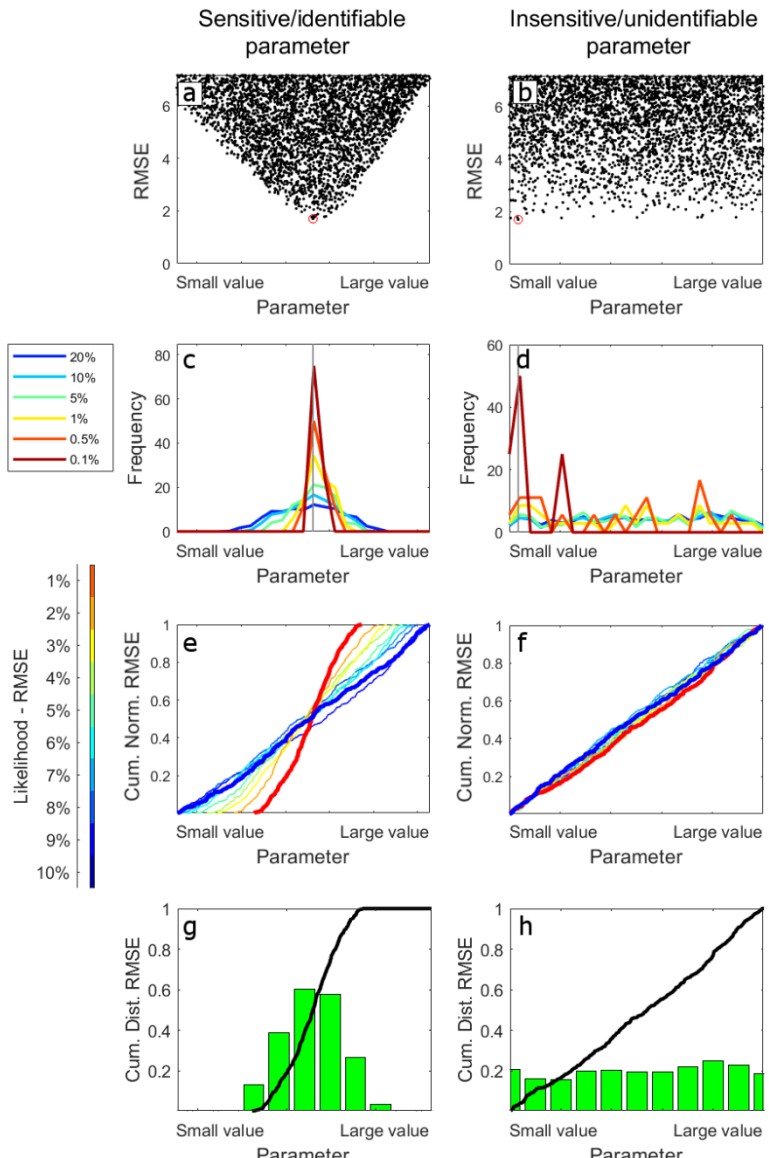

**Figure A1: Examples of the four types of visualizations intended for parameter identifiability and sensitivity with the plots in the first column (a, c, e, and g) reporting an example of plots for sensitive and identifiable parameter and plots in the second column (b, d, f, and h) reporting an example of plots for insensitive and non-identifiable parameter. (a) and (b) parameter vs likelihood plots; (c) and (d) parameter distribution plots for the top 20, 10, 5, 1, and 0.1% of the results; (e) and (f) regional sensitivity analysis plots from the top 1% to the top 10% of the results; (g) and (h) identifiability plots for the top 1% of the model results.**

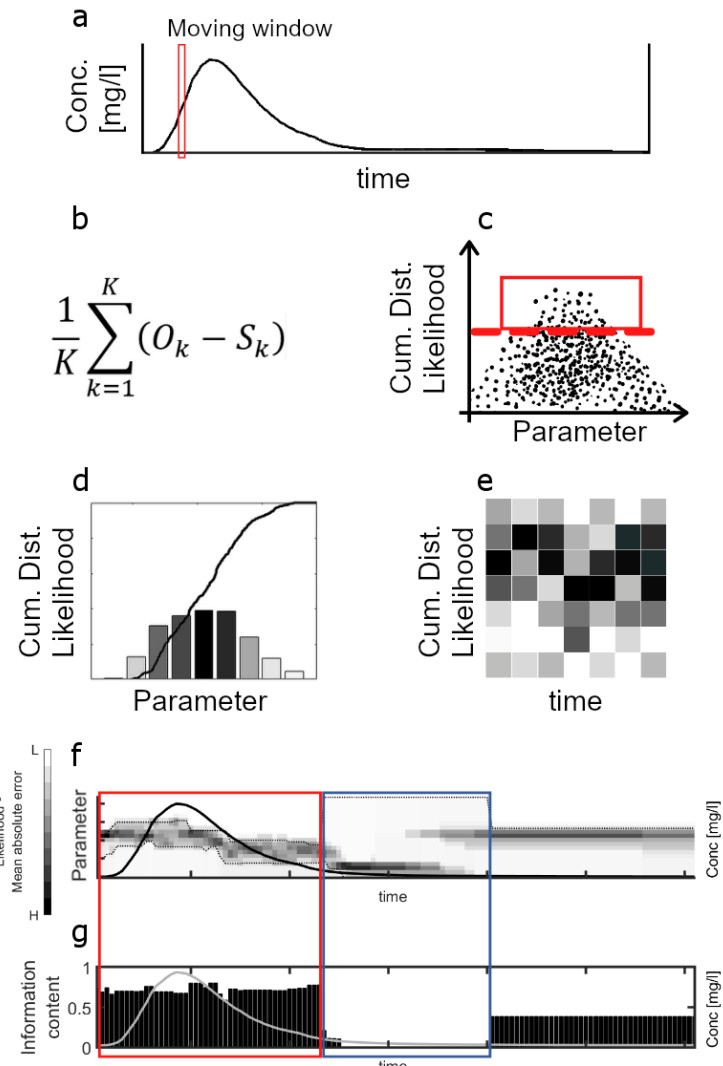

1110 **Figure A2. Dynamic identifiability analysis algorithm flowchart. (a) The BTC is subdivided in moving windows (size equal to three times the BTC timestep, Wagener et al., 2002); (b) In each moving window the likelihood (efficiency) of every TSM simulation is evaluated via mean absolute error (Wagener and Kollat, 2007); (c) an efficiency-threshold is chosen (e.g. top 10%); (d) for the chosen model results, the cumulative distribution function is built for each investigated parameter; (e) steps from (b) to (d) are repeated for each moving window and model likelihood for the**

1115 **investigated parameter is plotted over time (white: minimum likelihood; black: maximum likelihood). (f) cumulative distribution function of the parameter distribution is plot vs the observed BTC together with 90% confidence limits. Narrow limits indicate identifiable parameter while wide limits indicate unidentifiable parameter. (g) a second plot reports the metric of one minus the normalized distance between the 90% confidence limits. Small values of this metric indicate that the selected time window contain a narrow identifiability range for the investigated parameter and,**

1120 **therefore, that it is informative on that part of the BTC (Wagener et al., 2002).**

**Appendix B – Observed vs simulated BTCs**

The figure B shows the observed BTC for the three tracer experiments plotted against the top 100 simulated BTC obtained using the proposed iterative approach. The observed poor visual fit on the tail of the BTC obtained at the end of the iterative modelling approach (Figure B1d, e, f) is controlled by two factors: (i) the modelling structure of the TSM which assumes an exponential residence time distribution and (ii) the chosen objective function. By using alternative residence time distributions, TSM proved to have a more accurate fitting on the tail of the BTC (Haggerty et al., 2002; Bottacin-Busolin et al., 2011). Also, the RMSE could not be the best objective function for addressing a model fit on the tail of BTC because it gives higher importance on the sections of the BTC with higher concentration values (peak of the BTC) compared to the sections of the BTC with low concentration values (at the tail of the BTC). As an example, the best-fitting BTC obtained at the end of the second TSM iteration (E1) shows a visually better fit on the BTC tail (Figure B2) despite the large RMSE (1.5197 mg/l).

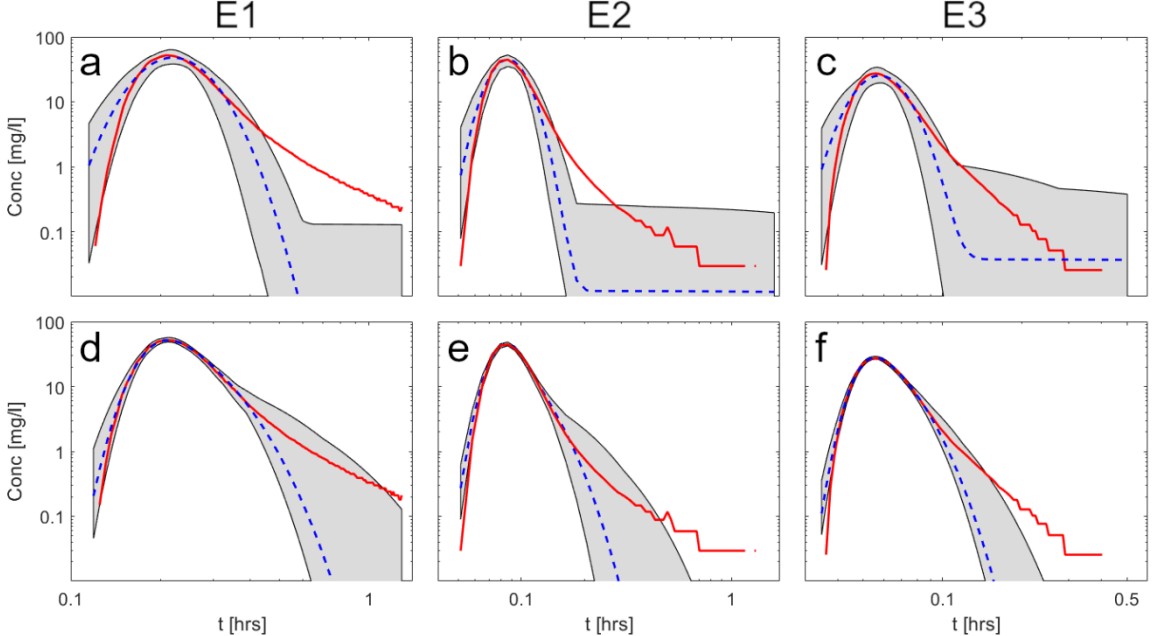

Figure B1: Observed BTC (red line) together with the grey area comprised between the top 100 simulated BTCs and the best-fitting BTC (blue dashed line) for (a, d) E1, (b, e) E2, and (c, f) E3. Results reported for the first (a, b, c) and last (d, e, f) TSM iterations.

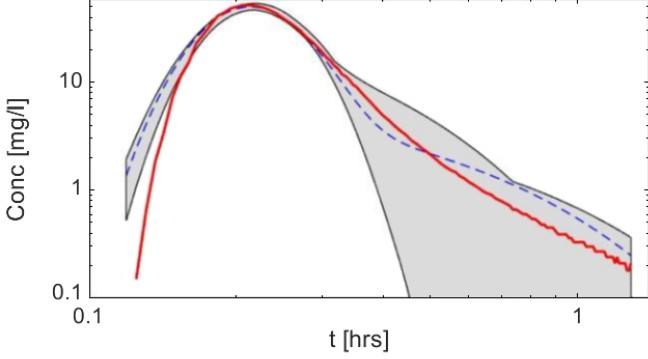

Figure B2: Observed BTC (red line) together with the grey area comprised between the top 100 simulated BTCs and the best-fitting BTC (blue dashed line) for the second TSM iteration (E1).