# Peer review of "Exploring tracer information in a small stream to improve parameter identifiability and enhance the process interpretation in transient storage models"

_Hydrology and Earth System Sciences, 2022_

## Referee Comment (RC2)

**General comments**

a) The text content is generally well written but the theme's complexity and the structuring makes honestly the reading rather difficult, which can force the reader going back and forth the chapters to follow a storyline. I found very logical how the text was structured in the 'Study site and Methods' section, and I would really try to stick to this same structure when presenting 'Results', in the 'Discussion' section and even in the 'Conclusion' paragraph. My personal taste would be:

    a.1)      ADE parameters
    a.2)      TSM parameters (i.e. hydrologic exchange parameters)
               a.2.1.  Identifiability of TSM parameters when $v = v_{variable}$
               a.2.2.  Identifiability of TSM parameters when $v = v_{peak}$
    a.3)      Model iterations
               a.3.1.  TSM first iterations
               a.3.2.  TSM last iterations
               a.3.3.  DYNIA
               a.3.4.  Comparison with inverse modeling results (OTIS-P)
               a.3.5.  Comparison with random sampling approaches (OTIS-MCAT)
    a.4)      Metrics and hydrologic interpretations of model results

According to this numbering, the 'Results' section starts with the a.3.1 jumps to a.3.3, then back to a.3.2, etc. The 'Discussion' section starts with a.4, then goes back to a.3.1 and a.3.2, follows with a.3.4, etc. This can make a difference between a quick and effective reading, and a tedious reading.

b) I would like to draw the attention about the use of the term 'parameters' in the manuscript. I personally have no experience with the identifiability of parameters in a TS model and it can bias my understanding of the term. Thus, this might be probably my wrong perception.
As a hydrologist, my understanding of **parameters** involved in the transport of solutes in a stream/river can be stream discharge, components of flow velocity, flow turbulence, grain size of streambed sediments, groundwater-stream (GW-SW) water exchange fluxes, stream channel topography, existence/absence of riparian vegetation, etc. In the manuscript, these parameters are treated as such, but then it introduces the term **TSM parameters** and **parameter sets**. In this manuscript's TSM model, each iteration simulates 115,000 parameter sets. An unexperienced reader would tend to think that > 100,000 parameters as the ones I mentioned above are involved in the transport of solutes in the stream. This should be obviously clarified.

c) The manuscript's title states 'to reduce the uncertainty' but I have missed more uncertainty analysis throughout the text.

**Specific comments**

**Abstract**

The abstract is concise and clearly written. How to fill the research gap is properly addressed.

1. Line 14: I would consider adding 'adjacent groundwater bodies' as a exchanging agent.
2. Line 24: … TSM parameters, **respectively**. The severe differences…
3. Line 26: … at **the** study site → The article 'the' makes reference to a determined object, but the study site has not yet been introduced. Consider changing 'the' for 'our' or 'a study site in western Luxembourg', for example.

**Introduction**

The introduction chapter needs to be revised in order to be fully understandable. The content is good, but an improved organization can elevate it.

4. Tend to list consecutive citations in chronological order (e.g. lines 72-73, 78-79, etc.).
5. Line 49: Many readers can disagree with such a definition of 'hyporheic zone'. I don't agree with the idea of a fully saturated thickness. I usually consider the definition given by Cardenas & Wilson, 2007 (Cardenas, M. B., & Wilson, J. L. (2007). Exchange across a sediment–water interface with ambient groundwater discharge. Journal of Hydrology, 346, 69– 80. ), as a good approach.
6. Lines 79-82: This reads like a Research gap, and it is well stated, but it should fit at a later stage, before enumerating hypothesis and research questions.
7. Lines 103 and 113: I would not change paragraph in this sentence.
8. Line 105: 'stream velocity' → The stream itself does not move. Better using 'streamflow velocity'.
9. Line 128: I would state the hypotheses in a new paragraph.

**Study site and methods**

This chapter is very well written according to me. Clear and concise. I would try to keep this structure in the following chapters.

10. Line 152: **Study site and 'data'** → Better specify which data.
11. **Study site and data** → How long is the studied stream reach?
12. Lines 161-162: How was Q calculated? (± Analytical Error?)
13. Line 164: How was EC calibrated according to NaCl concentrations? Was EC temperature compensated? (± Analytical Error?)
14. Was there any difference in background EC between the injection and the measurement locations? Or was it assumed to be equal? Or is it one of the testing parameters? These aspects are actually interesting.
15. Line 181 (Equation 3): Why the second term of the second equation is negative?
    If $C_s < C$, the concentration of a certain solute in stream tends to get diluted. But according to eq. 3, $\partial C/\partial t$ would become larger…
16. 2.2: How the accuracy of the ADE and TSM results are assessed?
17. Line 207: I assume CDF is the Cumulative Distribution Function, but it has not been defined as such yet.
18. 2.3: How is the uncertainty of the model iterations' results, DYNIA's results and the comparison with OTIS-P and OTIS-MCAT assessed?
19. Line 270 (Equation 7): Introduce a tab in the equation's label, so it can be in line with the rest of the equations.
20. 2.4: I assume that including FMED reduced the systematic error, but is it quantifiable?

**Results**

This chapter is properly written but the sub-chapters could be re-organized to build-up a smoother storyline. See the last part of General comment (a).

21. Lines 279-282: This reads more like in the 'Methods' section, right?
22. Why was $v_{peak}$ and no other velocity chosen for the fixed velocity scenario? Why not median velocity which could be more a representative velocity value?
    What does $v_{peak}$ mean?
    - Peak velocity during tracer experiments?
    - Mean velocity during the experiment E3 which had the largest discharge?
    - Peak velocity during the entire monitoring period?
23. Line 292: Figure 3c-f →Is Figure 3f informative in the tailing of the BTC?
24. Line 297: Add a comma after > 0.05**,**
25. Line 342: 8.8 l/s for E3 → I would not consider this discharge as a low $q_s$, since you state that the exchanging flow is > 1/3 of the total Q of the stream.
26. Line 343: 15 hrs for E3 → I would not consider this as a long residence time. The flux velocity is roughly 4.6 m/hr, which according to me can be slow for dead zones, but not for hyporheic flux, or GW-SW exchange.
27. Lines 345-348: are these simulations physically possible at all, knowing the actual stream discharge? As for $T_{sto}$, do 0.8 (in E1) and 3.3 (in E3) m/s make sense?
28. Lines 353-354: Again… double check the applicability of these results into the actual flow regime.
    Comments 26, 27 and 28: Since you are doing a comparison between methods, rather than using terms like 'high' or 'low', it is better to use relative terms such as 'higher' or 'lower', or even 'distinct'.
29. Line 349: 'whether' does not sounds appropriate in this context. I would rather use 'regardless'?

**Discussion**

Same as in the previous chapter, the sub-chapters could be re-organized to build-up a smoother storyline. See the last part of General comment (a). Aren't 4.1 and 4.3 more related to each other, and 4.2 a unique sub-chapter. I think starting with 4.2 and then following with 4.1 and 4.3 would improve the chapter's sequence.

30. 4.1: Are the uncertainties specified?
31. Lines 402 and 406-407: Aren't these contradictory statements? Identifiability is contradictory under the same scenario (in both cases $v = v_{peak}$).
32. Line 438: 100,000 instead of 100'000.
33. Line 446-447: Seems like a very generalist closing statement.
34. Lines 460-462: Aren't these contradictory statements? The assumption $v = v_{peak}$ might not be representative of the advection role, but can encompass the effect of advection in the entire BTC.
35. Lines 480-503: Figure 8 is actually a very good review from other authors' findings. I quite like it. The piece of text relating to this figure is actually an important part of your discussion, but again it reads like a succession of sentences with vague organization, and it forces the reader to continuously go back and forth between the text and the figure. I propose the following. Instead of travelling through all the cited authors and comparing your results with their results, why can't it be presented in concordance with the different BTC's features, e.g.:

- Rising Limb
- Peak
- Falling Limb
- Tail

One can travel along the BT curve and compare your data with literature data. The reading would be more graphical and intuitive, and it could probably help to safe some word spacing.

By the way, it is interesting to know whether there is a method or a simple threshold value to distinguish between the end of the falling limb and the beginning of the tail?

36. Lines 502-503: '… Kelleher et al., 2013 also indicate …" → whereas it has a strong influence based on your results, right?

37. Lines 503-504: 'Different sensitivity…' → This reads like the staring of the next paragraph.

**Conclusion**

The conclusion reads well, but again, I would try to expose the ideas trying to follow the same sequence of findings shown in the results.

38. Line 523: '… that the BTC…' -→ Revise wording.

**Tables and Figures**

39. Line 739 (Table 2): $v = v_{peak}$ → Try to keep the same notation as in the text (i.e. Italic and subscript).

40. Line 777 (Figure 2): The meaning of colored dots for (m-o) plots not specified.

41. Line 778 (Figure 2): Try to keep the same notation as in the text (i.e. Italic and subscript).

42. Figure 3: Here you show the BTC in E1 when the streamflow is smaller and the identifiability is probably less dynamic. I would have liked to see differences in dynamism between the different tracer experiments.
    This does not mean that you need to re-do the figures for other tracer experiments, but probably comment a bit on it in the text.

43. Figure 4: I would probably combine Figs. 3 and 4 in the same figure. Both are showing exactly the same info (Experiment E1 and $v = v_{variable}$) but only for different iteration states, and can be potentially confusing.

44. Figure 7: In the Y-axis of plots g-I and j-l, please use similar notation. Use either 0.01 or $10^{-2}$.

45. Lines 866-867 (Figure 7): 'm','n','o' seem to be leftovers.

46. Line 868 (Figure 7): '… and equal to $v_{peak}$, **respectively**'.

47. Figure 8: Plots g-I are presented first and explained later. They can both combine in the same piece of text.

---

## Author Comment (AC1)

**R1**: "This paper has promise for benefitting the hydrologic community in regards to understanding parameterization of transport metrics via modeling, however, I recommend the authors make some substantial revisions to broaden the audience and better clarify the significance of their results. The following are suggestions/ comments I have that I believe would improve this paper and its impact."

**Authors**: We thank the reviewer (R1) for the positive view on our work and for the suggested improvements. We will implement the suggestions in a revised version. In particular, we will improve terminology and structure of the manuscript to broaden the audience and enhance readability.

**R1**: "The effort put into answering and addressing questions 2 and 3 are not to the same rigor as question 1. Methods, results, and discussion for question 1 is clear, but not for questions 2 and 3."

**Authors**: In the revised version of the manuscript we will address these concerns by pointing out the link between methods/results and the respective research questions more clearly. We also may reformulate the research questions to make this clearer.

**R1**: "The paper is jargon heavy and not readable to a wide audience."

**Authors**: We will carefully revise the manuscript to minimize jargon and improve readability.

**R1**: "Certain vocabulary is not defined or explained well, such as global identifiability vs dynamic identifiability."

**Authors**: We will accurately define global identifiability analysis and dynamic identifiability analysis in the revised method section.

**R1**: "Additionally, many sentences and paragraphs are not fluent and have poor sentence structure, especially throughout the introduction."

**Authors**: We will do a careful language editing before resubmission.

**R1**: "The introduction addresses certain open questions / problems but then lists specific questions being answered. It's then confusing as to what questions will and will not be addressed in this paper. I think it would benefit readers to remove the earlier questions mentioned and stick with the ones at the end of the introduction."

**Authors**: In the introduction we indicated certain open questions / problems to indicate what are the possible causes that could have induced non-identifiability of TSM parameters in previous studies. However, based on the comments we understand that they were unclearly presented in relation to the investigated research question. We will re-write or remove these open questions / problems to better develop the introduction toward the investigated research questions.

**R1:** "For the methods section, there is no explanation of how question 2 was answered."

**Authors**: We will modify the manuscript to clearly present how the dynamic identifiability analysis was used to address research question 2.

**R1**: "What specific sections of the BTC did you investigate and how did you break the BTC up into different sections?"

**Authors**: In the manuscript we refer to three different sections of the BTC: (i) "rising limb" that relates to the sections of the BTC preceding the peak, (ii) the "peak" indicates the maximum recorded concentration of the BTC, and (iii) "tail" that indicates the sections of the BTC following the peak. We will describe how we separated the BTC in the different sections in the revised method section.

**R1**: "Also, why are the tracer experiments all done during the same season? I acknowledge that the discharge is still varied, but are there other hydrologic processes that change throughout the year not captured by winter tracer studies?"
**Authors**: This is true, different hydrologic conditions can be characterized by different hydrologic processes at the study site (see Bonanno et al., 2021). However, the focus of this study is on the identifiability of TSM parameters to derive solute transport metrics. The question of how changing processes with changing hydrological conditions impact transient storage is, however, highly interesting and open research. Building on the current modelling work, our follow-up research will address this. However, we believe that this goes beyond a realistic scope of the current manuscript.

**R1**: "I believe a simple reactive transport study using the different parameter estimates would enhance this study and better highlight the impact and importance of the uncertainty in parameter estimation."
**Authors**: This would absolutely be a very interesting complementary study. We believe that coupling reactive tracer experiments with the proposed methodology will offer valuable insights on the physical meaning of transient storage parameters and can be a valuable follow-up of the current study. There is a lot of future research needed to address the complexity of in-stream transport. However, we believe that this would need to be a standalone study.

**R1**: "There is little explanation on whether or not their results make sense in the context of physical hydrologic processes and past studies"
**Authors:** We will clarify how and why our results make sense in the hydrologic context of the study site. The study site is equipped with a dense network of groundwater monitoring wells (Bonanno et al., 2021). The measured groundwater levels in the well network showed that the stream channel is almost entirely in gaining conditions during the investigated tracer injections. These data indicate a lack (or very limited) hyporheic exchange at the study site, which is in line with the obtained TSM transport metrics that show short residence time in the transient storage zone (from ~150 s for E1 to ~33 s for E3, line 354). We will refer to the groundwater monitoring network in the method section and we will implement in the discussion section how our TSM results are in agreement in the context of physical hydrologic processes occurring at the study site. We will also briefly compare our results with previous studies at other sites. Specifically, a decrease of both $A\_TS$ and tracer residence time in the storage zone (Tsto) with increasing discharge conditions was argued to indicate an increase of groundwater gradients toward the stream channel with a consequent decrease of hyporheic zone (Fabian et al., 2011; Morrice et al., 1997). However, groundwater measurements at the study site exclude the presence of significant hyporheic exchange during the three simulated tracer

experiments. In our study, the observed link between TSM results with discharge rather suggests that with increasing discharge conditions the wet stream area and the water depth increased more than the wetted perimeter. This in turn let the wet stream areas to inundate dead-zones and streambed heterogeneities causing a decrease of in-stream transient storage with increasing discharge conditions (Gooseff et a., 2008; Zarnetske et al., 2007).

**R1:** "Does it make sense that a variable velocity that is not determined a priori make sense?"
**Author**: It is important to know what are the consequences and the challenges related to the interpretation of modelling result when one or more parameters are not calibrated and evaluated a-priori. This is especially true for TSM where "parameters may be interactive and correlated, meaning that parameters cannot be changed independently of each other" (cfr. Knapp & Kelleher, 2020). This is because a separate estimation of advection parameters (velocity and area) can lead to changes in the evaluation of the transient storage parameters "exactly because of the observed parameter interactions. Consequently, transport parameters may be mis-estimated, if a two-step approach is applied (Lemke et al., 2013)." (cited from Knapp & Kelleher, 2020). Constraining the values of area in random sampling approach for TSM proved to have a role on the identifiability of Alpha and A_TS (Kelleher et al., 2013; Ward et al., 2017), however, no study so far evaluated the role of velocity on the identifiability of TSM parameter. Thus, we tested if calibrating velocity together with the other TSM parameters may have avoided any possible (and previously unknown) parameter interaction leading to a more robust assessment of Alpha and A_TS. We will clarify the research gap that motivated our modelling choice of considering velocity as a calibration parameter in the revised introduction.

**R1**: "Further, the discussion is sometimes a repeat of results rather than a discussion of the implications of the results. Why do the results matter? How does that align with our current understanding?"
**Authors:** We will revise the discussion to avoid repetition of the results. By doing so, we believe that we will emphasize the sections where we compare our results with previous literature.

**R1**: "Also, the paper is not well organized. The intro clearly outlines 3 questions to be addressed. It would make sense to me, if the methods, results, and discussion followed that order. Instead, section 4.1 addresses section 3.2 which addresses question 1. Section 4.3 addresses question 2, but question 2 does not have a methods or results section."
**Authors**: We will revise the structure of methods and discussion to mirror the order of the research questions. However, we do not think that it will be feasible to separate results related to research question 1 from results addressing research question 2. This is because the results of the dynamic identifiability analysis (research question 2) are used jointly with the global identifiability analysis to obtain identifiable TSM parameters (with and without considering velocity as a calibration parameter). We believe that splitting dynamic identifiability analysis (research question 2) from the results reporting parameter

identifiability (research question 1) will hamper the robustness of the results about the identifiability of TSM parameters. However, to address the reviewer's concerns and improve readability we will iterate clarity and organization. To do so, the research questions at the end of the introduction will be followed by a short paragraph stating how different methods will be used to address the different research questions.

**R1**: "Finally, the discussion and conclusion are missing an explanation of how to go forward from these findings. Line 535 states that process interpretation and parameter evaluation should be used with caution, but how do we address that? What are the next steps?"
**Authors:** We will include a statement addressing this in the revised version of the manuscript. Namely, we will indicate how the adopted modelling strategy can improve parameter evaluation and process interpretation for mathematical modification of the classical formulation of TSM (such as 2-storage zones TSM (Choi et al., 2000), or TSM including sorption kinetics for reactive tracers (Gooseff et al., 2005; Kelleher et al., 2019)).

**R1**: "The study is novel and can be beneficial for those studying hydrologic exchange in river networks and using tracer experiments, but this paper would greatly benefit from some restructuring and re-writing for improved readability and explanation of methods and results."
**Authors**: We thank R1 for the supportive feedback and for highlighting the weakness of the manuscript. We will rework the structure as indicated above and revise accurately both the terminology and the verbosity in every section of the text.

**R1:** "Specific things to address are listed below:

First paragraph of intro is confusing. You mention modeling, then experiments, then talk about issues with modeling again. I think it's worth re-organizing in the following way:

- Understanding transport along river networks is important
- One way is with experiments
- However, current models to describe the processes seen from experiments have contradictory results
- So, it is unknown how informative modeling is"

**Authors**: Thank you very much on this. We fully agree. We will modify the introduction as suggested.

**R1**: "Lines 67-73 is jargon heavy and phrasing is clunky."
**Authors**: We will reformulate the respective sections.

**R1**: "Sentence 103-104, repetitive".
**Authors**: As suggested, we will remove the indicated sentence.

**R1**: "The whole paragraph starting at 113 is confusing."
**Authors**: We will accurately revise and reformulate the paragraph.

**R1**: "Also unclear on if the results make sense in terms of physical processes and our current understanding or if they're spurious relationships (ie non-id A_TS coupled with ID alpha, and when v=vpeak the reverse of that)"

**Authors**: Our TSM results are consistent with the general understanding of the water movement in the investigated stream reach during the tracer experiments. This is because both the measured groundwater table for the stream reach and the obtained TSM transport metrics excluded any relevant contribution of hyporheic exchange on solute transport during the performed experiments. In the discussion, we will improve the relevant section on the interpretation of the TSM parameters and the derived physical processes in relation of the current understanding of the physical processes at study site. Non-identifiable TSM results indicated non-identifiable A_TS coupled with identifiable alpha, and when v=vpeak the reverse of that, however we do not believe the observed relationship are spurious. This is because other studies also found alpha and A_TS to be both non-identifiable when v=vpeak (Ward et al., 2017, Kelleher et al., 2019).

**R1**: "Should review and cite Rathore et al., 2021, "On the Reliability of Parameter Inferences in a Multiscale Model for Transport in Stream Corridors". I believe this paper aligns with your study and might provide some further insight."

**Authors**: We thank R1 for the suggested reference that will surely support our discussion.

**R1**: "No explanation on global identifiability vs dynamic identifiability"

**Authors**: We will improve the definition of global identifiability analysis and dynamic identifiability analysis in the revised methods section.

**R1**: "Should reread Gooseff et al., 2005 and update the sentence starting at line 475 as the statement is incorrect as it stands."

**Authors**: We agree with R1, as this statement was about sensitivity of transient storage parameters that was not the object of study of Gooseff et al., 2005.

**R1:** "Figure 8, what is DYNIA analysis? This is not previously explained."

**Authors**: DYNIA is the "dynamic identifiability analysis". We will replace the acronym.

---

## Author Comment (AC2)

**General comments**

**R2**: "The text content is generally well written but the theme's complexity and the structuring makes honestly the reading rather difficult, which can force the reader going back and forth the chapters to follow a storyline. I found very logical how the text was structured in the 'Study site and Methods' section, and I would really try to stick to this same structure when presenting 'Results', in the 'Discussion' section and even in the 'Conclusion' paragraph. My personal taste would be:

  a.1) ADE parameters
  a.2) TSM parameters (i.e. hydrologic exchange parameters)
    a.2.1. Identifiability of TSM parameters when v = vvariable
    a.2.2. Identifiability of TSM parameters when v = vpeak
  a.3) Model iterations
    a.3.1. TSM first iterations
    a.3.2. TSM last iterations
    a.3.3. DYNIA
    a.3.4. Comparison with inverse modeling results (OTIS-P)
    a.3.5. Comparison with random sampling approaches (OTIS-MCAT)
  a.4) Metrics and hydrologic interpretations of model results

According to this numbering, the 'Results' section starts with the a.3.1 jumps to a.3.3, then back to a.3.2, etc. The 'Discussion' section starts with a.4, then goes back to a.3.1 and a.3.2, follows with a.3.4, etc. This can make a difference between a quick and effective reading, and a tedious reading."

**Authors:** We thank the Reviewer 2 (R2) for the positive and constructive manuscript feedback. We understand the need for a revised manuscript structure. We will adopt the suggested structure for the revised results and discussion sections with a slight modification:

  a.1) ADE parameters
  a.2) TSM parameters
    a.2.1 Identifiability of TSM parameters when v = vvariable
      a.2.1.1 TSM first iterations
      a.2.1.2 TSM last iterations
    a.2.2 Identifiability of TSM parameters when v = vpeak
      a.2.2.1 TSM first iterations
      a.2.2.2 TSM last iterations
  a.3) DYNIA
    a.3.1 DYNIA (first + last iteration - v = vvariable)
    a.3.2 DYNIA (first + last iteration - v = vpeak)
  a.4) Comparison
    a.4.1 Comparison with inverse modeling results (OTIS-P)
    a.4.2 Comparison with random sampling approaches (OTIS-MCAT)
  a.5) Metrics and hydrologic interpretations of model results

We slightly modified the structure suggested by R2, because:
- For achieving identifiability of TSM parameters we need to address model iterations;
- This modified structure is also in agreement with the review of R1, who required the manuscript structure to follow the order of the research questions.

**R2**: "I would like to draw the attention about the use of the term 'parameters' in the manuscript. I personally have no experience with the identifiability of parameters in a TS model and it can bias my understanding of the term. Thus, this might be probably my wrong perception.
As a hydrologist, my understanding of parameters involved in the transport of solutes in a stream/river can be stream discharge, components of flow velocity, flow turbulence, grain size of streambed

sediments, groundwater-stream (GW-SW) water exchange fluxes, stream channel topography, existence/absence of riparian vegetation, etc. In the manuscript, these parameters are treated as such, but then it introduces the term TSM parameters and parameter sets. In this manuscript's TSM model, each iteration simulates 115,000 parameter sets. An unexperienced reader would tend to think that > 100,000 parameters as the ones I mentioned above are involved in the transport of solutes in the stream. This should be obviously clarified."

**Authors:** We agree with R2 that the term "parameters" and "parameter sets" have to be better clarified in the manuscript. In the main text we defined A, v, and D as "advection-dispersion parameters", A_TS and α as "transient storage parameters". The "advection-dispersion parameters" together with the "transient storage parameters" are referred to as "TSM parameters" (cf. lines 183-185). We agree that the similarity between "transient storage parameters" and "TSM parameters" can be confusing, thus we will modify "TSM parameters" as "model parameters" in the revised manuscript version. We will define the term "parameter set" as "a single combination of A, v, D, A_TS, and α from random-sampling" in the revised method section.

**R2**: "The manuscript's title states 'to reduce the uncertainty' but I have missed more uncertainty analysis throughout the text."

**Authors**: It is true that we have not used the term "uncertainty" in the manuscript, and we adopted in the main text the term "identifiability" or "non-identifiability" since we addressed parameter identifiability. Thus, we apologize for the lack of consistency in terms of used terminology. We will uniform and double-check the terminology in the revised manuscript. Namely, we will always use "identifiability" and "non-identifiability" and the title will be modified in: "Exploring tracer information in a small stream to improve the identifiability and enhance the process interpretation of transient storage models". We addressed parameter identifiability via global identifiability analysis based on Generalized Likelihood Uncertainty Estimation (Beven & Binley, 1992). This analysis includes parameter vs likelihood plots, parameter distribution plots, regional sensitivity analysis plots, and identifiability plots (Appendix A). Identifiability of a parameter is generally investigated via visual inspection of regional sensitivity analysis plots (e.g. Kelleher et al., 2019) or visual investigation of parameter distribution plots and parameter vs likelihood plots (e.g. Ward et al., 2017). We here addressed identifiability also via the two-sample Kolmogorov-Smirnov (K-S) test. In this way we could combine visual identifiability of TSM parameters with numerical thresholds defined by K-S test. In the revised manuscript we will emphasize the role of global identifiability analysis and K-S test to address identifiability of TSM parameters.

**General comments**

**Abstract**

**R2**: "The abstract is concise and clearly written. How to fill the research gap is properly addressed.
  1. Line 14: I would consider adding 'adjacent groundwater bodies' as a exchanging agent.
  2. Line 24: … TSM parameters, respectively. The severe differences…
  3. Line 26: … at the study site → The article 'the' makes reference to a determined object, but the study site has not yet been introduced. Consider changing 'the' for 'our' or 'a study site in western Luxembourg', for example."
**Authors**: We thank R2 for the constructive suggestions. We will include the suggested changes in the abstract to improve its clarity.

**Introduction**

**R2**: "The introduction chapter needs to be revised in order to be fully understandable. The content is good, but an improved organization can elevate it."
**Authors**: We thank R2 for the supportive feedback. We will revise the introduction to improve its readability. In agreement with R1, we will re-organize the research gaps. We will also remove lines from 83 to 92, since we do not address a numerical modification of TSM.

**R2**: "Tend to list consecutive citations in chronological order (e.g. lines 72-73, 78-79, etc.)."
**Authors**: We will list consecutive citation in chronological order in the entire manuscript.

**R2**: "Line 49: Many readers can disagree with such a definition of 'hyporheic zone'. I don't agree with the idea of a fully saturated thickness. I usually consider the definition given by Cardenas & Wilson, 2007 (Cardenas, M. B., & Wilson, J. L. (2007). Exchange across a sediment–water interface with ambient groundwater discharge. Journal of Hydrology, 346, 69– 80), as a good approach."
**Authors**: We thank R2 for the suggested definition. The definition of the "hyporheic zone" is still controversial and not univocal. We will clarify as follows: "the saturated area that is physically influenced by water and solutes exchange between the stream channel and the adjacent groundwater" including the suggested paper in the citations.

**R2**: "Lines 79-82: This reads like a Research gap, and it is well stated, but it should fit at a later stage, before enumerating hypothesis and research questions."
**Authors**: We agree with R2 that this statement should come before addressing research questions. We will move it before the section that introduces the research question

**R2:** "Lines 103 and 113: I would not change paragraph in this sentence."
**Authors**: We will modify accordingly.

**R2:** "Line 105: 'stream velocity' → The stream itself does not move. Better using 'streamflow velocity'."
**Authors**: We will double-check the use of "streamflow velocity" in the entire manuscript.

**R2:** "Line 128: I would state the hypotheses in a new paragraph."
**Authors**: We will move the hypothesis in a new paragraph, together with the research gap moved from lines 79-82.

**Study site and methods**

**R2:** "Line 152: Study site and 'data' → Better specify which data."
**Authors**: We will clarify.

**R2:** "Study site and data → How long is the studied stream reach?"
**Authors**: The stream reach is 55 m long (cf. line 165 of the original manuscript).

**R2:** "Lines 161-162: How was Q calculated? (± Analytical Error?)"
**Authors**: Q was calculated from the tracer BTCs. As we did not perform repeated measurements, we cannot estimate the analytical error. However, the most common estimated error for discharge from dilution gauging method is ± 8% (Schmadel et al. 2010).

**R2:** "Line 164: How was EC calibrated according to NaCl concentrations? Was EC temperature compensated? (± Analytical Error?)"
**Authors**: EC-Cl conversion was done in laboratory using a known-volume sample of streamwater (taken before tracer injection) and adding known-quantities of a solution with a known concentration of Na-Cl. The regression line between EC-Cl had a R2=0.9999 (line 165-166). Temperature compensation was automatically considered via Nonlinear temperature compensation (nLF, according to EN 27 888). We will clarify this in the revised method section.

**R2:** "Was there any difference in background EC between the injection and the measurement locations? Or was it assumed to be equal? Or is it one of the testing parameters? These aspects are actually interesting."
**Authors**: The background concentration at the injection point is slightly lower from the background concentration at the measurement location. We account for this difference during the EC-Cl conversion by sampling the streamwater used for the EC-Cl conversion at the measurement location. We will clarify this detail in the revised version of the manuscript.

**R2:** "Line 181 (Equation 3): Why the second term of the second equation is negative?

If Cs < C, the concentration of a certain solute in stream tends to get diluted. But according to eq. 3, ∂C/∂t would become larger"

**Authors**: We double checked the equation 3, and we confirm that the second equation is negative. It could also be written as (Harvey, Wagner, & Bencala, 1996):

$$\frac{dC}{dt} = \alpha \, \frac{A}{A_{TS}}(C - C_{TS}).$$

Note that C_TS is not defined a-priori, as larger or smaller than C, but it is function of C itself and of the concentration C_TS at the time-step antecedent the one investigated. Crank-Nicolson approximation for the second equation of the system of equation 3 reads:

$$C_{TS}^{j+1} = \frac{(2 - \gamma)C_{TS}^{j} + \gamma\left(C_i^j + C_i^{j+1}\right)}{2 + \gamma}$$

Where: $\gamma = \alpha A \Delta t / A\_TS$; $i$ denotes the central segment of the central-divided differences; $j$ denotes the investigated initial timestep, while $j+1$ denotes an advanced time. The full explanation of the Crank-Nicolson approximation for TSM resolution can be found in Runkel & Chapra (1993).

**R2:** "2.2: How the accuracy of the ADE and TSM results are assessed?"
**Authors**: Accuracy of both ADE and TSM results is assessed via Root Mean Squared Error objective function (lines 186-187).

**R2:** "Line 207: I assume CDF is the Cumulative Distribution Function, but it has not been defined as such yet."
**Authors**: R2 is right, we will clarify the abbreviation in the revised methods section.

**R2:** "2.3: How is the uncertainty of the model iterations' results, DYNIA's results and the comparison with OTIS-P and OTIS-MCAT assessed?"
**Authors**: We clarified uncertainty vs. identifiability above. Non-identifiability of TSM results and OTIS-MCAT was determined from global identifiability analysis and K-S test (lines 195-216). Identifiability in the dynamic identifiability analysis can be addresses via the metric called "Information content", defined as: "one minus the width of the 90 % confidence interval over the entire parameter range (Wagener et al., 2002)." Thus, a large 90 % confidence interval is associated to a low information content and uncertain/non-identifiable parameter (lines 221-229). Identifiability in OTIS-P (95% confidence limits) are derived in OTIS-P package via linear numerical approximation of the variance–covariance matrix within a local neighborhood of the solution (Runkel 1998; Ward et al., 2017). We will clarify how identifiability in OTIS-P is evaluated.

**R2:** "Line 270 (Equation 7): Introduce a tab in the equation's label, so it can be in line with the rest of the equations."
**Authors**: Nice catch! We will insert a tab prior to "Eq. 7".

**R2:** "2.4: I assume that including FMED reduced the systematic error, but is it quantifiable?"
**Authors**: Unfortunately, Fmed does not act on the reduction of error and non-identifiability of transient storage modelling. It is a non-dimensional metric used to interpret TSM results in comparison to other reaches or discharge conditions (Gooseff et al., 2013; Runkel, 2002, and section 2.4).

**Results**

**R2:** "This chapter is properly written but the sub-chapters could be re-organized to build-up a smoother storyline. See the last part of General comment (a)."
**Authors**: We thank R2 for the supportive suggestions for improvement. In light of the feedback of both R1 and R2 we will revise the structure of the results section to mirror the order of the research questions and the order of the revised method sections.

**R2:** "Lines 279-282: This reads more like in the 'Methods' section, right?"

**Authors**: We will remove this short paragraph to avoid redundancy between method section and discussion section.

**R2:** "Why was vpeak and no other velocity chosen for the fixed velocity scenario? Why not median velocity which could be more a representative velocity value?
What does vpeak mean?
- Peak velocity during tracer experiments?
- Mean velocity during the experiment E3 which had the largest discharge?
- Peak velocity during the entire monitoring period?"
**Authors**: We will clarify that v_peak is the velocity obtained via v_peak=L/tpeak. Where L is the length of the investigated reach and tpeak is the time for concentration peak to travel through study reach. We used v_peak as it is commonly used in many transient storage studies (Ward et al., 2013; Kelleher et al., 2013; Wlostowski et al., 2017; Ward et al., 2017; Ward et al., 2019a; Ward et al., 2019b). We will clarify this choice in the revised version of the method section.

**R2:** "Line 292: Figure 3c-f →Is Figure 3f informative in the tailing of the BTC?"
**Authors**: Yes, it is in our view. We will clarify in the revised manuscript that by "tail" we meant sections of the BTC from the peak to the end of the BTC.

**R2:** "Line 297: Add a comma after > 0.05),"
**Authors**: We will add a comma after "0.05".

**R2:** "8.8 l/s for E3 → I would not consider this discharge as a low qs, since you state that the exchanging flow is > 1/3 of the total Q of the stream."
**Authors**: We will remove subjective terms as "low", "high", "long" in the manuscript to avoid unprecise writing.

**R2:** "Line 343: 15 hrs for E3 → I would not consider this as a long residence time. The flux velocity is roughly 4.6 m/hr, which according to me can be slow for dead zones, but not for hyporheic flux, or GW-SW exchange."
**Authors**: We will remove subjective terms as "low", "high", "long" in the manuscript to avoid unprecise writing and subjectivity.

**R2:** "Lines 345-348: are these simulations physically possible at all, knowing the actual stream discharge? As for Tsto, do 0.8 (in E1) and 3.3 (in E3) m/s make sense?"
**Authors**: After the first TSM iteration, the parameters were not identifiable thus they may not make physically sense. We would recommend to only check for physical realism after the parameters became identifiable. However, in the following we will present the physical realism of the exchange rate (qs). We exemplified that for E1 (Q = 2.52 l/s).
Following the OTIS_MCAT results, the exchange rate over the stream length of 55 m was 23.1 l/s (cf. line 345). In this case the estimated storage volume in the stream channel was 1930 l. After the fourth TSM iteration the exchange rate over the stream length of 55 m was 2.7 l/s (cf. line 353). In this case the estimated storage volume in the stream channel was 2029 l. It is evident that both rates would be physically possible. This holds true for the other transport metrics. These results make clear that is important to reach parameter identifiability since non-identifiable results can reproduce physically possible behavior.

**R2:** "Lines 353-354: Again… double check the applicability of these results into the actual flow regime."
**Authors**: The results are physically possible. We have already outlined for E1 in the answer above. Here we summarize the same results for E2 and E3 after the last TSM iteration. Storage volume in the stream channel for E2 is ~2'850 l while the exchange rate over the stream length of 55 m was ~5 l/s. Storage in the stream channel for E3 is ~4'500 l while the exchange rate over the stream length of 55 m was ~23 l/s. Since all the evaluated storage metrics are interlinked by Alpha and A_TS the physically realism is preserved.

**R2:** "Comments 26, 27 and 28: Since you are doing a comparison between methods, rather than using terms like 'high' or 'low', it is better to use relative terms such as 'higher' or 'lower', or even 'distinct'."
**Authors**: We will account for this in the revised manuscript.

**R2:** "Line 349: 'whether' does not sound appropriate in this context. I would rather use 'regardless'?"
**Authors**: We will adapt accordingly.

**Discussion**

**R2:** "Same as in the previous chapter, the sub-chapters could be re-organized to build-up a smoother storyline. See the last part of General comment (a). Aren't 4.1 and 4.3 more related to each other, and 4.2 a unique sub-chapter. I think starting with 4.2 and then following with 4.1 and 4.3 would improve the chapter's sequence."
**Authors**: We will adapt by starting with 4.2, followed by 4.3 and 4.1. These modifications will allow us to respect the order of the research question, as suggested from R1.

**R2:** "4.1: Are the uncertainties specified?"
**Authors**: The uncertainties of the transport metrics are reported as boxplot limits (5-95 percentiles) of the top 100 model results for every TSM iteration in Figure 7.

**R2:** "Lines 402 and 406-407: Aren't these contradictory statements? Identifiability is contradictory under the same scenario (in both cases v = vpeak)."
**Authors**: In the revised manuscript we will clarify that non-identifiable alpha was found when velocity was assumed equal to v_peak.

**R2:** "Line 438: 100,000 instead of 100'000."
**Authors**: We will change accordingly.

**R2:** "Line 446-447: Seems like a very generalist closing statement."
**Authors**: We will revise to be more specific.

**R2:** "Lines 460-462: Aren't these contradictory statements? The assumption v = vpeak might not be representative of the advection role, but can encompass the effect of advection in the entire BTC."
**Authors**: As it is written, these statements could indicate that we suggest that vpeak can and cannot be representative of the advection role on the BTC. We will reformulate the sentences to make clear that we were comparing our findings to past studies.

**R2:** "Lines 480-503: Figure 8 is actually a very good review from other authors' findings. I quite like it. The piece of text relating to this figure is actually an important part of your discussion, but again it reads like a succession of sentences with vague organization, and it forces the reader to continuously go back and forth between the text and the figure. I propose the following. Instead of travelling through all the cited authors and comparing your results with their results, why can't it be presented in concordance with the different BTC's features, e.g.:
- Rising Limb
- Peak
- Falling Limb
- Tail
One can travel along the BT curve and compare your data with literature data. The reading would be more graphical and intuitive, and it could probably help to safe some word spacing.
By the way, it is interesting to know whether there is a method or a simple threshold value to distinguish between the end of the falling limb and the beginning of the tail?"
**Authors**: We will reformulate this section as suggested by R2. There is no univocal method or threshold value to distinguish between falling limb and the tail of the BTC. Therefore, we decided to consider as "tail" the section from the peak to the end of the BTC. We will include this information in the revised manuscript.

**R2:** "Lines 502-503: '… Kelleher et al., 2013 also indicate…" → whereas it has a strong influence based on your results, right?"
**Authors**: We will emphasize the difference between our results and Kelleher et al. (2013) results in the revised 4.3 section.

**R2:** "Lines 503-504: 'Different sensitivity…' → This reads like the staring of the next paragraph."
**Authors**: We will change accordingly.

**Conclusion**

**R2:** "The conclusion reads well, but again, I would try to expose the ideas trying to follow the same sequence of findings shown in the results."
**Authors**: We will adapt the conclusion to follow the revised structure of the methods, results, and discussion.

**R2:** "Line 523: '… that the BTC…' -→ Revise wording."
**Authors**: We will revise and clarify this sentence.

**Tables and Figures**

**R2:** "Line 739 (Table 2): $v = vpeak$ → Try to keep the same notation as in the text (i.e. Italic and subscript)."
**Authors**: We will uniform the fonts of *vpeak* and others.

**R2:** "Line 777 (Figure 2): The meaning of colored dots for (m-o) plots not specified."
**Authors**: We will clarify that the indicated colors refer to plots from "k" to "o".

**R2:** "Line 778 (Figure 2): Try to keep the same notation as in the text (i.e. Italic and subscript)."
**Authors**: We will unify the font of *vpeak*.

**R2:** "Figure 3: Here you show the BTC in E1 when the streamflow is smaller and the identifiability is probably less dynamic. I would have liked to see differences in dynamism between the different tracer experiments. This does not mean that you need to re-do the figures for other tracer experiments, but probably comment a bit on it in the text."
**Authors**: The dynamic identifiability analysis showed that alpha and A_TS are highly identifiable (inf content > 0.66) for smaller section of the tail of the BTC with increasing discharge conditions. For example, the information content of A_TS is above 0.66 for 51% of the tail of the BTC for E1, for 23% for E2, and for 19% for E3. However, we decided to not discuss differences and similarities for E1, E2, E3 after the 1st TSM iteration, since the dynamic identifiability analysis after the 1st TSM iteration is conducted with non-identifiable TSM results (figure 2, green points). We decided to compare dynamic identifiability analysis results only for identifiable TSM results via the qualitative plots in Figure 8 to keep the manuscript at a readable length. However, we will briefly comment the observed difference in terms of identifiable alpha and A_TS on the tail of the BTC for the first TSM iteration in the revised result sections.

**R2:** "Figure 4: I would probably combine Figs. 3 and 4 in the same figure. Both are showing exactly the same info (Experiment E1 and v = vvariable) but only for different iteration states, and can be potentially confusing."
**Authors**: We will change accordingly. The revised figure 3 will include subplots from (a) to (t). Subplots (a)-(j) will report results of dynamic identifiability analysis for the 1st TSM (old figure 3). Subplots (k)-(t) will report results of dynamic identifiability analysis for the 4th TSM (old figure 4).

**R2:** "Figure 7: In the Y-axis of plots g-I and j-l, please use similar notation. Use either 0.01 or 10-2."
**Authors**: We will unify the numerical notion.

**R2:** "Lines 866-867 (Figure 7): 'm','n','o' seem to be leftovers."
**Authors**: We will remove these letters.

**R2:** "Line 868 (Figure 7): '… and equal to vpeak, **respectively**'."
**Authors**: We will adapt accordingly.

**R2:** "Figure 8: Plots g-l are presented first and explained later. They can both combine in the same piece of text."
**Authors**: We will present and explain the plots (g)-(l) in the same part of the figure description.

---

## Author Comment (AC3)

**General**

**R3:** "The manuscript highlights an important aspect of inverse modeling based on stream tracer test – the uncertainty and identifiability of the estimated model parameters and the connected metrics used to assess solute transport processes in streams. Although I think that the text is generally well written, the structure of the paper (e.g. order of sections) could be streamlined and improved for clarity. As it is right now, I found myself going back and forth between the different sections in order to understand the iterative and rather complex structure. Overall, I appreciate the approach and the general topic, but I believe the manuscript would benefit from addressing a number of issues highlighted below."

**Authors**: We thank Reviewer 3 (R3) for the time spent in reviewing our manuscript and for the constructive and supportive feedback. In the revised manuscript version, we will re-arrange the structure of the method, results, and discussion section to improve the overall clarity. Consistent with comments and suggestions from R2, we will modify the method section as follows:

      a.1) ADE parameters

      a.2) TSM parameters

           a.2.1 Identifiability of TSM parameters when v = vvariable

                a.2.1.1 TSM first iterations

                a.2.1.2 TSM last iterations

           a.2.2 Identifiability of TSM parameters when v = vpeak

                a.2.2.1 TSM first iterations

                a.2.2.2 TSM last iterations

      a.3) DYNIA

           a.3.1 DYNIA (first + last iteration - v = vvariable)

           a.3.2 DYNIA (first + last iteration - v = vpeak)

      a.4) Comparison

           a.4.1 Comparison with inverse modeling results (OTIS-P)

           a.4.2 Comparison with random sampling approaches (OTIS-MCAT)

      a.5) Metrics and hydrologic interpretations of model results

The revised results and discussion sections will mirror the same order that will be used in the revised method section. This modification will be in line with the suggestion of R2 and with the requirements of R1.

**Major comments:**

**R3**: "I think that the terms "identifiability" and "sensitivity" needs to be defined early on in the introduction so that there is no doubt what is meant by these key terms in the context of the study. As it is now the words are used already from the start of the paper, but the definitions are "hidden" in Appendix A. I suggest moving the definitions (lines 902-905) and placing them in the main paper where they are first introduced."

**Authors**: We will implement these changes.

**R3**: "There are several threshold values used to select behavioral parameter sets and subsequently to define identifiability. However, these thresholds are not sufficiently motivated and discussed, which makes it difficult for the reader to assess the results. This holds for the definitions of both the global identifiability (e.g. the top 0.1-10% of the models when assessing the CDF deviation from the 1:1 line having, the grouping based on the K-S results) and the dynamic identifiability (e.g. information content > 0.66). This leads to questions about the subjectivity of these thresholds when assessing the identifiability and how sensitive the overall assessment of the model parameters are to the chosen thresholds (e.g. in Figure 8)."

**Authors**: Selecting a certain behavioral threshold introduces a certain degree of subjectivity to the results. Thus, it is important to remain consistent with previous literature and the previously selected behavioral thresholds. We assured consistency with previous work by selecting the same behavioral thresholds used in a wide range of transient storage studies that used a 10% threshold (Wagener et al., 2002; Wlostowski, 2013; Ward 2013; Ward 2019; Kelleher 2019) and 0.1% threshold (Ward et al., 2017). We will clarify this choice in the revised manuscript.

For K-S test we also oriented our selection of thresholds on current literature (Ouyang et al. 2014). However, K-S test was never used in TSM. We decided to adopt K-S test and the reported thresholds to assure a more robust assessment of identifiability of TSM parameters, since parameter identifiability is usually assessed via visual interpretation of global identifiability analysis results (Wagener et al., 2002; Wlostowski et al., 2013; Ward et al., 2013; Ward et al. 2018; Ward et al., 2019; Kelleher et al., 2019). We will clarify this choice in the revised manuscript.

For the dynamic identifiability analysis, we were interested if a certain parameter was poorly identifiable (information content < 0.33) or strongly identifiable (information content > 0.66) in a certain section of the BTC. While the reported thresholds are arbitrary, they assure consistency in terms of parameter interpretation compared to previous work, where the role of a certain parameter on the BTC was controlled by a certain degree of subjectivity since dynamic identifiability analysis was visually interpreted (Wagener et al., 2002; Wlostowski et al., 2013).

We will extend the discussion of the revised manuscript by accounting for the role of the thresholds in the adopted modelling approach. It is clearly of interest to explore the impact of the selection of the threshold on the physical realism of the results in future work. However, our results are physically realistic, and the selected thresholds proved to be effective in the interpretation of model outcomes and in the achievement of parameter identifiability.

**R3**: "The objective function used to assess the model performance, i.e. the RMSE, is based on the difference between observed and modelled BTC over a given time scale/number of observations. What are the effects on the RMSE for concentration values that may differ more than an order of magnitude over assessed time window, i.e. for high values (peak of the BTC corresponding to 50 mg/l) compared to the low values (tail of the BTC corresponding to <1 mg/l)? How does this affect (i) the global identifiability analysis and (ii) the dynamic identifiability? Previously alternative objective functions have been suggested, including RMSE with a logarithmic (e.g. Ward et al., 2018) or mixed scale (e.g. Bottacin-Busolin et al., 2011; Riml et al., 2013), to account for the different magnitudes of the concentrations across the BTC. I suggest that the authors assess this and discuss the implications.

**Authors**: We used RMSE because it is an equivalent form of RSS that is used in the calibration of OTIS-P and of mean absolute error used in the dynamic identifiability analysis. Thus, RMSE allowed us a consistent comparison of our TSM results with OTIS-P and DYNIA outcomes. For the same reason, RMSE is the preferred objective function in studies where random-sampling TSM simulation are compared to OTIS-P results (Ward et al., 2017) or coupled with DYNIA approach (Wlostowski et al., 2013). We will clarify our choice in the revised manuscript.

However, our model tested several objective functions for every TSM iteration (Nash-Sutcliff efficiency, Pearson's R2, Kling–Gupta efficiency, normalized RMSE, log-transformed RMSE, log-transformed R2, log-transformed NSE). As expected, there are some consistencies and some inconsistencies between the different objective functions. This has been shown for other modelling applications (Gupta et al., 2009; Ouyang et al. 2014; Wagener et al., 2002). It is surely a valuable follow-up study to evaluate the consistency of identifiability of TSM parameters as function of the selected objective function, as this problem has never been directly tackled in TSM. However, this analysis is far beyond the scope of the paper. We will highlight this important point in the revised discussion section.

**R3**: "Moreover, I miss a visual comparison of the observed and simulated BTC as a complement to the presented RMSE values, preferable using log-transformed concentrations to highlight how well the

model captures the tail of BTC that is argued to be of importance for transient storage processes (e.g. lines 55-57).”

**Authors**: We will add in the appendix the following figure (Figure A). Despite the achieved parameter identifiability and the relatively low RMSE values obtained at the end of the proposed iterative approach, it is evident that the best-fitting BTCs are unable to reproduce the last section of the tail of the observed BTCs. This might be driven by the selected exponential RTD in the formulation of the TSM, and/or by the selected objective function (RMSE). As an example, the best-fitting BTC obtained at the end of the second TSM iteration shows a visually better fit on the BTC tail (Figure B) despite the large RMSE (1.5197 mg/l). We will implement the manuscript discussing the role of the used objective function and of the exponential RTD in the BTC fitting. We will extend the discussion accordingly.

[Figure]

Figure A: Observed BTC (red line) together with the grey area comprised between the top 100 simulated BTCs and the best-fitting BTC (blue dashed line) for (a, d) E1, (b, e) E2, and (c, f) E3. Results reported for the first (a, b, c) and last (d, e, f) TSM iterations.

[Figure]

Figure B: Observed BTC (red line) together with the grey area comprised between the top 100 simulated BTCs and the best-fitting BTC (blue dashed line) for the second TSM iteration (E1).

**R3**: “The fact that using erroneous model parameter estimates (obtained either from the literature, from a simplified model (ADE) or from a Monte-Carlo simulation with to wide parameter ranges and/or not sufficient iterations) leads to uncertainty/errors when estimating the transport metrics (Eqn 5-8) is rather intuitive. Firstly, I find (the rather long) discussion in Section 4.1 as well as the conclusion (line

24-26) and abstract (lines 21-26) about misinterpretations/uncertainty when comparing the different models "unfair". The conditions for the OTIS-MCAT simulations seems to be equivalent to the first iteration of the proposed methodology. Thus the conditions when OTIS-MCAT was used differ substantially from the 3-4 iterations with a successive refinement of the parameter ranges of the proposed methodology. I understand that the authors would like to make a point and compare the results against an existing model framework, but I think that manuscript would benefit from significantly downplaying the role of OTIS-MCAT. I would prefer a stronger focus on how the refinement of parameter ranges using the existing model framework resulted in a reduced uncertainty/increased identifiability of the model parameters."

**Authors**: We fully agree with R3. One of the major points of the manuscript is the show how our approach can clearly reduce the non-identifiability of the parameters via the proposed iterative approach. We will revise the manuscript to emphasize the benefits of the refinement of the parameter range to increase parameter identifiability. To do so, we will include in the appendix the following figure (Figure C). This figure emphasizes the role of the defined parameter range over the role of the number of parameter sets. In the original version of the manuscript, we felt the need to compare our results with a currently existing framework (OTIS-MCAT). However, in the revised manuscript we will downplay the results of OTIS-MCAT, and we will rather emphasize the role of the iterative narrowing of the parameter range to achieve parameter identifiability.

[Figure]

Figure C: Mean (red lines, left axes) and standard deviation (blue lines, right axes) for RMSE values relative to the top 10% of the modelling results as a function of the number of parameter sets used in the TSM. The results are reported for the (a) first TSM iteration and the (b) last TSM iteration (E1). Vertical black lines indicate the number of parameter sets needed to have the shown percentage difference between the mean RMSE value calculated at a certain number of parameter sets and at 115'000 parameter sets. Eg: In plot (a) after 50'000 parameter sets there is less than 25% difference in top 10% RMSE values compared to results using 115'000 parameter sets.

**R3**: "Secondly, although the authors successfully reduced the uncertainty of the model parameters by an iterative and smart sampling of the parameter space, it was surprising to see that the results from the OTIS-P outperformed (2 out of 3 experiments) the proposed methodology when using the objective function preferred by the authors (RMSE, table 2). This is something that is not sufficiently discussed in the paper and, in my view, opens for questions when the iterative sampling procedure is needed."

**Authors**: We thank R3 for this comment. Indeed, the similarity with OTIS-P results is interesting and we will discuss this more clearly in the revised manuscript. This finding raises the question *if* and *when*

the random sampling approach for TSM is really needed and *if* and *when* OTIS-P results can be considered reliable.

**R3**: "Could the DYNIA approach be combined with OTIS-P using a given confidence interval as input for the parameters ranges to assess the identifiability in parameter estimates from OTIS-P?
**Authors**: We thank R3 for this comment. The idea to use the DYNIA together OTIS-P is really interesting and could be a novel modelling practice and a strong follow-up of this project.

**R3**: "I guess that similarity in performance between parameters obtained by OTIS-P and the proposed sampling procedure might connect to the objective function used to evaluate the performance (see major comment #3), the limited number of experiments (in a single reach) and how initial values (and the possibility of finding a local minimum when optimizing) were defined in the OTIS-P model."
**Authors**: We will implement the discussion with these observations. However, OTIS-P was used starting from several initial values and it was iteratively run until convergence of the model results. Thus, we exclude the finding of local minimum during the optimization. We will clarify this in the revised manuscript.

**R3**: "I believe that the paper lacks a thorough discussion regarding different model representations of transient storage. Eq. 3 assumes an exponential residence time distribution (RTD) in the transient storage zone as originally defined in the TSM model (e.g. Bencala and Walters 1983). Subsequently other type of exchange models and RTDs in the transient storage zone have been introduced (e.g. Wörman et al., 2002; Haggerty et al., 2002). I see a great advantage of the proposed model framework – compared to the OTIS-MCAT and OTIS-P – to explore alternative model formulations (including multiple transient storage zones) and alternative RTDs. This flexibility, when discussed properly, could levitate the readers understanding of the usefulness of the model framework."
**Authors**: This is a very interesting suggestion. As also suggested by R1, our discussion section lacks a paragraph exploring modelling implication for reactive solutes or for models with different RTD or with multiple storage zones. We will implement the discussion section with a paragraph exploring possible implication for other TSM formulations.

**Detailed comments:**

**R3**: "Line 26-29: This is unclear, what is meant by "clear potential"?"
**Authors**: By "clear potential for increasing identifiability" we referred to the used iterative modelling approach to improve identifiability of TSM parameters. We will clarify.

**R3**: "Line 53: "The numerous contradictory outcomes", in what context? Please clarify"
**Authors**: We referred to contrasting interpretation of TSM outcomes and the change of TSM parameters values with hydrologic conditions and scales. We will clarify this in the revised introduction.

**R3**: "Line 109-111: Unclear. "to keep constant a rather identifiable parameter". Please rephrase."
**Authors**: We will reformulate this sentence.

**R3**: "Eq. 3: I believe that CS should be replaced by CTS in the bottom equation
**Authors**: We will correct it in the revised manuscript.

**R3**: "Line 216: I miss information of the width of the window in the DYNIA.
**Authors**: Following Wagener et al. (2002), we used a window size of three time steps (~1 min for E1 and E2, and ~15 secs for E3). We will clarify this detail in the revised manuscript.

**R3**: "Line 233-234 "The best 1% of the results were used to define its parameter space in the successive TSM iteration". Not in agreement with Figure 1 that says "New parameter range defined from the top 10% of the results". Please revise.
**Authors**: The new parameter range is defined from the top 10% of the results. We will revise in the text.

**R3**: "Line 245-246: vpeak is not clearly defined. Is this the time from injection to the BTC peak divided by the stream length (i.e. 55 m)? Please clarify."
**Authors**: We will clarify that $v\_peak$ is the velocity obtained via $v\_peak=L/tpeak$. Where L is the length of the investigated reach and tpeak is the time for concentration peak to travel through study reach. We used $v\_peak$ as it is commonly used in many transient storage studies (Ward et al., 2013; Kelleher et al., 2013; Wlostowski et al., 2017; Ward et al., 2017; Ward et al., 2019a; Ward et al., 2019b).

**R3**: "Line 249: How was the set up of the OTIS-P simulations in terms of initial parameter values? Does "multiple OTIS-P iterations" mean that several initial conditions were tested to reduce the risk of ending up with in local minimum in the optimization? Please clarify."
**Authors**: We will clarify that we used different initial parameter values to avoid a local minimum in the OTIS-P simulation. Following Runkel (2008), we ran OTIS-P multiple times and used, as initial values of a subsequent run, the final estimate from the previous run. We finalized the simulations when the parameter estimates changed less than 0.1% between subsequent runs. We will clarify this in the revised manuscript.

**R3**: "Line 255-258: What parameter ranges were used and how many iterations were performed with the OTIS-MCAT? It is difficult to compare the results if the simulation conditions are not provided."
**Authors**: OTIS-MCAT results are obtained using the range indicated in Table 1 with the condition of velocity considered as a fixed-parameter equal to $v\_peak$. Thus, OTIS-MCAT results are the results of the 1st TSM iteration for the case v=v_peak (cf. lines 257-258).

**R3**: "Line 293-295: From Figure 3, it is unclear how ATS < 5.356 m2 has information content > 0.66, Figure 3 i,j). Previously (line 223-224) it is stated that the information content is expressed as one minus the width of the 90 confidence interval, which I assume uses the entire parameter distribution. Please clarify why there is no lower bound on the confidence interval."
**Authors**: In the revised results we will specify that the lower boundaries of the confidence interval for $A\_TS$ on the tail of the BTC were 0.77 $m^2$ for E1, 0.32 $m^2$ for E2, and 0.33 $m^2$ for E3 after the first TSM iteration.

**R3**: "Moreover, although I realize that this is the first iteration, to have a transient storage area several orders of magnitude larger than the cross-sectional area of the stream makes little sense."
**Authors**: The rather large order of magnitude ($10^1 \, m^2$) for the A_TS upper limit in the first TSM iteration is justified by two observations: 1) groundwater monitoring in the study site let us observe the occurrence of water gradients pointing from the stream channel toward the adjacent groundwater several meters away the stream talweg (see red arrows on the right bank in figure 7f in Bonanno et al., 2021). This suggests the occurrence of a hyporheic zone adjacent the stream channel potentially extending for more than 10 $m^2$ despite the relatively small size of the investigates reach; 2) The chosen order of magnitude is consistent with previous literature for TSM applied in headwater stream reaches (Wagner and Harvey, 1997; Ward et al., 2013; cf. Table 1).

**R3**: "Figure 3. I suggest to show the y-axis of the alpha plot (Figure 3 g) using a log scale, due to the small values."
**Authors**: We will modify this plot as suggested by R3.

**R3**: "Line 312-314: How much of this result can be derived to the used parameter ranges and the number of iterations? If the MC analysis would been set up differently (smaller parameter ranges, larger number of iterations), how would the result differ?"

**Authors**: We outlined the role of parameter range and number of parameter sets in the answer before by introducing Figure C. This figure will be inserted in the revised Appendix. We believe Figure C highlights the pivotal role of the parameter range over the number of parameter sets. It also increases the value of the used iterative approach in constraining the parameter range in random sampling TSM.

[Figure]

Figure C: Mean (red lines, left axes) and standard deviation (blue lines, right axes) for RMSE values relative to the top 10% of the modelling results as a function of the number of parameter sets used in the TSM. The results are reported for the (a) first TSM iteration and the (b) last TSM iteration (E1). Vertical black lines indicate the number of parameter sets needed to have the shown percentage difference between the mean RMSE value calculated at a certain number of parameter sets and at 115'000 parameter sets. Eg: In plot (a) after 50'000 parameter sets there is less than 25% difference in top 10% RMSE values compared to results using 115'000 parameter sets.

**R3**: "Line 338: "orange boxplots" is labeled "red boxplots" in Figure 7. Please revise."
**Authors**: We will revise it.

---

## Author Response (AR1)

**Response to Reviewers**

Bonanno et al.

We would like to thank the Editor and the reviewers for their effort and time in reading and improving the manuscript. The reviewers' comments are reported in black font, while the authors' responses are indicated in red font. The lines mentioned in the response refers to the revised version of the manuscript.

**Response to Reviewer 1**

**R1**: "This paper has promise for benefitting the hydrologic community in regards to understanding parameterization of transport metrics via modeling, however, I recommend the authors make some substantial revisions to broaden the audience and better clarify the significance of their results. The following are suggestions/ comments I have that I believe would improve this paper and its impact."

**Authors**: We thank the reviewer 1 (R1) for the positive feedback and the extensive revision. We accurately revised the entire manuscript, we unified the terminology, and reduced the verbosity. We hope the proposed changes satisfy R1's requirements for improved readability.

**R1**: "The effort put into answering and addressing questions 2 and 3 are not to the same rigor as question 1. Methods, results, and discussion for question 1 is clear, but not for questions 2 and 3."

**Authors**: In the revised manuscript version, the research questions have been slightly modified and read (lines 136 – 144):

1) How does the identifiability of model parameters change in the random sampling of TSM when velocity is considered as a calibration parameter and when it is assumed fixed and equal to $v_{peak}$?
2) Does the identifiability analysis on specific sections of the BTC reduce the parameter non-identifiability in random sampling of TSM?
3) How much does the identifiability of model parameters in random sampling approaches depend to the used parameter range and on the number of parameter sets?

With the outcomes of these questions we will address:

4) How does the hydrologic interpretation of TSM results vary when model parameters are identifiable and when they are not?"

The introduction of a new research question (research question 3) was done to satisfy the requirement to have a research question targeting each research gap presented in the introduction (see a later comment) and the requirement of R3 to emphasize the role of the parameter range to obtain parameter identifiability. We modified the method section to clearly show how we targeted each research question. The revised method structure reads:

> 2 Study site and methods
>> 2.1 Study site and tracer data
>> 2.2 Advection-dispersion equation and Transient Storage Model formulation
>> 2.3 Random sampling and global identifiability analysis (**Targeting research question 1**)
>> 2.4 Identifiability analysis on specific sections of the BTC (**Targeting research question 2**)
>> 2.5 Iterative approach to achieve model identifiability
>> 2.6 Number of parameter sets, parameter range, and identifiability of model parameters (**Targeting research question 3**)
>> 2.7 Comparison with an inverse modelling scheme and a Monte Carlo random sampling approach
>> 2.8 Metrics and hydrologic interpretation of TSM results (**Targeting research question 4**)

**R1:** "The paper is jargon heavy and not readable to a wide audience."

**Authors**: We carefully revised the entire manuscript improving its readability.

**R1**: "Certain vocabulary is not defined or explained well, such as global identifiability vs dynamic identifiability."

**Authors**: In the revised method sections we defined global identifiability analysis as the identifiability analysis that uses the "model performance (RMSE) evaluated on the entire BTC" (line 204). The Dynamic identifiability analysis has been defined in the revised section 2.4 (lines 224 – 225): "Compared to the global identifiability analysis, the dynamic identifiability analysis evaluates the identifiability of a parameters on a moving window along the BTC".

**R1**: "Additionally, many sentences and paragraphs are not fluent and have poor sentence structure, especially throughout the introduction."
**Authors**: We accurately edited the language, and we re-wrote several paragraphs in the revised manuscript to improve its readability.

**R1**: "The introduction addresses certain open questions / problems but then lists specific questions being answered. It's then confusing as to what questions will and will not be addressed in this paper. I think it would benefit readers to remove the earlier questions mentioned and stick with the ones at the end of the introduction."
**Authors**: We revised the order of the research gaps presented in the manuscript (lines 85 – 119). Also, we introduced another research question (research question 3) that targets the role of the parameter range and the role of the number of parameter sets (third research gap, lines 111 – 119). This choice also satisfies the requirement of the Reviewer 3 for a deeper focus on the role of the parameter range for achieving parameter identifiability in TSM.
The new research question reads (lines 140 – 141): "3) How much does the identifiability of model parameters in random sampling approaches depend to the used parameter range and on the number of parameter sets?"

**R1:** "For the methods section, there is no explanation of how question 2 was answered."
**Authors**: We modified the structure of the method section to clearly indicate that the dynamic identifiability analysis was used to address research question 2 (see above).

**R1**: "What specific sections of the BTC did you investigate and how did you break the BTC up into different sections?"
**Authors**: We defined three sections of the BTC as follows (lines 238 – 241, method section 2.4): "We also specified sections of the BTC as follows: "peak" of the BTC is the section of the BTC corresponding to a neighbourhood interval of three time steps (± ~1 min for E1 and E2, and ± ~15 secs for E3) around the maximum observed concentration; "rising limb" and the "tail" are respectively the BTC sections before and after the peak."

**R1**: "Also, why are the tracer experiments all done during the same season? I acknowledge that the discharge is still varied, but are there other hydrologic processes that change throughout the year not captured by winter tracer studies?"
**Authors**: While the focus of this study is on the identifiability of TSM parameters, we implemented the discussion section by specifying that (lines 609 – 610): "Our modelling outcomes are also in line with the physical understanding of the studied stream reach". Also (lines 614 – 616): "Other modelling and experimental studies also outlined that the stream above the study section is dominated by inflow of groundwater or surface water from wetlands (Antonelli et al. 2020; Glaser et al., 2016, 2020)."

**R1**: "I believe a simple reactive transport study using the different parameter estimates would enhance this study and better highlight the impact and importance of the uncertainty in parameter estimation."
**Authors**: This is a very interesting research aim, but it would need to be a standalone work. However, we implemented discussion section by presenting future developments for solute transport work (lines 640 – 644): "). Our approach offers a promising flexible tool to target parameter identifiability and physical interpretation also in TSM formulation with increasing complexity, such as multiple storage zone models (Choi et al., 2002), or for TSMs considering sorption kinetics (Gooseff et al., 2005) or different residence time distribution laws such as log-normal distribution (Wörman et al., 2002),

exponential plus pumping distribution (Bottacin-Busolin et al., 2011), and power law distribution (Haggerty et al., 2002)."

**R1**: "There is little explanation on whether or not their results make sense in the context of physical hydrologic processes and past studies"
**Authors:** In the revised discussion we clarified how and why our results make sense in the hydrologic context of the study site (lines 609 – 616): "Our modelling outcomes are also in line with the physical understanding of the studied stream reach. The study site is equipped with a dense network of groundwater monitoring wells that showed that the stream channel is almost entirely in gaining conditions for the investigated tracer injections with the groundwater gradients pointing toward the stream channel (Bonanno et al., 2021). This is in line with the obtained TSM transport metrics that indicate a very limited or even a lack of hyporheic exchange. Other modelling and experimental studies also outlined that the stream above the study section is dominated by inflow of groundwater or surface water from wetlands (Antonelli et al. 2020; Glaser et al., 2016, 2020)."

**R1:** "Does it make sense that a variable velocity that is not determined a priori make sense?"
**Author**: We implemented the introduction with a revised section addressing why velocity was considered as a calibration parameter and not determined a-priori (lines 85 – 94): "The parameters describing the advection-dispersion process (streamflow velocity, cross-sectional area of the stream channel, and longitudinal dispersion) are known to be the best identifiable in the TSM (Ward et al., 2017). However, due to the known high interactivity among model parameters, it is generally not recommended to use a fixed value for a rather identifiable parameter, since this strategy may result in a mis-estimation of the other model parameters (Knapp and Kelleher, 2020). Constraining the values of the stream area and longitudinal dispersion proved to have a role on the identifiability of transient storage parameters (Lees et al., 2000; Kelleher et al., 2013; Ward et al., 2017). However, no study so far evaluated the role of flow velocity on the identifiability of model parameters despite the velocity parameter was often considered to be known and thus fixed to equal the velocity of the arrival time of the BTC peak (i.e. $v_{peak}$, Ward et al., 2013; Kelleher et al., 2013; Wlostowski et al., 2017; Ward et al., 2017; Ward et al., 2018)."
The reasoning of why considering the velocity as a calibration parameter was also discussed in the revised section 4.1 with the following (lines 446 – 455): "The practice of setting $v$ equal to $v_{peak}$ in past studies was justified by the notion that $v_{peak}$ can be considered as a reasonable good approximation for the advection process in the stream channel (Ward et al., 2013; Wlostowski et al., 2017) and by the modelling advantage that assuming $v$ equals $v_{peak}$ would reduce model dimensionality (Knapp and Kelleher, 2020). While reducing the number of model parameters is advantageous for reduced model dimensionality, considering $v$ as a calibration parameter is a needed testing strategy in TSMs. This is because measurement uncertainty is inevitable in determining discharge or flow velocity, thus we don't know how big the effect of measurement uncertainty is on model performance, especially considering parameter interaction. Also, constraining the advection-dispersion parameters $A$ and $D$ already proved to affect the identifiability of the other model parameters (Lees et al., 2000; Kelleher et al., 2013; Ward et al., 2017), but no study assessed the role of velocity on parameter identifiability."

**R1**: "Further, the discussion is sometimes a repeat of results rather than a discussion of the implications of the results. Why do the results matter? How does that align with our current understanding?"
**Authors:** We revised accurately the discussion section to avoid repetition of the results. We also indicated how our results align with previous literature (lines 616 – 625): "The observed link of $L_s$, $q_s$, and $T_{sto}$ values with discharge (Figure 7) also suggested that the transient storage at our site became less important in controlling solute transport with increasing discharge. The decrease of $A_{TS}$ and $T_{sto}$ with increasing discharge has been argued to indicate an increase of groundwater gradients toward the stream channel with a consequent decrease in the hyporheic zone at different study sites (Morrice et al., 1997; Fabian et al., 2011). However, the observed groundwater gradients at the study site exclude the presence of significant hyporheic exchange during the three simulated tracer experiments. The observed trend between modelling results with discharge might be interpreted by the fact that, as the discharge

increases, the wetted profile at the study site incorporates into the advective part of the channel the dead zones and the low-flow areas that are responsible for in-stream transient storage at lower flow rates (Zarnetske et al., 2007; Gooseff et a., 2008). This would cause a progressive increase in piston-flow transport and a reduced role of in-stream solute retention with increasing flow and water level in the stream channel."

**R1**: "Also, the paper is not well organized. The intro clearly outlines 3 questions to be addressed. It would make sense to me, if the methods, results, and discussion followed that order. Instead, section 4.1 addresses section 3.2 which addresses question 1. Section 4.3 addresses question 2, but question 2 does not have a methods or results section."

**Authors**: We completely revised the manuscript structure. Now the three research gaps presented in the introduction are targeted by the revised research questions 1, 2 and 3. Every research question is targeted by a specific section in the method, result and discussion sections.

The revised method structure reads:

 2 Study site and methods

  2.1 Study site and tracer data

  2.2 Advection-dispersion equation and Transient Storage Model formulation

  2.3 Random sampling and global identifiability analysis (**Targeting research question 1**)

  2.4 Identifiability analysis on specific sections of the BTC (**Targeting research question 2**)

  2.5 Iterative approach to achieve model identifiability

  2.6 Number of parameter sets, parameter range, and identifiability of model parameters (**Targeting research question 3**)

  2.7 Comparison with an inverse modelling scheme and a Monte Carlo random sampling approach

  2.8 Metrics and hydrologic interpretation of TSM results (**Targeting research question 4**)

The revised result structure reads:

 3 Results

  3.1 ADE parameters

  3.2 TSM parameters (**Targeting research question 1**)

   3.2.1 Identifiability of model parameters when velocity is considered as a calibration parameter

   3.2.2 Identifiability of model parameters when velocity is set equal to $v_{peak}$

  3.3 Dynamic identifiability analysis (**Targeting research question 2**)

   3.3.1 Dynamic identifiability analysis when velocity is considered as a calibration parameter

   3.3.2 Dynamic identifiability analysis when velocity is set equal to $v_{peak}$

  3.4 Role of the used parameter range and the number of parameter sets for the identifiability of model parameters (**Targeting research question 3**)

  3.5 Comparison with OTIS-P and OTIS-MCAT results

  3.6 Variation of transport metrics with increasing identifiability of model parameters (**Targeting research question 4**)

The revised discussion structure reads:

 4 Discussion

  4.1 The role of velocity in random-sampling approaches for TSM (**Targeting research question 1**)

  4.2 Control of model parameters on the rising limb, the peak, and the tail of the BTC (**Targeting research question 2**)

  4.3 On the importance of parameter range, parameter sets, and challenges associated to parameter identifiability in TSM (**Targeting research question 3**)

  4.4 Implications of identifiable TSM parameters for hydrologic interpretation of modelling results (**Targeting research question 4**).

**R1**: "Finally, the discussion and conclusion are missing an explanation of how to go forward from these findings. Line 535 states that process interpretation and parameter evaluation should be used with caution, but how do we address that? What are the next steps?"

**Authors:** We revised the discussion to include next-steps and take-home messages for each investigated discussion section and research question.

Future steps for section 4.1 (lines 456 – 465): "Our results provide valuable guidance for future studies addressing parameter identifiability in TSM. Specifically, our results support the current praxis of considering velocity fixed and equal to $v_{peak}$, especially when research aims at evaluating the distribution of "behavioural" parameter sets in TSMs (i.e. parameter sets satisfying certain performance thresholds). This is due to the fact that using velocity as calibration parameter leads to the same parameter identifiability compared to the case when velocity is considered fixed (Figure 2, 3, Table 2). Yet, setting velocity equal to $v_{peak}$, requires a considerably lower amount of computational power due to the lower degrees of freedom of the TSM. However, when research aims to evaluate the control of the model parameters on the shape of the BTC, our results suggest that increasing the model complexity by considering velocity as a varying model parameter can offer more detailed insights into the role of advection-dispersion processes on the tail of the BTC and of the transient storage parameters on the rising limb and peak of the BTC (Figure 4, 5)."

Future steps for section 4.2 (lines 527 – 531): "This study offers significant insights in understanding which model parameter influence the shape of the BTC, suggesting that only behavioural parameter sets should be considered in models aiming to understand the control of model parameters on the rising limb, peak, and tail of the BTC. Future work should address the interaction of model parameters on controlling different sections of the BTC for more complex model formulations (e.g. TSM with two or several transient storage zones, Choi et al., 2002; Bottacin-Busolin et al., 2011)."

Section 4.3 (lines 566 – 571): "The adopted iterative approach allowed to achieve parameter identifiability and to obtain physically realistic transport metrics. However, this approach is based on the specific objective function used (*RMSE*) and on the subjective thresholds to control the refinement of the parameter range for successive iterations (top 10% results for the global identifiability analysis, and information content > 0.66 for the dynamic identifiability analysis). Future work should explore the impact of the selection of the thresholds and of different objective functions on the physical realism of the modelling results and of the identifiability of the parameters."

Section 4.4 (lines 637 – 645): "Our results also open developments for research seeking to increase the physical realism of the TSM and its results. Increased model complexity is both associated with a better analytical fitting to the observed BTC, but also with an increased degree of freedom of the model with a consequent reduction of parameter identifiability (Knapp and Kelleher, 2020). Our approach offers a promising flexible tool to target parameter identifiability and physical interpretation also in TSM formulation with increasing complexity, such as multiple storage zone models (Choi et al., 2002), or for TSMs considering sorption kinetics (Gooseff et al., 2005) or different residence time distribution laws such as log-normal distribution (Wörman et al., 2002), exponential plus pumping distribution (Bottacin-Busolin et al., 2011), and power law distribution (Haggerty et al., 2002)."

**R1**: "The study is novel and can be beneficial for those studying hydrologic exchange in river networks and using tracer experiments, but this paper would greatly benefit from some restructuring and re-writing for improved readability and explanation of methods and results."

**Authors**: We thank R1 for the time spent in the exhaustive revision and for highlighting the weakness of the manuscript.

**R1:** "Specific things to address are listed below:
First paragraph of intro is confusing. You mention modeling, then experiments, then talk about issues with modeling again. I think it's worth re-organizing in the following way:
• Understanding transport along river networks is important
• One way is with experiments
• However, current models to describe the processes seen from experiments have contradictory results

• So, it is unknown how informative modeling is"

**Authors**: We fully agree and we thank R1 for the proposed structure, we revised the first paragraph of the introduction as follows (lines 39 – 54): "It is of crucial importance to understand how nutrients, solutes, and pollutants are transported in streams, since this process can drastically affect stream water quality along river networks (Smith, 2005; Krause et al., 2011; Rathfelder, 2016). A widely used technique to capture and study the processes controlling water transport downstream is via in-stream tracer injections. The measurement of the concentration over time of a tracer released in an upstream section (i.e. the breakthrough curve, BTC) reflects stream discharge (Beven et al., 1979; Butterworth et al., 2000) and longitudinal tracer advection and dispersion (Gooseff et al., 2008). A milestone in the study of solute transport was that in-stream solutes and water are exchanged with slowly-moving channel waters, the dead zones (Hays, 1966), and with the saturated area that is physically influenced by water and solutes exchange between the stream channel and the adjacent groundwater (i.e., the hyporheic zone, Triska et al., 1989; White, 1993; Cardenas and Wilson, 2007). This hydrologic exchange results in a skewed non-Fickian BTC with a pronounced tail, which makes the advection-dispersion equation (ADE) unable to correctly describe the observed tracer transport in stream channels (Bencala and Walters, 1983; Castro and Hornberger, 1991). Despite the large amount of studies, the results of TSM offer numerous contradictory model interpretation (Ward and Packman, 2019), and model parameters are often non-identifiable, meaning that several parameter combination return same model performances (Ward et al., 2017). These outcomes raise the question about how informative such modelling results are (Knapp and Kelleher, 2020)."

**R1**: "Lines 67-73 is jargon heavy and phrasing is clunky."

**Authors**: We reformulated the indicated sections as reported (lines 69 – 75): "Also, calibrated parameter obtained via inverse modelling approach are not necessarily meaningful, as non-identifiable parameters can provide a good inverse model fit (Kelleher et al., 2019). These modelling uncertainties have led to a progressive abandonment of the search for a single best set of parameters and advocated the identification of "behavioural" parameter populations (i.e. parameter sets satisfying certain performance thresholds) via random sampling approaches in transient storage modelling (Wlostowski et al., 2013; Ward et al., 2017; Ward et al., 2018; Kelleher et al., 2019; Rathore et al., 2021)."

**R1**: "Sentence 103-104, repetitive".

**Authors**: We removed the indicated sentence.

**R1:** "The whole paragraph starting at 113 is confusing."

**Authors**: We revised the indicated paragraph. It now reads (lines 96 – 111): "A second cause for non-identifiable model parameters relates to the selected approach for addressing parameter identifiability. The identifiability analysis used in most studies is based on the Generalized Likelihood Uncertainty Estimation that assesses parameter certainty by evaluating model performance on the entire BTC (GLUE, Beven and Binley, 1992; Camacho and González, 2008; Kelleher et al., 2013; Ward et al., 2017; Kelleher et al., 2019). However, such global identifiability analysis is unable to assess if a certain parameter is more or less identifiable in certain sections of the BTC (Wagener et al., 2002; Wagener et al., 2003; Wagener and Kollat, 2007). This information is particularly important for BTC modelling, since advection-dispersion parameters are physically responsible for the bulk solute transport in the stream and they are therefore expected to act on the rising limb and peak of the BTC (Gooseff et al., 2008). Contrary, the parameters describing the exchange between the stream channel and the transient storage zone are responsible for delaying solute transport compared to the advective-dispersive transport, most likely acting on the falling limb and tail of the BTC (Runkel, 2002). By investigating parameter identifiability across the entire BTC, global identifiability analysis is unable to capture an increase in parameter identifiability towards the tail of the BTC. However, studies addressing the identifiability of model parameters in different sections of the BTC reported an increased identifiability for transient storage parameters on the tail of the BTC (Wagener et al., 2002; Scott et al., 2003; Wlostowski et al., 2013; Kelleher et al., 2013)"

**R1**: "Also unclear on if the results make sense in terms of physical processes and our current understanding or if they're spurious relationships (ie non-id A_TS coupled with ID alpha, and when v=vpeak the reverse of that)"

**Authors**: In the revised discussion, we described why our TSM results are consistent with the general understanding of the water movement in the investigated stream reach (lines 609 – 614): "Our modelling outcomes are also in line with the physical understanding of the studied stream reach. The study site is equipped with a dense network of groundwater monitoring wells that showed that the stream channel is almost entirely in gaining conditions for the investigated tracer injections with the groundwater gradients pointing toward the stream channel (Bonanno et al., 2021). This is in line with the obtained TSM transport metrics that indicate a very limited or even a lack of hyporheic exchange."

**R1**: "Should review and cite Rathore et al., 2021, "On the Reliability of Parameter Inferences in a Multiscale Model for Transport in Stream Corridors". I believe this paper aligns with your study and might provide some further insight."

**Authors**: We cited Rathore et al., 2021 to support our introduction and our discussions.

**R1:** "No explanation on global identifiability vs dynamic identifiability"

**Authors**: We defined global identifiability analysis and dynamic identifiability analysis in the revised sections 2.3 and 2.4, respectively.

**R1**: "Should reread Gooseff et al., 2005 and update the sentence starting at line 475 as the statement is incorrect as it stands."

**Authors**: We removed the cited reference in the indicated section.

**R1:** "Figure 8, what is DYNIA analysis? This is not previously explained."

**Authors**: DYNIA is the "dynamic identifiability analysis". We replaced the acronym.

Response to Reviewer 2

**R2:** "The text content is generally well written but the theme's complexity and the structuring makes honestly the reading rather difficult, which can force the reader going back and forth the chapters to follow a storyline. I found very logical how the text was structured in the 'Study site and Methods' section, and I would really try to stick to this same structure when presenting 'Results', in the 'Discussion' section and even in the 'Conclusion' paragraph. My personal taste would be:

    a.1) ADE parameters

    a.2) TSM parameters (i.e. hydrologic exchange parameters)

        a.2.1. Identifiability of TSM parameters when v = vvariable

        a.2.2. Identifiability of TSM parameters when v = vpeak

    a.3) Model iterations

        a.3.1. TSM first iterations

        a.3.2. TSM last iterations

        a.3.3. DYNIA

        a.3.4. Comparison with inverse modeling results (OTIS-P)

        a.3.5. Comparison with random sampling approaches (OTIS-MCAT)

    a.4) Metrics and hydrologic interpretations of model results

According to this numbering, the 'Results' section starts with the a.3.1 jumps to a.3.3, then back to a.3.2, etc. The 'Discussion' section starts with a.4, then goes back to a.3.1 and a.3.2, follows with a.3.4, etc. This can make a difference between a quick and effective reading, and a tedious reading."

**Authors:** We thank the Reviewer 2 (R2) for the time and the effort spent in reviewing our manuscript. To satisfy the requirement of improvement, together with the requirement of R1 (that the manuscript structure has to mirror the order of the research gaps) and of R3 (that we need to emphasize the role of the parameter range to achieve parameter identifiability), we revised the entire manuscript as follows. Revised research questions read (lines 136 – 144):

1)     How does the identifiability of model parameters change in the random sampling of TSM when velocity is considered as a calibration parameter and when it is assumed fixed and equal to $v_{peak}$?

2)     Does the identifiability analysis on specific sections of the BTC reduce the parameter non-identifiability in random sampling of TSM?

3)     How much does the identifiability of model parameters in random sampling approaches depend to the used parameter range and on the number of parameter sets?

With the outcomes of these questions we will address:

4)     How does the hydrologic interpretation of TSM results vary when model parameters are identifiable and when they are not?"

The revised method structure reads:

    2 Study site and methods

        2.1 Study site and tracer data

        2.2 Advection-dispersion equation and Transient Storage Model formulation

        2.3 Random sampling and global identifiability analysis (**Targeting research question 1**)

        2.4 Identifiability analysis on specific sections of the BTC (**Targeting research question 2**)

        2.5 Iterative approach to achieve model identifiability

        2.6 Number of parameter sets, parameter range, and identifiability of model parameters (**Targeting research question 3**)

        2.7 Comparison with an inverse modelling scheme and a Monte Carlo random sampling approach

        2.8 Metrics and hydrologic interpretation of TSM results (**Targeting research question 4**)

The revised result structure reads:

    3 Results

        3.1 ADE parameters

        3.2 TSM parameters (**Targeting research question 1**)

3.2.1 Identifiability of model parameters when velocity is considered as a calibration parameter

3.2.2 Identifiability of model parameters when velocity is set equal to $v_{peak}$

3.3 Dynamic identifiability analysis (**Targeting research question 2**)

3.3.1 Dynamic identifiability analysis when velocity is considered as a calibration parameter

3.3.2 Dynamic identifiability analysis when velocity is set equal to $v_{peak}$

3.4 Role of the used parameter range and the number of parameter sets for the identifiability of model parameters (**Targeting research question 3**)

3.5 Comparison with OTIS-P and OTIS-MCAT results

3.6 Variation of transport metrics with increasing identifiability of model parameters (**Targeting research question 4**)

The revised discussion structure reads:

4 Discussion

4.1 The role of velocity in random-sampling approaches for TSM (**Targeting research question 1**)

4.2 Control of model parameters on the rising limb, the peak, and the tail of the BTC (**Targeting research question 2**)

4.3 On the importance of parameter range, parameter sets, and challenges associated to parameter identifiability in TSM (**Targeting research question 3**)

4.4 Implications of identifiable TSM parameters for hydrologic interpretation of modelling results (**Targeting research question 4**).

**R2:** "I would like to draw the attention about the use of the term 'parameters' in the manuscript. I personally have no experience with the identifiability of parameters in a TS model and it can bias my understanding of the term. Thus, this might be probably my wrong perception.

As a hydrologist, my understanding of parameters involved in the transport of solutes in a stream/river can be stream discharge, components of flow velocity, flow turbulence, grain size of streambed sediments, groundwater-stream (GW-SW) water exchange fluxes, stream channel topography, existence/absence of riparian vegetation, etc. In the manuscript, these parameters are treated as such, but then it introduces the term TSM parameters and parameter sets. In this manuscript's TSM model, each iteration simulates 115,000 parameter sets. An unexperienced reader would tend to think that > 100,000 parameters as the ones I mentioned above are involved in the transport of solutes in the stream. This should be obviously clarified."

**Authors:** We defined "parameter sets" in the revised method section (lines 195 – 197): "A single combination of model parameters ($A$, $v$, and $D$ for ADE and $A$, $v$, $D$, $A_{TS}$, and $\alpha$ for TSM) obtained from the random sampling approach is herein referred to as "parameter set".

**R2**: "The manuscript's title states 'to reduce the uncertainty' but I have missed more uncertainty analysis throughout the text."

**Authors**: The term "uncertainty" was used as a synonym for "non-identifiability". We uniformed the terminology in the revised manuscript and used only "identifiability". The title of the manuscript was accordingly modified in "Exploring tracer information in a small stream to improve parameter identifiability and enhance the process interpretation in transient storage models".

**R2**: "The abstract is concise and clearly written. How to fill the research gap is properly addressed.

1. Line 14: I would consider adding 'adjacent groundwater bodies' as a exchanging agent.
2. Line 24: … TSM parameters, respectively. The severe differences…
3. Line 26: … at the study site → The article 'the' makes reference to a determined object, but the study site has not yet been introduced. Consider changing 'the' for 'our' or 'a study site in western Luxembourg', for example."

**Authors**: We thank R2 for the suggestions. We have implemented these changes in the revised manuscript.

**R2**: "The introduction chapter needs to be revised in order to be fully understandable. The content is good, but an improved organization can elevate it."

**Authors**: We revised the introduction. We have modified the first paragraph following the suggested structure proposed by R1, we removed the paragraph from lines from 83 to 92 (old manuscript version), we reduced the verbosity by re-writing several sections, and we re-organized the structure of the three research gaps in order to mirror the order of the investigated research questions (lines 80 – 115).

**R2**: "Tend to list consecutive citations in chronological order (e.g. lines 72-73, 78-79, etc.)."

**Authors**: We revised the order of all the citations by listing them in chronological order.

**R2**: "Line 49: Many readers can disagree with such a definition of 'hyporheic zone'. I don't agree with the idea of a fully saturated thickness. I usually consider the definition given by Cardenas & Wilson, 2007 (Cardenas, M. B., & Wilson, J. L. (2007). Exchange across a sediment–water interface with ambient groundwater discharge. Journal of Hydrology, 346, 69 – 80), as a good approach."

**Authors**: We modified the definition of the hyporheic zone as follows (lines 46 – 48): "the saturated area that is physically influenced by water and solutes exchange between the stream channel and the adjacent groundwater (i.e., the hyporheic zone, Triska et al., 1989; White, 1993; Cardenas and Wilson, 2007)."

**R2**: "Lines 79-82: This reads like a Research gap, and it is well stated, but it should fit at a later stage, before enumerating hypothesis and research questions."

**Authors**: We moved the indicated section, together with the used hypothesis (see later comment), right before the research questions (lines 129 – 133): "Despite the increasing need for achieving parameter identifiability in TSMs, only few studies have explored the reliability of results obtained from inverse modelling, and model interpretation is often based on a single set of parameters without testing their robustness (Knapp and Kelleher, 2020). We hypothesise that addressing the identifiability of model parameters in different sections of the BTC is key in increasing the identifiability of the parameters describing solute retention in streams."

**R2**: "Lines 103 and 113: I would not change paragraph in this sentence."

**Authors**: We have listed the first, second and third research gaps in the same paragraph (lines 85 – 119).

**R2**: "Line 105: 'stream velocity' → The stream itself does not move. Better using 'streamflow velocity'."

**Authors**: We have unified the terminology and used "streamflow velocity" in the entire manuscript.

**R2**: "Line 128: I would state the hypotheses in a new paragraph."

**Authors**: We have moved the hypothesis in a new paragraph together with the research gap (see above, lines 129 – 133).

**R2**: "Line 152: Study site and 'data' → Better specify which data."

**Authors**: We clarified. The Section 2.1 now reads (line 142): "Study site and tracer data".

**R2**: "Study site and data → How long is the studied stream reach?"

**Authors**: The stream reach is indicated at the line 159: "Electric conductivity (EC) was measured via a portable conductivity meter (WTW) 55 m downstream of the injection point."

**R2**: "Lines 161-162: How was Q calculated? (± Analytical Error?)"

**Authors**: The revised method section reads (lines 163 – 165): "Discharge was calculated for every slug injection via the dilution gauging method using the Cl$^-$ concentration obtained for each BTC (Beven et al., 1979; Butterworth et al., 2000)". Since we did not perform repeated measurements, we could not indicate the analytical error.

**R2**: "Line 164: How was EC calibrated according to NaCl concentrations? Was EC temperature compensated? (± Analytical Error?)"

**Authors**: We clarified as follows (lines 158 – 163): "Electric conductivity (EC) was measured via a portable conductivity meter (WTW) 55 m downstream of the injection point. Automatic compensation of stream temperature occurred (nLF, according to EN 27 888). EC-Cl$^-$ conversion was obtained using a known-volume sample of stream water taken before tracer injection at the measurement location and adding known quantities of a solution with a known concentration of Na-Cl. Conversion into Cl-concentration was obtained via an EC-Cl- regression line (R2 = 0.9999)."

**R2**: "Was there any difference in background EC between the injection and the measurement locations? Or was it assumed to be equal? Or is it one of the testing parameters? These aspects are actually interesting."

**Authors**: The background concentration at the injection point is slightly lower from the background concentration at the measurement location. We account for this difference during the EC-Cl conversion by sampling the stream water at the measurement location. We clarified this detail (see above, lines 158 – 163).

**R2**: "Line 181 (Equation 3): Why the second term of the second equation is negative? If Cs < C, the concentration of a certain solute in stream tends to get diluted. But according to eq. 3, $\partial C/\partial t$ would become larger"

**Authors**: We double checked the equation 3, and we confirm that the second equation is negative.

**R2**: "2.2: How the accuracy of the ADE and TSM results are assessed?"

**Authors**: We specified that (lines 184 – 185): "The performances of both ADE and TSM results are evaluated using the Root Mean Squared Error objective function (*RMSE*)".

**R2**: "Line 207: I assume CDF is the Cumulative Distribution Function, but it has not been defined as such yet."

**Authors**: We implemented this clarification in line 207 of the revised 2.3 section.

**R2**: "2.3: How is the uncertainty of the model iterations' results, DYNIA's results and the comparison with OTIS-P and OTIS-MCAT assessed?"

**Authors**: We uniformed terminology in the manuscript, and we used only the term "identifiability" (see above). Identifiability of TSM results was studied via (lines 201 – 203): "parameter vs *RMSE* plots (Wagener et al., 2003), parameter distribution plots (Ward et al., 2017), regional sensitivity analysis (Wagener and Kollat, 2007; Kelleher et al., 2019), and parameter distribution plots (Wagener et al., 2002; Ward et al., 2017)." While "OTIS-P model estimates the best-fitting model parameter values and their identifiability via the 95% confidence interval" (lines 286 – 287). And, for the dynamic identifiability analysis (lines 230 – 238): "The information content is expressed as one minus the width of the 90% confidence interval over the entire parameter range (Wagener et al., 2002). A wide 90% confidence interval indicates that various parameter values are associated to equally good performances resulting in low information content. Conversely, narrow 90% confidence intervals and corresponding high information content values suggest that the best-performing parameters are contained in a relatively narrow range compared to the feasible range. To evaluate the degree of identifiability of a certain parameter on specific sections of the BTC, we grouped parameter identifiability in three categories: highly identifiable (information content ≥ 0.66), moderately identifiable (0.33 ≤ information content < 0.66), and poorly identifiable (information content < 0.33)."

**R2**: "Line 270 (Equation 7): Introduce a tab in the equation's label, so it can be in line with the rest of the equations."

**Authors**: We inserted a tab prior to "Eq. 7".

**R2**: "2.4: I assume that including FMED reduced the systematic error, but is it quantifiable?"

**Authors**: $F_{MED}$ is a non-dimensional metric used to interpret TSM results in comparison to other reaches or discharge conditions. Unfortunately, it has no role on parameter identifiability and model error.

**R2**: "This chapter is properly written but the sub-chapters could be re-organized to build-up a smoother storyline. See the last part of General comment (a)."

**Authors**: We revised the structure of the results section to mirror the order of the research questions and the order of the revised method sections (see above).

**R2**: "Lines 279-282: This reads more like in the 'Methods' section, right?"

**Authors**: We removed these lines.

**R2**: "Why was vpeak and no other velocity chosen for the fixed velocity scenario? Why not median velocity which could be more a representative velocity value? What does vpeak mean?
- Peak velocity during tracer experiments?
- Mean velocity during the experiment E3 which had the largest discharge?
- Peak velocity during the entire monitoring period?"

**Authors**: We clarified that (lines 267 – 271): "The same approach (Figure 1) was used also in the case where v was assumed fixed and equal to $v_{peak} = L/t_{peak}$, where $t_{peak}$ is the arrival time of the concentration peak. This choice was motivated by the fact that $v_{peak}$ is commonly adopted as a value for velocity in many transient storage studies (Ward et al., 2013; Kelleher et al., 2013; Wlostowski et al., 2017; Ward et al., 2017; Ward et al., 2018)."

**R2**: "Line 292: Figure 3c-f →Is Figure 3f informative in the tailing of the BTC?"

**Authors**: Yes, it is in our view. We clarified that (lines 240 – 241): "rising limb" and the "tail" are respectively the BTC sections before and after the peak of the BTC.

**R2**: "Line 297: Add a comma after > 0.05),"

**Authors**: We implemented this change (line 328).

**R2**: "8.8 l/s for E3 → I would not consider this discharge as a low qs, since you state that the exchanging flow is > 1/3 of the total Q of the stream."

**Authors**: We removed subjective terms as "low", "high", "long" from section 3.6 ("Variation of transport metrics with increasing identifiability of TSM parameters") to avoid unprecise writing.

**R2**: "Line 343: 15 hrs for E3 → I would not consider this as a long residence time. The flux velocity is roughly 4.6 m/hr, which according to me can be slow for dead zones, but not for hyporheic flux, or GW-SW exchange."

**Authors**: We removed subjective terms (see above).

**R2**: "Lines 345-348: are these simulations physically possible at all, knowing the actual stream discharge? As for Tsto, do 0.8 (in E1) and 3.3 (in E3) m/s make sense?"

**Authors**: We reported explanation for the physical realism of these results in the HESS discussion. However, we did not examine the physical realism of these results in the manuscript since they refer to non-identifiable TSM parameters.

**R2**: "Lines 353-354: Again… double check the applicability of these results into the actual flow regime."

**Authors**: The results are physically possible (see HESS discussions) and also in line with the physical understanding of the investigated stream reach. In the revised discussion section, we implemented this in lines 610 – 616: "The study site is equipped with a dense network of groundwater monitoring wells that showed that the stream channel is almost entirely in gaining conditions for the investigated tracer injections with the groundwater gradients pointing toward the stream channel (Bonanno et al., 2021). This is in line with the obtained TSM transport metrics that indicate a very limited or even a lack of hyporheic exchange. Other modelling and experimental studies also outlined that the stream above the study section is dominated by inflow of groundwater or surface water from wetlands (Antonelli et al. 2020; Glaser et al., 2016, 2020)."

**R2**: "Comments 26, 27 and 28: Since you are doing a comparison between methods, rather than using terms like 'high' or 'low', it is better to use relative terms such as 'higher' or 'lower', or even 'distinct'."

**Authors**: We removed subjective terms (see above).

**R2**: "Line 349: 'whether' does not sound appropriate in this context. I would rather use 'regardless'?"
**Authors**: We implemented this change (line 410).

**R2**: "Same as in the previous chapter, the sub-chapters could be re-organized to build-up a smoother storyline. See the last part of General comment (a). Aren't 4.1 and 4.3 more related to each other, and 4.2 a unique sub-chapter. I think starting with 4.2 and then following with 4.1 and 4.3 would improve the chapter's sequence."
**Authors**: We revised the entire structure of the manuscript to meet these changes (see above).

**R2**: "4.1: Are the uncertainties specified?"
**Authors**: The uncertainties of the transport metrics are reported as boxplot limits in Figure 7.

**R2**: "Lines 402 and 406-407: Aren't these contradictory statements? Identifiability is contradictory under the same scenario (in both cases v = vpeak)."
**Authors:** We clarified these sentences as follows (line 437 – 440): "This was particularly evident when $v$ was considered as a calibration parameter, and the non-identifiability of $A_{TS}$ was coupled with identifiable $v$ and $\alpha$ (Figure 2, green and yellow dots). On the contrary, $A_{TS}$ was found to be identifiable and $\alpha$ to be non-identifiable when $v$ was fixed equal to $v_{peak}$ (Figure 3, yellow dots)."

**R2**: "Line 438: 100,000 instead of 100'000."
**Authors**: We modified accordingly (line 545).

**R2**: "Line 446-447: Seems like a very generalist closing statement."
**Authors**: We removed this closing statement. We discussed the role of number of parameter sets on identifiability of TSM in a new section (Section 4.3 in the revised manuscript).

**R2**: "Lines 460-462: Aren't these contradictory statements? The assumption v = vpeak might not be representative of the advection role, but can encompass the effect of advection in the entire BTC."
**Authors**: We reformulated these sentences. Now they read (lines 471 – 476): "The assumption used in previous work of streamflow velocity equalling $v_{peak}$ implies that $v_{peak}$ should encompass the effect of advection on the entire BTC or at least in the rising limb and peak of the BTC (Ward et al., 2013; Kelleher et al., 2013; Wlostowski et al., 2017; Ward 2018). However, when $v$ was used as a calibration parameter, our results showed that $v$ is one of the least meaningful parameters for simulating the peak of the BTC at low discharge (Figure 4k, i), while higher information content for $v$ is obtained at higher discharge rates for values larger than $v_{peak}$ at the peak of the BTC (dynamic identifiability plots not shown)."

**R2**: "Lines 480-503: Figure 8 is actually a very good review from other authors' findings. I quite like it. The piece of text relating to this figure is actually an important part of your discussion, but again it reads like a succession of sentences with vague organization, and it forces the reader to continuously go back and forth between the text and the figure. I propose the following. Instead of travelling through all the cited authors and comparing your results with their results, why can't it be presented in concordance with the different BTC's features, e.g.:
- Rising Limb
- Peak
- Falling Limb
- Tail
One can travel along the BT curve and compare your data with literature data. The reading would be more graphical and intuitive, and it could probably help to safe some word spacing.
By the way, it is interesting to know whether there is a method or a simple threshold value to distinguish between the end of the falling limb and the beginning of the tail?"
**Authors**: We defined the sections of the BTC in the revised method section (lines 239 – 241): ""peak" of the BTC is the section of the BTC corresponding to a neighbourhood interval of three time steps ($\pm$

~1 min for E1 and E2, and ± ~15 secs for E3) around the maximum observed concentration; "rising limb" and the "tail" are respectively the BTC sections before and after the peak."

Also, we have reformulated the content of the paragraph 4.2 (Control of model parameters on the rising limb, the peak, and the tail of the BTC) to compare the previous findings with ours, by travelling through the BTC sections from the rising limb to the tail of the BTC. The revised section (lines 489 – 500) reads: "Several studies addressed how different model parameters affect the shape of the BTC and showed partly similar but also contrasting outcomes to our findings (Figure 8g-l, Wagner and Harvey, 1997; Wagener et al., 2002; Scott et al., 2003; Wlostowski et al., 2013; Kelleher et al., 2013). Past studies found that the rising limb of the BTC was controlled by the stream channel area $A$ alone (Wagener et al., 2002), by the combination of $A$ and the longitudinal dispersion coefficient $D$ (Wagner and Harvey, 1997; Wlostowski et al., 2013; Kelleher et al., 2013), or by $A$, $D$, and $A_{TS}$ (Scott et al., 2003). The peak of the BTC was found to be controlled by advection-dispersion parameters in most past TSM applications (Wagener et al., 2002; Wlostowski et al., 2013; Scott et al., 2003; Kelleher et al., 2013). However, Wagner and Harvey (1997) reported a non-negligible role of the transient storage parameters $\alpha$ and $A_{TS}$ in controlling the arrival time of the peak concentration (Figure 8g). Eventually, while the majority of the studies found the transient storage parameters $\alpha$ and $A_{TS}$ to control the tail of the BTC (Wagner and Harvey, 1997; Scott et al., 2003; Wlostowski et al., 2013), results reported by Wagener et al., (2002) and by Kelleher et al. (2013) highlight the role of the stream channel area $A$ on controlling a large portion of the tail of the BTC."

Differences between past studies and our results have then been discusses in the following paragraphs (lines 501 – 526) which report the revised discussion of the section 4.3 of the previous version of the manuscript.

**R2**: "Lines 502-503: '… Kelleher et al., 2013 also indicate…" → whereas it has a strong influence based on your results, right?"

**Authors**: We revised the indicated section as follows (lines 517 – 521): "Compared to our results, the different role of the model parameters on controlling the shape of the BTC in previous studies (e.g. Kelleher et al., 2013) could be driven by the different approach used for evaluating the sensitivity (i.e. Sobol' sensitivity analysis). However, our results suggest that the number of parameter sets (42,000) selected by Kelleher et al. (2013) might not have been sufficient to obtain identifiability of the model parameters with the rather wide parameter range chosen for their Monte Carlo sampling (Table 1)."

**R2**: "Lines 503-504: 'Different sensitivity…' → This reads like the staring of the next paragraph."
**Authors**: The differences between our results and results from Kelleher et al. (2013) have been revised and are indicated in a unique paragraph (lines 517 – 526).

**R2**: "The conclusion reads well, but again, I would try to expose the ideas trying to follow the same sequence of findings shown in the results."
**Authors**: We revised completely the content of the conclusion section (lines 645 – 670).

**R2**: "Line 523: '… that the BTC…' -→ Revise wording."
**Authors**: We revised entirely the conclusion. The indicated section is no longer present.

**R2**: "Line 739 (Table 2): $v = vpeak$ → Try to keep the same notation as in the text (i.e. Italic and subscript)."
**Authors**: We uniformed the fonts of $v_{peak}$ and others.

**R2**: "Line 777 (Figure 2): The meaning of colored dots for (m-o) plots not specified."
**Authors**: We clarified that the indicated colours refer to plots from "k" to "o".

**R2:** "Line 778 (Figure 2): Try to keep the same notation as in the text (i.e. Italic and subscript)."
**Authors**: We uniformed the fonts of $v_{peak}$ and others.

**R2**: "Figure 3: Here you show the BTC in E1 when the streamflow is smaller and the identifiability is probably less dynamic. I would have liked to see differences in dynamism between the different tracer

experiments. This does not mean that you need to re-do the figures for other tracer experiments, but probably comment a bit on it in the text."

**Authors**: After describing results for E1, the revised result section includes now information about the dynamic identifiability of $\alpha$ and $A_{TS}$ for E2 and E3.

Lines 356 – 359 (section 3.3.1) read: "Results from E2 and E3 showed that $\alpha$ and $A_{TS}$ were highly identifiable (information content > 0.66) for smaller sections of the tail of the BTC when the experiments were conducted at higher discharge stages (information content of $A_{TS}$ > 0.66 for 51% of the tail of the BTC for E1, for 23% for E2, and for 19% for E3, results not shown)."

Lines 363 – 364 (section 3.3.1) read: "Dynamic identifiability analysis after the last TSM iteration for E2 and E3 showed comparable results (not shown)."

Lines 372 – 374 (section 3.3.2) read: "The dynamic identifiability analysis for the BTC of E2 and E3 yielded similar results, with narrow confidence intervals for both $A_{TS}$ and $\alpha$ on the tail of the BTC and no clear trend between information content and discharge (results not shown)."

Lines 377 – 379 (section 3.3.2) read: "For E2 and E3, results after the last TSM iteration showed lower information content of $A_{TS}$ on the tail of BTC for increasing discharge stages compared to E1, while the information content of $\alpha$ was above 0.33 on the entire BTC (results not shown)."

**R2**: "Figure 4: I would probably combine Figs. 3 and 4 in the same figure. Both are showing exactly the same info (Experiment E1 and v = vvariable) but only for different iteration states, and can be potentially confusing."

**Authors**: We adopted the indicated change, and we combined old Figure 3 and 4 in the new Figure 4. For consistency in terms of result representation, the revised Figure 5 now reports dynamic identifiability analysis of TSM parameters for E1 when $v$ was set equal to $v_{peak}$ for the first and the last TSM iteration, in the same way as for the revised Figure 4.

**R2**: "Figure 7: In the Y-axis of plots g-I and j-l, please use similar notation. Use either 0.01 or 10-2."

**Authors**: We unified the numerical notion and used always 0.01 instead of $10^{-2}$. We have always used $10^{-3}$ notion for number lower than $10^{-3}$.

**R2**: "Lines 866-867 (Figure 7): 'm','n','o' seem to be leftovers."

**Authors**: We removed these letters.

**R2**: "Line 868 (Figure 7): '… and equal to vpeak, respectively'."

**Authors**: We adopted this change (line 1036).

**R2**: "Figure 8: Plots g-l are presented first and explained later. They can both combine in the same piece of text."

**Authors**: We both presented and explained the plots (g)-(l) in the same part of the figure description (lines 1050 – 1063).

Response to Reviewer 3

**R3:** "The manuscript highlights an important aspect of inverse modeling based on stream tracer test – the uncertainty and identifiability of the estimated model parameters and the connected metrics used to assess solute transport processes in streams. Although I think that the text is generally well written, the structure of the paper (e.g. order of sections) could be streamlined and improved for clarity. As it is right now, I found myself going back and forth between the different sections in order to understand the iterative and rather complex structure. Overall, I appreciate the approach and the general topic, but I believe the manuscript would benefit from addressing a number of issues highlighted below."

**Authors**: We thank Reviewer 3 (R3) for the time and the effort spent in reviewing our manuscript and for the constructive and supportive feedback. As suggested by R3, we revised the structure of the manuscript and, consistent with comments from R1 and R2, we modified the structure of the manuscript as follows.

The revised method structure reads:

2 Study site and methods

    2.1 Study site and tracer data

    2.2 Advection-dispersion equation and Transient Storage Model formulation

    2.3 Random sampling and global identifiability analysis (**Targeting research question 1**)

    2.4 Identifiability analysis on specific sections of the BTC (**Targeting research question 2**)

    2.5 Iterative approach to achieve model identifiability

    2.6 Number of parameter sets, parameter range, and identifiability of model parameters (**Targeting research question 3**)

    2.7 Comparison with an inverse modelling scheme and a Monte Carlo random sampling approach

    2.8 Metrics and hydrologic interpretation of TSM results (**Targeting research question 4**)

The revised result structure reads:

3 Results

    3.1 ADE parameters

    3.2 TSM parameters (**Targeting research question 1**)

        3.2.1 Identifiability of model parameters when velocity is considered as a calibration parameter

        3.2.2 Identifiability of model parameters when velocity is set equal to $v_{peak}$

    3.3 Dynamic identifiability analysis (**Targeting research question 2**)

        3.3.1 Dynamic identifiability analysis when velocity is considered as a calibration parameter

        3.3.2 Dynamic identifiability analysis when velocity is set equal to $v_{peak}$

    3.4 Role of the used parameter range and the number of parameter sets for the identifiability of model parameters (**Targeting research question 3**)

    3.5 Comparison with OTIS-P and OTIS-MCAT results

    3.6 Variation of transport metrics with increasing identifiability of model parameters (**Targeting research question 4**)

The revised discussion structure reads:

4 Discussion

    4.1 The role of velocity in random-sampling approaches for TSM (**Targeting research question 1**)

    4.2 Control of model parameters on the rising limb, the peak, and the tail of the BTC (**Targeting research question 2**)

    4.3 On the importance of parameter range, parameter sets, and challenges associated to parameter identifiability in TSM (**Targeting research question 3**)

    4.4 Implications of identifiable TSM parameters for hydrologic interpretation of modelling results (**Targeting research question 4**).

**Major comments:**

**R3**: "I think that the terms "identifiability" and "sensitivity" needs to be defined early on in the introduction so that there is no doubt what is meant by these key terms in the context of the study. As it is now the words are used already from the start of the paper, but the definitions are "hidden" in Appendix A. I suggest moving the definitions (lines 902-905) and placing them in the main paper where they are first introduced."

**Authors**: In the revised manuscript, we introduced the term identifiability in the abstract (lines 14 – 17): "Transient storage models (TSMs) are a powerful tool for testing hypotheses related to solute transport in streams. However, model parameters often do not show a univocal increase of model performances in a certain parameter range (i.e. are non-identifiable) leading to an unclear understanding of the processes controlling solute transport in streams.".

Also, we defined identifiability also in the revised introduction (lines 65 – 67): "The term "identifiability" describes whenever good model performances are constrained in a relatively narrow parameter range (identifiable parameter) or spread (non-identifiable parameter) across the entire distribution of the possible parameter values (Ward et al., 2017)."

**R3**: "There are several threshold values used to select behavioral parameter sets and subsequently to define identifiability. However, these thresholds are not sufficiently motivated and discussed, which makes it difficult for the reader to assess the results. This holds for the definitions of both the global identifiability (e.g. the top 0.1-10% of the models when assessing the CDF deviation from the 1:1 line having, the grouping based on the K-S results) and the dynamic identifiability (e.g. information content > 0.66). This leads to questions about the subjectivity of these thresholds when assessing the identifiability and how sensitive the overall assessment of the model parameters are to the chosen thresholds (e.g. in Figure 8)."

**Authors**: We clarified the choice for the selected thresholds in the following sections of the revised method section.

Lines 208 – 210 "We selected these behavioural thresholds (top 0.1% and top 10%) to assure consistency with previous work (Wagener et al., 2002; Wlostowski, 2013; Ward 2013; Ward 2017; Kelleher 2019)."

Lines 210 – 214: "Parameter identifiability is usually evaluated via visual inspection of the plots from the global identifiability analysis (Wagener et al., 2002; Wlostowski et al., 2013; Ward et al., 2017; Ward et al. 2018; Kelleher et al., 2019). To couple visual inspection with a numerical measure able to express the degree of identifiability of a certain parameter, we evaluated the two-sample Kolmogorov-Smirnov (K-S) test."

Lines 218 – 220: "Following the approach of Ouyang et al. (2014), we grouped parameter identifiability in four categories: highly identifiable ($K > 0.25$, $p \leq 0.05$), moderately identifiable ($0.1 \leq K \leq 0.25$, $p \leq 0.05$), poorly identifiable ($K < 0.1$, $p \leq 0.05$), and non-identifiable ($p > 0.05$)."

Lines 235 – 238: "To evaluate the degree of identifiability of a certain parameter on specific sections of the BTC, we grouped parameter identifiability in three categories: highly identifiable (information content $\geq 0.66$), moderately identifiable ($0.33 \leq$ information content $< 0.66$), and poorly identifiable (information content $< 0.33$)."

We also discussed the role of the selected objective function and thresholds in the revised discussions.

Lines 566 – 571 of section 4.3 reads: "The adopted iterative approach allowed to achieve parameter identifiability and to obtain physically realistic transport metrics. However, this approach is based on the specific objective function used (*RMSE*) and on the subjective thresholds to control the refinement of the parameter range for successive iterations (top 10% results for the global identifiability analysis, and information content > 0.66 for the dynamic identifiability analysis). Future work should explore the impact of the selection of the thresholds and of different objective functions on the physical realism of the modelling results and of the identifiability of the parameters."

**R3**: "The objective function used to assess the model performance, i.e. the RMSE, is based on the difference between observed and modelled BTC over a given time scale/number of observations. What are the effects on the RMSE for concentration values that may differ more than an order of magnitude over assessed time window, i.e. for high values (peak of the BTC corresponding to 50 mg/l) compared to the low values (tail of the BTC corresponding to <1 mg/l)? How does this affect (i) the global identifiability analysis and (ii) the dynamic identifiability? Previously alternative objective functions have been suggested, including RMSE with a logarithmic (e.g. Ward et al., 2018) or mixed scale (e.g.

Bottacin-Busolin et al., 2011; Riml et al., 2013), to account for the different magnitudes of the concentrations across the BTC. I suggest that the authors assess this and discuss the implications.

**Authors**: We better motivated the choice of the used objective function (*RMSE*) in the revised method section. Lines 185 – 189 read: "). *RMSE* is an equivalent form of Residual Sum of Squares (*RSS*) and Mean Absolute Error (*MAE*) objective functions that are used in OTIS-P (the most frequently adopted inverse modelling approach for TSM, Runkel, 1998) and by the dynamic identifiability analysis (Wagener et al., 2002). RMSE allowed us a comparison of our TSM results with OTIS-P and with dynamic identifiability analysis consistently to previous studies (Wlostowski et al., 2013; Ward et al., 2017)."

Also, we discussed the role of the selected objective function and thresholds in the revised discussions (lines 566 – 571 of section 4.3, see above).

**R3**: "Moreover, I miss a visual comparison of the observed and simulated BTC as a complement to the presented RMSE values, preferable using log-transformed concentrations to highlight how well the model captures the tail of BTC that is argued to be of importance for transient storage processes (e.g. lines 55-57)."

**Authors**: We added in the appendix an extra section (Appendix B) that reports a visual comparison of the simulated and observed BTCs with a short discussion on these results. Appendix B reads (line 1126 – 1135): "The figure B shows the observed BTC for the three tracer experiments plotted against the top 100 simulated BTC obtained using the proposed iterative approach. The observed poor visual fit on the tail of the BTC obtained at the end of the iterative modelling approach (Figure B1d, e, f) is controlled by two factors: (i) the modelling structure of the TSM which assumes an exponential residence time distribution and (ii) the chosen objective function. By using alternative residence time distributions, TSM proved to have a more accurate fitting on the tail of the BTC (Haggerty et al., 2002; Bottacin-Busolin et al., 2011). Also, the RMSE could not be the best objective function for addressing a model fit on the tail of BTC because it gives higher importance on the sections of the BTC with higher concentration values (peak of the BTC) compared to the sections of the BTC with low concentration values (at the tail of the BTC). As an example, the best-fitting BTC obtained at the end of the second TSM iteration (E1) shows a visually better fit on the BTC tail (Figure B2) despite the large RMSE (1.5197 mg/l)."

**R3**: "The fact that using erroneous model parameter estimates (obtained either from the literature, from a simplified model (ADE) or from a Monte-Carlo simulation with to wide parameter ranges and/or not sufficient iterations) leads to uncertainty/errors when estimating the transport metrics (Eqn 5-8) is rather intuitive. Firstly, I find (the rather long) discussion in Section 4.1 as well as the conclusion (line 24-26) and abstract (lines 21-26) about misinterpretations/uncertainty when comparing the different models "unfair". The conditions for the OTIS-MCAT simulations seems to be equivalent to the first iteration of the proposed methodology. Thus the conditions when OTIS-MCAT was used differ substantially from the 3-4 iterations with a successive refinement of the parameter ranges of the proposed methodology. I understand that the authors would like to make a point and compare the results against an existing model framework, but I think that manuscript would benefit from significantly downplaying the role of OTIS-MCAT. I would prefer a stronger focus on how the refinement of parameter ranges using the existing model framework resulted in a reduced uncertainty/increased identifiability of the model parameters."

**Authors**: One of the major points of the manuscript is the show how our approach can clearly reduce the non-identifiability of the parameters via the proposed iterative approach. Thus, we downplayed the role of OTIS-MCAT and emphasised the role of the chosen parameter range to achieve parameter identifiability by introducing another research question linked to one of the uncertainties related to random sampling approach for TSM reported in the introduction. The research question reads (line 140 – 141): "3) does the identifiability of model parameters in random sampling approaches depend to the used parameter range and on the number of parameter sets?"

This research question was targeted by a new method section (line 273 – "2.6 Number of parameter sets, parameter range, and identifiability of model parameters"), by a new result section (line 380 – "3.4 Role of the used parameter range and the number of parameter sets for the identifiability of model parameters"), by a new figure (Figure 6), and a new discussion section (line 532 – "4.3 On the importance of parameter range, parameter sets, and challenges associated to parameter identifiability in TSM").

We downplayed the role of OTIS-MCAT in the revised discussions."

**R3**: "Secondly, although the authors successfully reduced the uncertainty of the model parameters by an iterative and smart sampling of the parameter space, it was surprising to see that the results from the OTIS-P outperformed (2 out of 3 experiments) the proposed methodology when using the objective function preferred by the authors (RMSE, table 2). This is something that is not sufficiently discussed in the paper and, in my view, opens for questions when the iterative sampling procedure is needed."
**Authors**: We targeted the similarity with OTIS-P results in the revised discussion section 4.3 (lines 581 – 587): "Our simulations with OTIS-P resulted in excellent model performances for the investigated BTCs, with low *RMSE* values and with calibrated model parameters comparable to the behavioural parameter populations obtained via our global identifiability analysis (Figure 2, 3). While the obtained performances of the OTIS-P calibration are certainly specific to the investigated BTCs, the use of OTIS-P alone would have not provided enough information to address the reliability of the obtained model parameters. This, in turn, would have raised concerns about the credibility of the transport metrics obtained, eventually compromising the robustness of the derived physical process involved at the study site."

**R3**: "Could the DYNIA approach be combined with OTIS-P using a given confidence interval as input for the parameters ranges to assess the identifiability in parameter estimates from OTIS-P?
**Authors**: The idea to use the DYNIA together OTIS-P is really interesting and could be a novel modelling practice and a strong follow-up of this project.

**R3**: "I guess that similarity in performance between parameters obtained by OTIS-P and the proposed sampling procedure might connect to the objective function used to evaluate the performance (see major comment #3), the limited number of experiments (in a single reach) and how initial values (and the possibility of finding a local minimum when optimizing) were defined in the OTIS-P model."
**Authors**: We specified in the revised discussion that our results are "certainly specific to the investigated BTCs" (line 584) and "based on the specific objective function used (RMSE) and on the subjective thresholds to control the refinement of the parameter range for successive iterations (top 10% results for the global identifiability analysis, and information content > 0.66 for the dynamic identifiability analysis)" (lines 567 – 569).
We clarified that "Future work should explore the impact of the selection of the thresholds and of different objective functions on the physical realism of the modelling results and of the identifiability of the parameters" (lines 570 – 571) and that our approach should be tested also for "TSM formulation with increasing complexity, such as multiple storage zone models (Choi et al., 2002), or for TSMs considering sorption kinetics (Gooseff et al., 2005) or different residence time distribution laws such as log-normal distribution (Wörman et al., 2002), exponential plus pumping distribution (Bottacin-Busolin et al., 2011), and power law distribution (Haggerty et al., 2002)" (641 – 644).
We also reported the limitation of the adopted TSM formulation and the chosen likelihood function in the added Appendix B (see above).

**R3**: "I believe that the paper lacks a thorough discussion regarding different model representations of transient storage. Eq. 3 assumes an exponential residence time distribution (RTD) in the transient storage zone as originally defined in the TSM model (e.g. Bencala and Walters 1983). Subsequently other type of exchange models and RTDs in the transient storage zone have been introduced (e.g. Wörman et al., 2002; Haggerty et al., 2002). I see a great advantage of the proposed model framework – compared to the OTIS-MCAT and OTIS-P – to explore alternative model formulations (including multiple transient storage zones) and alternative RTDs. This flexibility, when discussed properly, could levitate the readers understanding of the usefulness of the model framework."
**Authors**: As also suggested by R1, our discussion section was implemented with exploring modelling implication for other formulations of TSM. Lines 637 – 644 read: "Our results also open developments for research seeking to increase the physical realism of the TSM and its results. Increased model complexity is both associated with a better analytical fitting to the observed BTC, but also with an increased degree of freedom of the model with a consequent reduction of parameter identifiability (Knapp and Kelleher, 2020). Our approach offers a promising flexible tool to target parameter identifiability and physical interpretation also in TSM formulation with increasing complexity, such as multiple storage zone models (Choi et al., 2002), or for TSMs considering sorption kinetics (Gooseff et

al., 2005) or different residence time distribution laws such as log-normal distribution (Wörman et al., 2002), exponential plus pumping distribution (Bottacin-Busolin et al., 2011), and power law distribution (Haggerty et al., 2002)."

**Detailed comments:**

**R3**: "Line 26-29: This is unclear, what is meant by "clear potential"?"
**Authors**: We revised the sentence as follows (lines 29 – 31): "Our results showed that coupling global identifiability analysis with dynamic identifiability analysis in iterative approach clearly increased parameter identifiability in random sampling approaches for TSMs."

**R3**: "Line 53: "The numerous contradictory outcomes", in what context? Please clarify"
**Authors**: We modified as follows (line 51): "the numerous contradictory model interpretation".

**R3**: "Line 109-111: Unclear. "to keep constant a rather identifiable parameter". Please rephrase."
**Authors**: We reformulated both the indicated section by R3 and, in agreement with R1, the entire paragraph. The revised paragraph now reads (lines 85 – 96): "First, the parameters describing the advection-dispersion process (streamflow velocity, cross-sectional area of the stream channel, and longitudinal dispersion) are known to be the best identifiable in the TSM (Ward et al., 2017). However, due to the known high interactivity among model parameters, it is generally not recommended to use a fixed value for a rather identifiable parameter, since this strategy may result in a mis-estimation of the other model parameters (Knapp and Kelleher, 2020). Constraining the values of the stream area and longitudinal dispersion proved to have a role on the identifiability of transient storage parameters (Lees et al., 2000; Kelleher et al., 2013; Ward et al., 2017). However, no study so far evaluated the role of flow velocity on the identifiability of model parameters despite the velocity parameter was often considered to be known and thus fixed to equal the velocity of the arrival time of the BTC peak (i.e. $v_{peak}$, Ward et al., 2013; Kelleher et al., 2013; Wlostowski et al., 2017; Ward et al., 2017; Ward et al., 2018). This leads to the question on how meaningful, and identifiable the transient storage parameters are when streamflow velocity is considered as a calibration parameter or is kept fixed in identifiability analysis."

**R3**: "Eq. 3: I believe that CS should be replaced by CTS in the bottom equation
**Authors**: We corrected the equation (line 180).

**R3**: "Line 216: I miss information of the width of the window in the DYNIA.
**Authors**: We implemented the method section as follows (Lines 225 – 226): "Following the approach of Wagener et al. (2002), we used a window size of three time steps (~1 min for E1 and E2, and ~15 secs for E3)."

**R3**: "Line 233-234 "The best 1% of the results were used to define its parameter space in the successive TSM iteration". Not in agreement with Figure 1 that says "New parameter range defined from the top 10% of the results". Please revise.
**Authors**: We have revised this in the text (line 256).

**R3**: "Line 245-246: vpeak is not clearly defined. Is this the time from injection to the BTC peak divided by the stream length (i.e. 55 m)? Please clarify."
**Authors**: We clarified that (lines 267 – 272): "The same approach (Figure 1) was used also in the case where $v$ was assumed fixed and equal to $v_{peak} = L/t_{peak}$, where $t_{peak}$ is the arrival time of the concentration peak. This choice was motivated by the fact that $v_{peak}$ is commonly adopted as a value for velocity in many transient storage studies (Ward et al., 2013; Kelleher et al., 2013; Wlostowski et al., 2017; Ward et al., 2017; Ward et al., 2018)."

**R3**: "Line 249: How was the set up of the OTIS-P simulations in terms of initial parameter values? Does "multiple OTIS-P iterations" mean that several initial conditions were tested to reduce the risk of ending up with in local minimum in the optimization? Please clarify."
**Authors**: We clarified. Lines 288 – 290: "We carried out multiple OTIS-P iterations starting from different initial parameter values to avoid a local minimum and interrupted the iterations when parameter values calibrated via OTIS-P changed less than 0.1% between subsequent runs (Runkel, 1998).".

**R3**: "Line 255-258: What parameter ranges were used and how many iterations were performed with the OTIS-MCAT? It is difficult to compare the results if the simulation conditions are not provided."

**Authors**: We clarified that (lines 292 – 295): "Compared to our approach, OTIS-MCAT considers Monte Carlo parameter sampling instead of LHS, velocity equal to $v_{peak}$ and it does not foresee iterative parameter sampling from results of dynamic identifiability analysis. Thus, we here indicate as "OTIS-MCAT results" the results we obtained after the first TSM iteration when $v$ was assumed fixed and equal to $v_{peak}$."

**R3**: "Line 293-295: From Figure 3, it is unclear how ATS < 5.356 m2 has information content > 0.66, Figure 3 i,j). Previously (line 223-224) it is stated that the information content is expressed as one minus the width of the 90 confidence interval, which I assume uses the entire parameter distribution. Please clarify why there is no lower bound on the confidence interval."

**Authors**: We clarified that (lines 354 – 355): "the identifiability of $A_{TS}$ increased on the tail of the BTC, where the information content was above 0.66 for $A_{TS}$ between 0.77 m$^2$ and 5.35 m$^2$ (Figure 4i, j)."

**R3**: "Moreover, although I realize that this is the first iteration, to have a transient storage area several orders of magnitude larger than the cross-sectional area of the stream makes little sense."

**Authors**: We indicated (lines 263 – 264) that "the first TSM iteration was conducted to investigate the identifiability of all the possible combinations in the feasible parameter space reported in literature and from results of ADE (Table 1)".

**R3**: "Figure 3. I suggest to show the y-axis of the alpha plot (Figure 3 g) using a log scale, due to the small values."

**Authors**: The old Figure 3, which is Figure 4 in the revised version of the manuscript, now shows the subplot 4g using a log-scale on the left y-axis.

**R3**: "Line 312-314: How much of this result can be derived to the used parameter ranges and the number of iterations? If the MC analysis would been set up differently (smaller parameter ranges, larger number of iterations), how would the result differ?"

**Authors**: We outlined the role of parameter range and number of parameter sets using a new specific research question (research question 3), method section (section 2.6), result section (section 3.4), and discussion section (section 4.3). We also introduced a new figure (Figure 6) in the revised manuscript to specifically target the role of the chosen parameter space over the role of the number of sampled parameter sets for achieving parameter identifiability.

**R3**: "Line 338: "orange boxplots" is labeled "red boxplots" in Figure 7. Please revise."

**Authors**: We revised it.